# RoPINN: Region Optimized Physics-Informed Neural Networks

**Haixu Wu, Huakun Luo, Yuezhou Ma, Jianmin Wang, Mingsheng Long**[✉]
School of Software, BNRist, Tsinghua University, China
{wuhx23,luohk19,mayz20}@mails.tsinghua.edu.cn, {jimwang,mingsheng}@tsinghua.edu.cn

## Abstract

Physics-informed neural networks (PINNs) have been widely applied to solve partial differential equations (PDEs) by enforcing outputs and gradients of deep models to satisfy target equations. Due to the limitation of numerical computation, PINNs are conventionally optimized on finite selected points. However, since PDEs are usually defined on continuous domains, solely optimizing models on scattered points may be insufficient to obtain an accurate solution for the whole domain. To mitigate this inherent deficiency of the default scatter-point optimization, this paper proposes and theoretically studies a new training paradigm as *region optimization*. Concretely, we propose to extend the optimization process of PINNs from isolated points to their continuous neighborhood regions, which can theoretically decrease the generalization error, especially for hidden high-order constraints of PDEs. A practical training algorithm, **R**egion **O**ptimized **PINN** (RoPINN), is seamlessly derived from this new paradigm, which is implemented by a straightforward but effective Monte Carlo sampling method. By calibrating the sampling process into trust regions, RoPINN finely balances optimization and generalization error. Experimentally, RoPINN consistently boosts the performance of diverse PINNs on a wide range of PDEs without extra backpropagation or gradient calculation. Code is available at this repository: `https://github.com/thuml/RoPINN`.

## 1 Introduction

Solving partial differential equations (PDEs) is the key problem in extensive areas, covering both engineering and scientific research [38, 40, 49]. Due to the inherent complexity of PDEs, they usually cannot be solved analytically [10]. Thus, a series of numerical methods have been widely explored, such as spectral methods [22, 40] or finite element methods [5, 7]. However, these numerical methods usually suffer from huge computational costs and can only obtain an approximate solution on discretized meshes [26, 43]. Given the impressive nonlinear modeling capability of deep models [4, 14], they have also been applied to solve PDEs, where physics-informed neural networks (PINNs) are proposed and have emerged as a promising and effective surrogate tool for numerical methods [44, 36, 35]. By formalizing PDE constraints (i.e. equations, initial and boundary conditions) as objective functions, the outputs and gradients of PINNs will be optimized to satisfy a certain PDE during training [36], which successfully instantiates the PDE solution as a deep model.

Although deep models have been proven to enjoy the universal approximation capability, the actual optimization process of PINNs still faces thorny challenges [8, 24, 35]. As a basic topic of PINNs, the optimization problem has been widely explored from various aspects [17, 44]. Previous methods attempt to mitigate this problem by using novel architectures to enhance model capacity [58, 3, 50], reweighting multiple loss functions for more balanced convergence [47], resampling data to improve important areas [51] or developing new optimizers to tackle the rough loss landscape [55, 37], etc.

38th Conference on Neural Information Processing Systems (NeurIPS 2024).

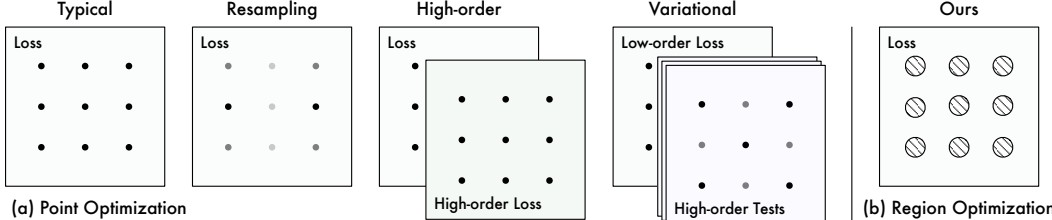

Figure 1: Comparison between previous methods and ours. Previous point optimization methods train PINNs via the loss on selected points, which is different from our region optimization paradigm.

Orthogonal to the above-mentioned methods, this paper focuses on a foundational problem, which is the objective function of PINNs. We notice that, due to the limitation of numerical calculation, it is almost impossible to optimize the loss function in the complete continuous domain. Thus, the conventional PINN loss is only defined on a series of selected points [36] (Figure 1). However, the scatter-point loss function obviously mismatches the PDE-solving objective, which is approximating the solution on a continuous domain. This mismatch may fundamentally limit the performance of PINNs. Several prior works also try to improve the canonical PINN loss function, which can be roughly categorized into the following two paradigms. One paradigm enhances the optimization by adding high-order derivatives of PDEs as a regularization term to the loss function [55]. However, calculating high-order gradients is numerically unstable and time-consuming, even with automatic differentiation in well-established deep learning frameworks [2, 34]. The other paradigm attempts to bypass the high-order derivative calculation in the PINN loss function with variational formulations [18, 19, 20]. Nevertheless, these variational methods still face difficulties in calculating the integral of deep models and will bring extra computations, thereby mainly limited to very shallow models or relying on massive sampled quadrature points and elaborative test functions [11, 57].

This paper proposes and studies a new training paradigm for PINNs as *region optimization*. As shown in Figure 1, we extend the optimization process from selected scatter points into their neighborhood regions, which can theoretically decrease the generalization error on the whole domain, especially for hidden high-order constraints of PDEs. In practice, we seamlessly transform this paradigm into a practical training algorithm, named **R**egion **O**ptimized **PINN** (RoPINN), which is implemented through simple but effective Monte Carlo sampling. In addition, to control the estimation error, we adaptively adjust the sampling region size according to the gradient variance among successive training iterations, which can constrain the sampling-based optimization into a neighborhood with low-variance loss gradients, namely *trust region*. In experiments, RoPINN demonstrates consistent and sharp improvement for diverse PINN backbones on extensive PDEs (19 different tasks) without any extra gradient calculation. Our contributions are summarized as follows:

- To mitigate the inherent deficiency of conventional PINN optimization, we propose the *region optimization* paradigm, which extends the scatter-point optimization to neighborhood regions that theoretically benefits both generalization and high-order constraints satisfaction.

- We present RoPINN for PINN training based on Monte Carlo sampling, which can effectively accomplish the region optimization. A trust region calibration strategy is proposed to reduce the gradient estimation error caused by sampling for more trustworthy optimization.

- RoPINN can consistently improve the performance of various PINN backbones (i.e. canonical and Transformer-based) on a wide range of PDEs without extra gradient calculation.

## 2 Preliminaries

A PDE with equation constraints, initial (ICs) and boundary conditions (BCs) can be formalized as

$$\mathcal{F}(u)(\boldsymbol{x}) = 0, \boldsymbol{x} \in \Omega; \ \mathcal{I}(u)(\boldsymbol{x}) = 0, \boldsymbol{x} \in \Omega_0; \ \mathcal{B}(u)(\boldsymbol{x}) = 0, \boldsymbol{x} \in \partial\Omega, \tag{1}$$

where $\mathcal{F}, \mathcal{I}, \mathcal{B}$ denote the PDE equations, ICs and BCs respectively [6]. $u : \mathbb{R}^{d+1} \to \mathbb{R}^m$ is the target PDE solution. $\boldsymbol{x} \in \Omega \subseteq \mathbb{R}^{d+1}$ represents the input coordinate, which is usually a composition of spatial and temporal positions, namely $\boldsymbol{x} = (x_1, \cdots, x_d, t)$. $\Omega_0$ corresponds to the $t = 0$ situation.

Correspondingly, the PINN loss function (*point optimization*) is typically defined as follows [17, 36]:

$$\mathcal{L}(u_\theta) = \frac{\lambda_\Omega}{N_\Omega} \sum_{i=1}^{N_\Omega} \|\mathcal{F}(u_\theta)(\boldsymbol{x}_i)\|^2 + \frac{\lambda_{\Omega_0}}{N_{\Omega_0}} \sum_{i=1}^{N_{\Omega_0}} \|\mathcal{I}(u_\theta)(\boldsymbol{x}_i)\|^2 + \frac{\lambda_{\partial\Omega}}{N_{\partial\Omega}} \sum_{i=1}^{N_{\partial\Omega}} \|\mathcal{B}(u_\theta)(\boldsymbol{x}_i)\|^2, \quad (2)$$

where $u_\theta$ represents the neural network parameterized by $\theta$. $N_\Omega, N_{\Omega_0}, N_{\partial\Omega}$ are the numbers of sampled points in $\Omega, \Omega_0, \partial\Omega$ respectively. $\lambda_*$ is the corresponding loss weight. Note that there is an additional data loss term in Eq. (2) when we can access the ground truth of some points [36]. Since we mainly focus on PDE constraints throughout this paper, we omit the data loss term in the above formalization, which is still maintained in our experiments. In this paper, we try to improve PINN solving by defining a new surrogate loss in place of the canonical definition of PINN loss in Eq. (2). In contrast, the relevant literature mainly improves the objective function in two different directions as follows. Appendix F provides a more comprehensive discussion on other relative topics.

**High-order regularization** The first direction is to add the high-order constraints of PDEs as regularization terms to the loss function [55]. Specifically, since PDEs are sets of identical relations, suppose that the solution $u$ is a $K$-order differential function, Eq. (1) can naturally derive a branch of high-order equations, where the $k$-th derivative for the $j$-th dimension is $\frac{\partial^k}{\partial x_j^k}\mathcal{F}(u)(\boldsymbol{x}) = 0$, $\boldsymbol{x} \in \Omega, 1 \le j \le (d+1), 1 \le k \le K$, corresponding to the following regularization:

$$\mathcal{L}_{k,j}^{\text{reg}}(u_\theta) = \frac{\lambda_{k,j}}{N_{k,j}} \sum_{i=1}^{N_{k,j}} \left\| \frac{\partial^k}{\partial x_j^k}\mathcal{F}(u_\theta)(\boldsymbol{x}_i) \right\|^2, \quad (3)$$

where $N_{k,j}$ denotes the number of sampled points with weight $\lambda_{k,j}$. Although this design can explicitly enhance the model performance in satisfying high-order constraints, the calculation of high-order derivatives can be extremely time-consuming and unstable [39]. Thus, in practice, the previous methods [55, 31] only consider a small value of $K$. In the next sections, we will prove that RoPINN can naturally incorporate high-order constraints. Besides, as presented in Eq. 3, this paradigm still optimizes PINNs on scattered points, while this paper extends optimization to neighborhood regions.

**Variational formulation** As a classical tool in traditional PDE solvers, the variational formulation is widely used to reduce the smoothness requirements of the approximate solution [42]. Concretely, the target PDEs are multiplied with a set of predefined test functions $\{v_1, \cdots, v_M\}$ and then the PDE equation term of the loss function is transformed as follows [18, 19, 20]:

$$\mathcal{L}^{\text{equ}}(u_\theta) = \frac{1}{M} \sum_{k=1}^{M} \left\| \left\langle \mathcal{F}^{(x_j)}(u_\theta)(\boldsymbol{x}), v_k(\boldsymbol{x}) \right\rangle \Big|_{\partial_{(x_j)}\Omega} - \int_\Omega \left\langle \mathcal{F}^{(x_j)}(u_\theta)(\boldsymbol{x}), \frac{\partial}{\partial x_j} v_k(\boldsymbol{x}) \right\rangle \mathrm{d}\boldsymbol{x} \right\|^2, \quad (4)$$

where $\mathcal{F}^{(x_j)}$ defines the antiderivative of $\mathcal{F}$ on the $j$-th dimension. Using integrals by parts, the derivative operation in $\mathcal{F}$ is transferred to test functions $\{v_k\}_{k=1}^M$, thereby able to bypass high-order derivatives. However, the integral on $\Omega$ is still hard to compute, which requires massive quadrature points for approximation [18]. Besides, test function selection requires extra manual effort and will bring $M$ times computation costs [57]. In contrast, RoPINN does not require test functions and will not bring extra gradient calculations. Also, RoPINN employs a trust region calibration strategy to limit the optimization in low-variance regions, which can control the estimation error of sampling.

## 3 Method

As aforementioned, we propose the region optimization paradigm to extend the optimization from scatter points to a series of corresponding neighborhood regions. This section will first present the region optimization and its theoretical benefits in both reducing generalization error and satisfying high-order PDE constraints. Then, we implement RoPINN in a simple but effective sampling-based way, along with a trust region calibration strategy to control the sampling estimation error.

### 3.1 Region Optimization

For clarity, we record the point optimization loss defined in Eq. (2) at $\boldsymbol{x}$ as $\mathcal{L}(u_\theta, \boldsymbol{x})$, where $\boldsymbol{x} \in \Omega \cup \Omega_0 \cup \partial\Omega$ denotes the point selected from inner domain, initial state or boundaries. We adopt $\mathcal{S}$ to denote the finite set of selected points. Then Eq. (2) can be simplified as $\mathcal{L}(u_\theta, \mathcal{S}) = \frac{1}{|\mathcal{S}|} \sum_{\boldsymbol{x} \in \mathcal{S}} \mathcal{L}(u_\theta, \boldsymbol{x})$.

Correspondingly, we define the objective function of our *region optimization* innovatively as

$$\mathcal{L}_r^{\text{region}}(u_\theta, \mathcal{S}) = \frac{1}{|\mathcal{S}|} \sum_{\boldsymbol{x} \in \mathcal{S}} \mathcal{L}_r^{\text{region}}(u_\theta, \boldsymbol{x}) = \frac{1}{|\Omega_r| \times |\mathcal{S}|} \sum_{\boldsymbol{x} \in \mathcal{S}} \int_{\Omega_r} \mathcal{L}(u_\theta, \boldsymbol{x} + \boldsymbol{\xi}) \mathrm{d}\boldsymbol{\xi}, \tag{5}$$

where $\Omega_r = [0, r]^{(d+1)}$ represents the extended neighborhood region with hyperparameter $r$. Although this definition seems to require more sampling points than point optimization, we can develop an efficient algorithm to implement it without adding sampling points (see next section). Besides, this formalization also provides us with a convenient theoretical analysis framework. Next, we will discuss the theoretical properties of the two optimization paradigms. All proofs are in Appendix A.

**Generalization bound**  Here we discuss the *generalization error in expectation* [13], which is independent of the point selection, thereby quantifying the error of PINN optimization more rigorously.

**Definition 3.1.** *The generalization error in expectation of model trained on dataset $\mathcal{S}$ is defined as*

$$\mathcal{E}_{\text{gen}} = \left| \mathbb{E}_{\mathcal{S}, \mathcal{A}} \left[ \mathcal{L} \left( u_{\mathcal{A}(\mathcal{S})}, \Omega \right) - \mathcal{L} \left( u_{\mathcal{A}(\mathcal{S})}, \mathcal{S} \right) \right] \right|, \tag{6}$$

*where $\mathcal{A}$ denotes the training algorithm and $\mathcal{A}(\mathcal{S})$ represents the optimized model parameters.*

**Assumption 3.2.** *The loss function $\mathcal{L}$ is $L$-Lipschitz and $\beta$-smooth with respect to model parameters, which means that $\forall \boldsymbol{x} \in \Omega$ the following inequalities hold:*

$$\|\mathcal{L}(u_{\theta_1}, \boldsymbol{x}) - \mathcal{L}(u_{\theta_2}, \boldsymbol{x})\| \le L\|\theta_1 - \theta_2\|, \quad \|\nabla_\theta \mathcal{L}(u_{\theta_1}, \boldsymbol{x}) - \nabla_\theta \mathcal{L}(u_{\theta_2}, \boldsymbol{x})\| \le \beta\|\theta_1 - \theta_2\|. \tag{7}$$

**Theorem 3.3 (Point optimization).** *Suppose that the loss function $\mathcal{L}$ is $L$-Lipschitz-$\beta$-smooth for $\theta$. If we run stochastic gradient descent with step size $\alpha_t$ at the $t$-th step for $T$ iterations, we have that:*

*(1) If $\mathcal{L}$ is convex for $\theta$ and $\alpha_t \le \frac{2}{\beta}$, then $\mathcal{E}_{\text{gen}} \le \frac{2L^2}{|\mathcal{S}|} \sum_{t=1}^{T} \alpha_t$ (proved by [13, 52]).*

*(2) If $\mathcal{L}$ is bounded by a constant $C$ for all $\theta, \boldsymbol{x}$ and is non-convex for $\theta$ with monotonically non-increasing step sizes $\alpha_t \le \frac{1}{\beta t}$, then $\mathcal{E}_{\text{gen}} \le \frac{C}{|\mathcal{S}|} + \frac{2L^2(T-1)}{\beta(|\mathcal{S}|-1)}$ (tighter bound than [13, 52]).*

**Lemma 3.4.** *If $\mathcal{L}$ is bounded for all $\theta, \boldsymbol{x}$ and is convex, $L$-Lipschitz-$\beta$-smooth with respect to model parameters $\theta$, then $\mathcal{L}_r^{\text{region}}$ is also bounded for all $\theta, \boldsymbol{x}$ and convex, $L$-Lipschitz-$\beta$-smooth for $\theta$.*

**Theorem 3.5 (Region optimization).** *Suppose that the point optimization loss function $\mathcal{L}$ is $L$-Lipschitz and $\beta$-smooth for $\theta$. If we run stochastic gradient descent with step size $\alpha_t$ for $T$ iterations based on region optimization loss $\mathcal{L}_r^{\text{region}}$ in Eq. (5), the generalization error in expectation satisfies:*

*(1) If $\mathcal{L}$ is convex for $\theta$ and $\alpha_t \le \frac{2}{\beta}$, then $\mathcal{E}_{\text{gen}} \le (1 - \frac{|\Omega_r|}{|\Omega|}) \frac{2L^2}{|\mathcal{S}|} \sum_{t=1}^{T} \alpha_t$.*

*(2) If $\mathcal{L}$ is bounded by a constant $C$ for all $\theta, \boldsymbol{x}$ and is non-convex for $\theta$ with monotonically non-increasing step sizes $\alpha_t \le \frac{1}{\beta t}$, then $\mathcal{E}_{\text{gen}} \le \frac{C}{|\mathcal{S}|} + \frac{2L^2(T-1)}{\beta(|\mathcal{S}|-1)} - JL(\frac{|\Omega_r|}{|\Omega|})^2$, where $J$ is a finite number that depends on the training property at the several beginning iterations.*

*Proof.* Based on the Lipschitz assumption, $\mathcal{E}_{\text{gen}}$ can be bounded by a term relating to the expectation of distance between parameter $\theta$ optimized from different training sets. The region optimization paradigm brings a more "consistent" gradient optimization direction than point optimization at each iteration, thereby benefiting the generalization property. See Appendix A.3 for complete proof. ☐

From Theorems 3.3 and 3.5, we can observe that region optimization can reduce the generalization error $\mathcal{E}_{\text{gen}}$. Furthermore, the region optimization theorem also provides a more general theoretical framework. For example, the conventional point optimization is equivalent to the case of $\Omega_r = 0$, where only one single point is selected for each region. For another extreme case, enlarging the region size to the whole domain (i.e. $\Omega_r = \Omega$), Eq. (5) is equivalent to directly optimizing the loss defined on $\Omega$, where the generalization error will be reduced to zero. Unfortunately, this ideal situation cannot be satisfied in practice, since Eq. (5) requires precise calculation of the integral over the whole domain. More discussions of practical implementation are deferred to the next section.

**High-order PDE constraints**  In our proposed region optimization (Eq. (5)), the integral operation on the input domain can also relax the smoothness requirements of the loss function $\mathcal{L}$. For example, without any additional assumption of the smoothness of $\mathcal{L}(u_\theta, \boldsymbol{x})$ on $\boldsymbol{x}$, we can directly derive the generalization error for the first-order loss $\frac{\partial}{\partial x_j} \mathcal{L}_r^{\text{region}}(u_\theta, \boldsymbol{x})$ on the $j$-th dimension as follows.

**Algorithm 1** Region Optimized PINN (RoPINN)

---

**Input:** number of iterations $T$, number of past iterations $T_0$ retained to estimate the trust region, default region size $r$, trust region calibration value $\sigma_0 = 1$, and initial PINN parameters $\theta_0$.
**Output:** optimized PINN parameters $\theta_T$.
Initialize an empty buffer to record gradients as **g**.
**for** $t = 0$ **to** $T$ **do**
    *// Region Optimization with Monte Carlo Approximation*
    Sample points from neighborhood regions: $\mathcal{S}' = \{\boldsymbol{x}_i + \boldsymbol{\xi}_i\}_{i=1}^{|S|}, \boldsymbol{x}_i \in \mathcal{S}, \boldsymbol{\xi}_i \sim U[0, \frac{r}{\sigma_t}]^{(d+1)}$
    Calculate loss function $\mathcal{L}_t = \mathcal{L}\left(u_{\theta_t}, \mathcal{S}'\right)$
    Update $\theta_t$ to $\theta_{t+1}$ with optimizer (Adam [21], L-BFGS [27], etc) to minimize loss function $\mathcal{L}_t$
    *// Trust Region Calibration*
    Record the gradient of parameters $g_t$ throughout optimization
    Update gradient buffer **g** by adding $g_t$ and keeping the latest $T_0$ elements
    Trust region calibration with $\sigma_{t+1} = \|\sigma(\mathbf{g})\|$
**end for**

---

**Corollary 3.6 (Region optimization for first-order constraints).** *Suppose that $\mathcal{L}$ is bounded by $C$ for all $\theta, \boldsymbol{x}$ and is $L$-Lipschitz and $\beta$-smooth for $\theta$. If we run stochastic gradient method based on first-order $j$-th dimension loss function $\frac{\partial}{\partial x_j} \mathcal{L}_r^{\text{region}}$ for $T$ iterations, the generalization error in Theorem 3.5(2) still holds when we adopt the monotonically non-increasing step size $\alpha_t \leq \frac{1}{2\beta t}$.*

Corollary 3.6 implies that the integral on the input domain in region optimization can help training PINNs with high-order constraints, which is valuable for high-order PDEs, such as wave equations. In contrast, this valuable property cannot be achieved by the classic point optimization. See Example 3.7.

**Example 3.7 (Point optimization fails in optimizing with first-order constraints).** *Under the same assumption with Corollary 3.6, we cannot obtain the Lipschitz and smoothness property of $\frac{\partial}{\partial x_j} \mathcal{L}(u_\theta, \boldsymbol{x})$. For example, suppose that $\mathcal{L}(u_\theta, \boldsymbol{x}) = |\theta^\mathsf{T} \sqrt{\boldsymbol{x}}|, \boldsymbol{x} \in [0, 1]^{(d+1)}$, which is 1-Lipschitz-1-smooth. However, $\nabla_\theta \frac{\partial}{\partial x_j} \mathcal{L}(u_\theta, \boldsymbol{x})$ is unbounded when $\boldsymbol{x} \to \boldsymbol{0}$, thereby not Lipschitz constant.*

### 3.2 Practical Algorithm

Derived from our theoretical insights of region optimization, we implement RoPINN as a practical training algorithm. As elaborated in Algorithm 1, RoPINN involves the following two iterative steps: Monte Carlo approximation and trust region calibration, where the former can efficiently approximate the optimization objective and the latter can effectively control the estimation error. Next, we will discuss the details and convergence properties of RoPINN. All proofs can be found in Appendix B.

**Monte Carlo approximation** Note that the region integral in Eq. (5) cannot be directly calculated, so we adopt a straightforward implementation based on the Monte Carlo approximation. Concretely, to approximate the gradient descent on the region loss $\mathcal{L}_r^{\text{region}}$, we uniformly sample one point within the region $\Omega_r$ for the gradient descent at each iteration, whose expectation is equal to the gradient descent of the original region optimization in Eq. (5):

$$\mathbb{E}_{\boldsymbol{\xi} \sim U(\Omega_r)} \left[\nabla_\theta \mathcal{L}(u_\theta, \boldsymbol{x} + \boldsymbol{\xi})\right] = \nabla_\theta \mathcal{L}_r^{\text{region}}(u_\theta, \boldsymbol{x}). \tag{8}$$

In addition to efficiently approximating region optimization without adding sampling points, our proposed sampling-based strategy is also equivalent to a high-order loss function, especially for the first-order term, which is essential in practice [55]. Concretely, with Taylor expansion, we have that:

$$\mathbb{E}_{\boldsymbol{\xi} \sim U(\Omega_r)} \left(\nabla_\theta \mathcal{L}(u_\theta, \boldsymbol{x} + \boldsymbol{\xi})\right) = \mathbb{E}_{\boldsymbol{\xi} \sim U(\Omega_r)} \left(\nabla_\theta \mathcal{L}(u_\theta, \boldsymbol{x}) + \nabla_\theta(\boldsymbol{\xi}^\mathsf{T} \mathcal{L}_1(u_\theta, \boldsymbol{x})) + \mathcal{O}(\|\boldsymbol{\xi}\|^2)\right), \tag{9}$$

where $\Omega_r = [0, r]^{d+1}$, and $\mathcal{L}_1$ represents the first order of loss function, namely $\frac{\partial}{\partial \boldsymbol{x}} \mathcal{L}(u_\theta, \boldsymbol{x})$.

**Theorem 3.8 (Convergence rate).** *Suppose that there exists a constant $H$, s.t. $\forall \boldsymbol{v}$ and $\forall \boldsymbol{x} \in \Omega$, $\left|\boldsymbol{v}^\mathsf{T} \nabla_\theta \mathcal{L}_r^{\text{region}}(u_\theta, \boldsymbol{x}) \boldsymbol{v}\right| \leq H \|\boldsymbol{v}\|^2$. If the step size $\alpha_t = \frac{1}{\sqrt{t+1}}$ decreases over time for $T$ iterations, the region optimization based on Monte Carlo approximation will converge at the speed of*

$$\mathbb{E}\left[\left\|\nabla_\theta \mathcal{L}_r^{\text{region}}(u_\theta, \boldsymbol{x})\right\|^2\right] \leq \mathcal{O}\left(\frac{1}{\sqrt{T}}\right). \tag{10}$$

**Theorem 3.9** (**Gradient estimation error**). *The estimation error of gradient descent between Monte Carlo approximation and the original region optimization satisfies:*

$$\mathbb{E}_{\boldsymbol{\xi} \sim U(\Omega_r)} \left[ \left\| \nabla_\theta \mathcal{L}(u_\theta, \boldsymbol{x} + \boldsymbol{\xi}) - \nabla_\theta \mathcal{L}_r^{\text{region}}(u_\theta, \boldsymbol{x}) \right\|^2 \right]^{\frac{1}{2}} = \left\| \sigma_{\boldsymbol{\xi} \sim U(\Omega_r)} \left( \nabla_\theta \mathcal{L}(u_\theta, \boldsymbol{x} + \boldsymbol{\xi}) \right) \right\|, \quad (11)$$

*where $\sigma$ represents the standard deviation of gradients in region $\Omega_r$.*

**Trust region calibration**   Although the expectation of the Monte Carlo sampling is equal to region optimization as shown in Eq. (8), this design will also bring estimation error in practice (Theorem 3.9). A large estimation error will cause unstable training and further affect convergence. To ensure a reliable gradient descent, we propose to control the sampling region size $r$ towards a trustworthy value, namely *trust region calibration*. Unlike the notion in optimization [56], here *trust region* is used to define the area of input domain where the variance of loss gradients for different points is relatively small. Formally, we adjust the region size in inverse proportion to gradient variance:

$$r \propto \frac{1}{\left\| \sigma_{\boldsymbol{\xi} \sim U(\Omega_r)} \left( \nabla_\theta \mathcal{L}(u_\theta, \boldsymbol{x} + \boldsymbol{\xi}) \right) \right\|}. \quad (12)$$

In practice, we initialize the trust region size as a default value $r$ and calculate the gradient estimation error during the training process for calibration (Algorithm 1). However, the calculation of the standard deviation of gradients usually requires multiple samples, which will bring times of computation overload. In pursuit of a practical algorithm, we propose to adopt the gradient variance among several successive iterations as an approximation. Similar ideas are widely used in deep learning optimizers, such as Adam [21] and AdaGrad [48], which adopt multi-iteration statistics as the momentum of gradient descent. The approximation process is guaranteed by the following theoretical results.

**Lemma 3.10** (**Trust region one-iteration approximation**). *Suppose that loss function $\mathcal{L}$ is $L$-Lipschitz and $\beta$-smooth for $\theta$ and the $t$-th step parameter is $\theta_t$. Two gradient difference sequences between successive iterations, $\|\nabla_\theta \mathcal{L}(u_{\theta_t}, \boldsymbol{z}_1) - \nabla_\theta \mathcal{L}(u_{\theta_{t-1}}, \boldsymbol{z}_2)\|$ and $\|\nabla_\theta \mathcal{L}(u_{\theta_t}, \boldsymbol{z}_1) - \nabla_\theta \mathcal{L}(u_{\theta_t}, \boldsymbol{z}_2)\|$, share the same limit, as the difference of the two sequences is dominated by the following inequality:*

$$\left| \left\| \nabla_\theta \mathcal{L}(u_{\theta_t}, \boldsymbol{z}_1) - \nabla_\theta \mathcal{L}(u_{\theta_{t-1}}, \boldsymbol{z}_2) \right\| - \left\| \nabla_\theta \mathcal{L}(u_{\theta_t}, \boldsymbol{z}_1) - \nabla_\theta \mathcal{L}(u_{\theta_t}, \boldsymbol{z}_2) \right\| \right| \leq \beta L \alpha_{t-1}, \quad (13)$$

*where $\alpha_{t-1}$ represents the step size at the $(t-1)$-th iteration, which approaches 0 as $t$ tends to $\infty$.*

**Theorem 3.11** (**Trust region multi-iteration approximation**). *Suppose that loss function $\mathcal{L}$ is $L$-Lipschitz and $\beta$-smooth for $\theta$ and the learning rate $\alpha_t \leq \frac{1}{\beta L}$ converges to zero over time $t$, then the estimation error can be approximated by the variance of optimization gradients in multiple successive iterations. Given hyperparameter $T_0$, our multi-iteration approximation is guaranteed by*

$$\lim_{t \to \infty} \sigma \left( \{ \nabla_\theta \mathcal{L}(u_{\theta_{t-i+1}}, \boldsymbol{z}_i) \}_{i=1}^{T_0} \right) = \sigma \left( \{ \nabla_\theta \mathcal{L}(u_{\theta_t}, \boldsymbol{z}_i) \}_{i=1}^{T_0} \right). \quad (14)$$

It is worth noting that, as presented in Algorithm 1, since the gradient of each iteration has already been on the shelf, our design will not bring any extra gradient or backpropagation calculation in comparison with point optimization. Besides, our algorithm is not limited to a certain optimizer, and in general, we can effectively obtain the gradients of parameters by retrieving the computation graph.

**Balance between generalization and optimization**   Recall that in Theorem 3.5, we observe that a larger region size will benefit the generalization error, while Theorem 3.9 demonstrates that too large region size will also cause unstable training because it will result in excessive gradient estimation error of Monte Carlo sampling in our implementation. The above analysis reveals the underlying trade-off between generalization and optimization of PINN models, which is formally stated below.

**Theorem 3.12** (**Region Optimization with gradient estimation error**). *Based on the same assumptions in Theorem 3.5 but optimizing the PINN model with the approximate region optimization loss $\mathcal{L}_r^{\text{approx}}(u_\theta, \boldsymbol{x}) = \nabla_\theta \mathcal{L}(u_\theta, \boldsymbol{x} + \boldsymbol{\xi}), \boldsymbol{\xi} \sim U(\Omega_r)$ for $T$ iterations, we further denote the upper bound of gradient estimation error as $\mathcal{E}_{r,\text{grad}} = \max_{t \leq T} \|\nabla_\theta \mathcal{L}_r^{\text{approx}} - \nabla_\theta \mathcal{L}_r^{\text{region}}\|$, then $\mathcal{E}_{\text{gen}}$ satisfies:*

*(1) If $\mathcal{L}$ is convex for $\theta$ and $\alpha_t \leq \frac{2}{\beta}$, $\mathcal{E}_{\text{gen}} \leq \big( \underbrace{(1 - |\Omega_r|/|\Omega|) L}_{\text{inversely proportional to } |\Omega_r|} + \underbrace{\mathcal{E}_{r,\text{grad}}}_{\text{generally} \propto |\Omega_r|} \big) \frac{2L}{|\mathcal{S}|} \sum_{t=1}^{T} \alpha_t.*

*(2) If $\mathcal{L}$ is bounded by a constant $C$ and is non-convex for $\theta$ with monotonically non-increasing step sizes $\alpha_t \leq \frac{1}{\beta t}$, then $\mathcal{E}_{\text{gen}} \leq \frac{C}{|\mathcal{S}|} + \frac{2L^2(T-1)}{\beta(|\mathcal{S}|-1)} \underbrace{-J'L(|\Omega_r|/|\Omega|)^2}_{\text{inversely proportional to } |\Omega_r|} + \underbrace{J'\mathcal{E}_{r,\text{grad}}(1 + |\Omega_r|/|\Omega|)}_{\text{generally} \propto |\Omega_r|},*

*where $J'$ is a finite number that depends on the training property at the several beginning iterations.*

*Proof.* In contrast to Theorem 3.5, here the gradient estimation error will bring extra optimization discrepancy between different training sets. See Appendix B.4 for complete proof. □

Based on Theorem 3.12, we have pinpointed that classical point optimization ($\Omega_r = 0$) and sampling points globally ($\Omega_r = \Omega$) discussed in Theorem 3.5 correspond to two extreme cases. The former makes $\mathcal{E}_{r,\mathrm{grad}} = 0$ but cannot reduce the generalization error, while the latter holds a large gradient estimation error. Thus, neither case can yield perfect generalization for PINNs . In contrast, the design for calibrating trust regions in RoPINN provides an adaptive strategy to better balance generalization and optimization, which can adjust the region size according to multi-iteration training stability.

## 4 Experiments

To verify the effectiveness and generalizability of our proposed RoPINN, we experiment with a wide range of PDEs, covering diverse physics processes and a series of advanced PINN models.

**Benchmarks** For a comprehensive evaluation, we experiment with four benchmarks: 1D-Reaction, 1D-Wave, Convection and PINNacle [12]. The first three benchmarks are widely acknowledged in investigating the optimization property of PINNs [47, 37]. Especially, 1D-Reaction and Convection are highly challenging and have

Table 1: Summary of benchmarks. *Dimension* means the input space and *Derivative* is the highest derivative order.

| Benchmark | Dimension | Derivative | Property |
|---|---|---|---|
| 1D-Reaction | 1D+Time | 1 (e.g. $\frac{\partial u}{\partial x}$) | Failure modes [24] |
| 1D-Wave | 1D+Time | 2 (e.g. $\frac{\partial^2 u}{\partial x^2}$) | / |
| Convection | 1D+Time | 1 (e.g. $\frac{\partial u}{\partial x}$) | Failure modes [24] |
| PINNacle [12] | 1D~5D+Time | 1~2 (e.g. $\frac{\partial^2 u}{\partial x^2}$) | 16 different tasks |

been used to demonstrate "PINNs failure modes" [24, 33]. As for PINNacle [12], it is a comprehensive family of 20 tasks, including diverse PDEs, e.g. Burgers, Poisson, Heat, Navier-Stokes, Wave and Gray-Scott equations in 1D to 5D space and on complex geometries. In this paper, to avoid meaningless comparisons, we remove the tasks that all the methods fail and leave 16 tasks.

**Base models** To verify the generalizability of RoPINN among different PINN models, we experiment with five base models, including canonical PINN [36], activation function enhanced models: QRes [3] and FLS [50], Transformer-based model PINNsFormer [58] and advanced physics-informed backbone KAN [28]. PINNsFormer [58] and KAN [28] are the most advanced PINN models.

**Baselines** As stated before, this paper mainly focuses on the objective function of PINNs. Thus, we only include the gradient-enhanced method gPINN [55] and variational-based method vPINN [18] as baselines. Notably, there are diverse training strategies for PINNs focusing on other aspects than objective function, such as sampling-based RAR [51] or neural tangent kernel (NTK) approaches [47]. We also experimented with them and demonstrated that they contribute orthogonally to RoPINN.

**Implementations** In RoPINN (Algorithm 1), we select the multi-iteration hyperparameter $T_0$ from $\{5, 10\}$ and set the initial region size $r = 10^{-4}$ for all datasets, where the trust region size will be adaptively adjusted to fit the PDE property during training. For 1D-Reaction, 1D-Wave and Convection, we follow [58] and train the model with L-BFGS optimizer [27] for 1,000 iterations. As for PINNacle, we strictly follow their official configuration [12] and train the model with Adam [21] for 20,000 iterations. Besides, for simplicity and fair comparison, we set the weights of PINN loss as equal, that is $\lambda_* = 1$ in Eq. (2). Canonical loss formalized in Eq. (2), relative L1 error (rMAE) and relative L2 error (rMSE) are recorded. All experiments are implemented in PyTorch [34] and trained on a single NVIDIA A100 GPU. See Appendix C for more implementation details.

### 4.1 Main Results

**Results** As shown in Table 2, we investigate the effectiveness of RoPINN on diverse tasks and base models and compare it with two well-acknowledged PINN objectives. Here are two key observations.

RoPINN can consistently boost performance on all benchmarks, justifying its generality on PDEs and base models. Notably, since the PDEs under evaluation are quite diverse, especially for PINNacle (Table 1), it is extremely challenging to obtain such a consistent improvement. We can find that the previous high-order regularization and variational-based methods could yield negative effects in many cases. For example, gPINN [55] performs badly on 1D-Wave, which may be due to second-order derivatives in the wave equation. Besides, vPINN [18] also fails in 1D-Reaction and QRes.

Table 2: Comparison between RoPINN and other objective functions (gPINN [55] and vPINN [18]) under different base models. Metrics for PINNacle [12] are the proportions of improved tasks over 16 tasks, where full results can be found in Appendix E. A lower loss, rMAE or rMSE indicates better performance. For clarity, we highlight the value with `blue` if it surpasses the vanilla PINN and the best is in bold. *Promotion* refers to the relative promotion of RoPINN over the vanilla version.

| Base Model | Objective | 1D-Reaction | | | 1D-Wave | | | Convection | | | PINNacle (16 tasks) | |
|---|---|---|---|---|---|---|---|---|---|---|---|---|
| | | Loss | rMAE | rMSE | Loss | rMAE | rMSE | Loss | rMAE | rMSE | rMAE | rMSE |
| PINN [36] | Vanilla | 2.0e-1 | 0.982 | 0.981 | 1.9e-2 | 0.326 | 0.335 | 1.6e-2 | 0.778 | 0.840 | - | - |
| | gPINN | 2.0e-1 | 0.978 | 0.978 | 2.8e-2 | 0.399 | 0.399 | 3.1e-2 | 0.890 | 0.935 | 18.8% | 18.8% |
| | vPINN | 2.3e-1 | 0.985 | 0.982 | 7.3e-3 | 0.162 | 0.173 | 1.1e-2 | 0.663 | 0.743 | 25.0% | 25.0% |
| | RoPINN | **4.7e-5** | **0.056** | **0.095** | **1.5e-3** | **0.063** | **0.064** | **1.0e-2** | **0.635** | **0.720** | **93.8%** | **100.0%** |
| | Promotion | 99% | 94% | 90% | 92% | 80% | 80% | 25% | 18% | 14% | | |
| QRes [3] | Vanilla | 2.0e-1 | 0.979 | 0.977 | 9.8e-2 | 0.523 | 0.515 | 4.2e-2 | 0.925 | 0.959 | - | - |
| | gPINN | 2.1e-2 | 0.984 | 0.984 | 1.3e-1 | 0.785 | 0.781 | 1.6e-1 | 1.111 | 1.222 | 12.5% | 12.5% |
| | vPINN | 2.2e-2 | 0.999 | 1.000 | 1.0e-1 | 0.709 | 0.721 | 5.5e-2 | 0.941 | 0.966 | 12.5% | 12.5% |
| | RoPINN | **9.0e-6** | **0.007** | **0.013** | **1.7e-2** | **0.309** | **0.321** | **1.2e-2** | **0.819** | **0.870** | **81.3%** | **81.3%** |
| | Promotion | 99% | 99% | 99% | 83% | 41% | 38% | 71% | 11% | 9% | | |
| FLS [50] | Vanilla | 2.0e-1 | 0.984 | 0.985 | 3.6e-3 | 0.102 | 0.119 | 1.2e-2 | 0.674 | 0.771 | - | - |
| | gPINN | 2.0e-1 | 0.978 | 0.979 | 9.2e-2 | 0.500 | 0.489 | 3.8e-1 | 0.913 | 0.949 | 12.5% | 18.8% |
| | vPINN | 2.1e-1 | 1.000 | 0.994 | 2.1e-3 | 0.069 | 0.069 | 1.1e-2 | 0.688 | 0.765 | 25.0% | 18.8% |
| | RoPINN | **2.2e-5** | **0.022** | **0.039** | **1.5e-4** | **0.016** | **0.017** | **9.6e-4** | **0.173** | **0.197** | **81.3%** | **87.5%** |
| | Promotion | 99% | 98% | 96% | 96% | 84% | 86% | 99% | 74% | 74% | | |
| PINNs-Former [58] | Vanilla | 3.0e-6 | 0.015 | 0.030 | 1.4e-2 | 0.270 | 0.283 | 3.7e-5 | 0.023 | 0.027 | - | - |
| | gPINN | 1.5e-6 | 0.009 | 0.018 | OOM | OOM | OOM | 3.7e-2 | 0.914 | 0.950 | 0.0% | 0.0% |
| | vPINN | 1.6e-4 | 0.065 | 0.124 | 4.5e-2 | 0.411 | 0.400 | 5.1e-5 | 0.016 | 0.022 | 0.0% | 0.0% |
| | RoPINN | **1.0e-6** | **0.007** | **0.017** | **6.5e-3** | **0.165** | **0.172** | **1.2e-5** | **0.005** | **0.006** | **100.0%** | **100%** |
| | Promotion | 66% | 53% | 43% | 54% | 39% | 39% | 68% | 78% | 78% | | |
| KAN [28] | Vanilla | 7.3e-5 | 0.031 | 0.061 | 9.2e-2 | 0.499 | 0.489 | 5.8e-2 | 0.922 | 0.954 | - | - |
| | gPINN | 2.9e-4 | 0.030 | 0.061 | 2.6e-1 | 1.131 | 1.110 | 1.2e-1 | 1.006 | 1.041 | 31.3% | 31.3% |
| | vPINN | 2.1e-1 | 0.998 | 0.996 | 9.0e-2 | 0.498 | 0.487 | 2.5e-2 | 0.853 | 0.853 | 43.8% | 43.8% |
| | RoPINN | **4.9e-5** | **0.026** | **0.051** | **9.6e-3** | **0.177** | **0.191** | **2.2e-2** | **0.805** | **0.801** | **100%** | **93.8%** |
| | Promotion | 33% | 16% | 16% | 89% | 65% | 61% | 62% | 13% | 16% | | |

As we stated in Table 1, 1D-Reaction and Convection are hard to optimize, so-called "PINNs failure modes" [24, 33]. In contrast, empowered by RoPINN, PINNs can mitigate this thorny challenge to some extent. Specifically, with RoPINN, canonical PINN [36], QRes [3] and FLS [50] achieve more than 90% improvements in 1D-Reaction. Besides, RoPINN can further enhance the performance of PINNsFormer [58] and KAN [28], which have already performed well in 1D-Recation or Convection, further verifying its effectiveness in helping PINN optimization.

**Combining with other strategies** Since RoPINN mainly focuses on the objective function design, it can be integrated seamlessly and directly with other strategies. As shown in Table 3, we experiment with the widely-used loss-reweighting method NTK [47] and data-sampling strategy RAR [51]. Although NTK can consistently improve the performance, it will take extra computation costs due to the calculation of neural tangent kernels [15]. Based on NTK, our RoPINN can obtain better results with slightly more time cost. As for RAR, it performs unstable in different tasks, while RoPINN can also boost it. These results verify the orthogonal contribution and favorable efficiency of RoPINN w.r.t. other methods.

Table 3: Adding RoPINN to other strategies based on PINN. *Time* is for every $10^2$ training iterations on 1D-Reaction.

| Method rMSE | 1D-Reaction | 1D-Wave | Convection | Time (s) |
|---|---|---|---|---|
| PINN [36] | 0.981 | 0.335 | 0.840 | 18.47 |
| +gPINN [55] | 0.978 | 0.399 | 0.935 | 37.91 |
| +vPINN [18] | 0.982 | 0.173 | 0.743 | 38.78 |
| +RoPINN | **0.095** | **0.064** | **0.720** | 20.04 |
| +NTK [47] | 0.098 | 0.149 | 0.798 | 27.99 |
| +NTK+RoPINN | **0.052** | **0.023** | **0.693** | 29.96 |
| +RAR [51] | 0.981 | 0.126 | 0.771 | 19.71 |
| +RAR+RoPINN | **0.080** | **0.030** | **0.695** | 20.89 |

## 4.2 Algorithm Analysis

**Initial region size in Algorithm 1** To provide an intuitive understanding of RoPINN, we plot the curves of training statistics in Figure 2, including temporally adjusted region size $\log(\frac{r}{\sigma_t})$, train loss, and test performance. From Figure 2(a), we can find that even though we initialize the region size as distinct values, RoPINN will progressively adjust the trust region size to similar values during training. This indicates that our algorithm can capture a potential "balance point" between training stability and generalization error, where the fluctuation of trust region size reveals the balancing process. Further, as shown in Figure 2(b-c), if $r$ is initialized as a value closer to the balance point (e.g. 1e-4 and 1e-5 in this case), then the training process will converge faster. And too large a region size (e.g. 1e-3) will decrease the convergence speed due to the optimization noise (Theorem 3.9).

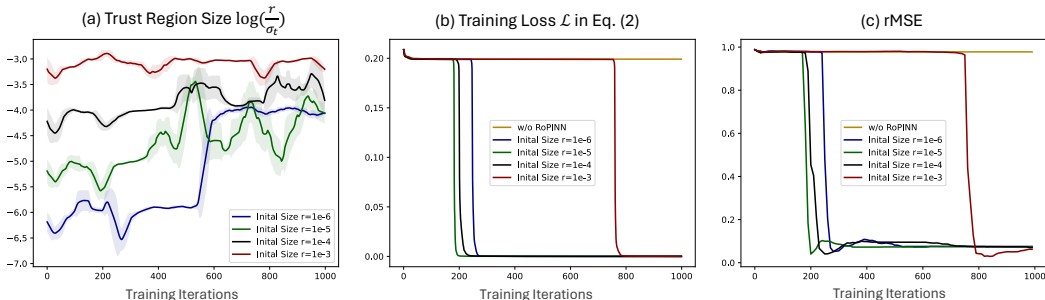

Figure 2: Optimization of canonical PINN [36] on the 1D-Reaction under different region sizes. To highlight the region size change, we adopt the moving average over time and mark the temporal standard deviation with shadow. The steep training loss is caused by the learning difficulty of PDE.

**Number of sampling points in Eq.** (8) For efficiency, RoPINN only samples one point within the trust region to approximate the region gradient descent. However, it is worth noticing that sampling more points will make the approximation in Eq. (8) more accurate, leading to a lower gradient estimation error. Further, since RoPINN employs an adaptive strategy to adjust region $\Omega_r$, a lower gradient estimation error will also make the optimization process adapt to a larger region size $r$. Therefore, we observe in Figure 3(a-b) that sampling more points will also increase the finally learned region size $r$ and speed up the convergence. In addition, Figure 3(c) shows that adding sampled points can also improve the final performance, which has also been theoretically justified in Theorem 3.12 that the upper bound of generalization error is inversely proportion to gradient estimation error.

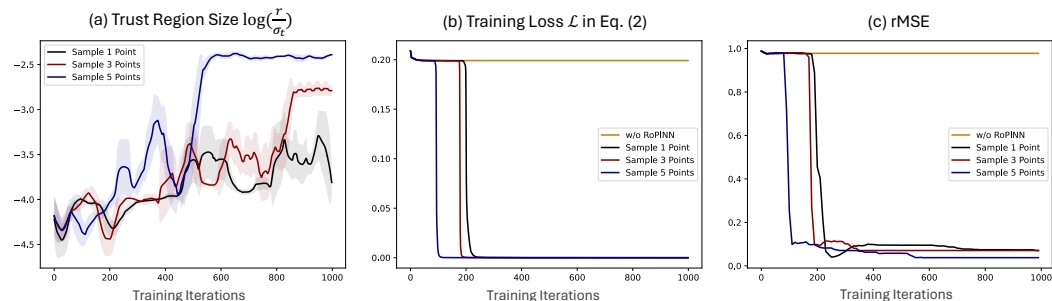

Figure 3: Optimization of canonical PINN [36] on the 1D-Reaction under different sample points.

**Efficiency analysis** As we discussed above, sampling more points can benefit the final performance, while we choose only sample one point as the default setting of RoPINN in the spirit of boosting PINNs without extra backpropagation or gradient calculation, which has already achieved significant promotion w.r.t. original PINNs (Table 2). To provide a more comprehensive understanding of algorithm property, we plot the efficiency-performance curve in Figure 4, where we can obtain the following observations. Firstly, computation costs will grow linearly when adding points. Secondly, more points will bring better performance but will saturate around 10 points, where the performance fluctuations of 9, 13, and 30 points are within three times the standard deviations (Appendix D.3).

**Ablations** To verify the effectiveness of our design in RoPINN, we present ablations in Figure 5. It is observed that although we only sample one point, even fixed-size region optimization can also boost

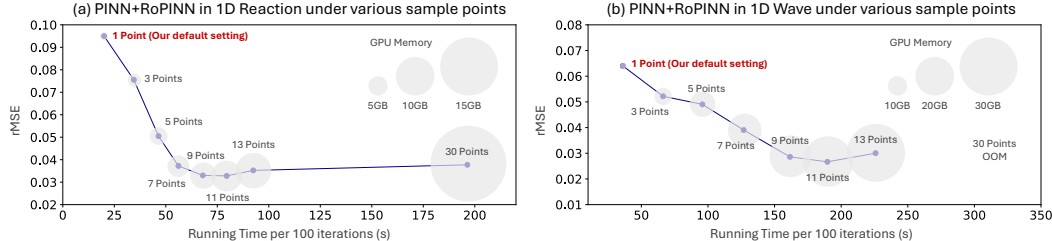

Figure 4: Efficiency and model performance w.r.t. number of samples. Note that the default setting of RoPINN is just sampling one point, which will not bring extra gradient calculation costs.

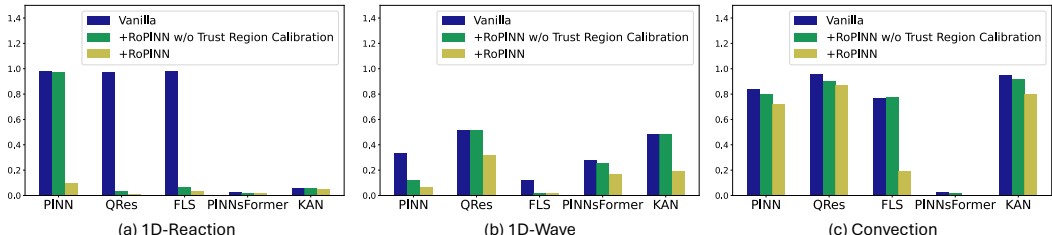

Figure 5: Ablation study of RoPINN on different PDEs and diverse base models. rMSE is recorded.

the performance of PINNs in most cases, demonstrating the effectiveness of introducing "region" to PINN optimization. However, as illustrated in Theorem 3.9, the sampling process may also cause gradient estimation error, so the relative promotion is inconsistent and unstable among different PDEs and base models. With our proposed trust region calibration, we can obtain a more significant and consistent improvement, indicating that achieving a better balance between optimization and generalization (formally stated in Theorem 3.12) performs an essential role in training PINN models.

**Loss landscape**    Previous research [24] has studied why PINN cannot solve the Convection equation and found that it is not caused by the limited model capacity but by the hard-to-optimize loss landscape. Here we also provide a loss landscape visualization in Figure 6, which is obtained by perturbing the trained model along the directions of the first two dominant Hessian eigenvectors [24, 25, 54]. We can find that vanilla PINN optimized by PINN loss in Eq. (2) presents sharp cones. In contrast, empowered by RoPINN, the loss landscape is significantly smoothed. This visualization intuitively interprets why RoPINN can mitigate "PINN failure modes". See Appendix D for more results.

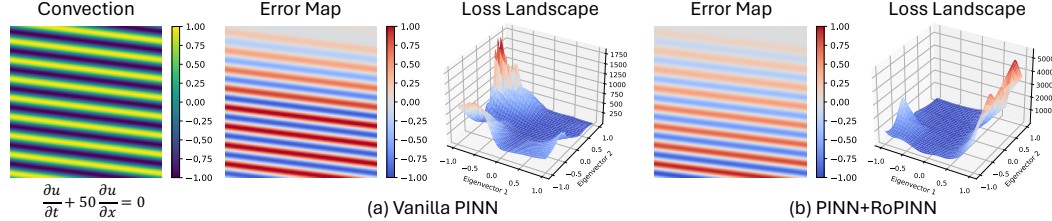

Figure 6: Loss landscape of RoPINN and vanilla PINNs on the Convection equation. *Error Map* refers to the distance between model prediction and the accurate solution, i.e. $(u_\theta - u)$.

## 5 Conclusion

This paper presents and analyzes a new PINN optimization paradigm: region optimization. Going beyond previous scatter-point optimization, we extend the optimization from selected points to their neighborhood regions. Based on this idea, RoPINN is implemented as a simple but effective training algorithm, where an efficient Monte Carlo approximation process is used along with a trust region calibration strategy to control the gradient estimation error caused by sampling, theoretically manifesting a better balance of generalization and optimization. In addition to theoretical advantages, RoPINN can consistently boost the performance of various PINN models without extra backpropagation or gradient calculation, demonstrating favorable efficiency, training stability and general capability.

## Acknowledgments and Disclosure of Funding

This work was supported by the National Natural Science Foundation of China (U2342217 and 62021002), the BNRist Project, and the National Engineering Research Center for Big Data Software.

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

# A Generalization Analysis in Section 3.1

This section will present the proofs for the theorems in Section 3.1.

## A.1 Proof for Point Optimization Generalization Error (Theorem 3.3)

The proof for the convex case is derived from previous papers [13, 52] under the Assumption 3.2. We derive a more compact upper bound for generalization error in expectation for the non-convex setting.

**Lemma A.1.** *Given two finite sets of selected points $\mathcal{S} = (\boldsymbol{x}_1, \cdots, \boldsymbol{x}_N)$ and $\mathcal{S}' = (\boldsymbol{x}'_1, \cdots, \boldsymbol{x}'_N)$, let $\mathcal{S}^{(i)} = (\boldsymbol{x}_1, \cdots, \boldsymbol{x}_{i-1}, \boldsymbol{x}'_i, \boldsymbol{x}_{i+1}, \cdots, \boldsymbol{x}_N)$ be the set that is identical to $\mathcal{S}$ except the $i$-th element, the generalization error in expectation is equal to the expectation of the error difference between these two sets, which can be formalized as follows:*

$$\mathcal{E}_{\mathrm{gen}} = \left| \mathbb{E}_{\mathcal{S},\mathcal{S}',\mathcal{A}} \left[ \frac{1}{N} \sum_{i=1}^{N} \mathcal{L}(\mathcal{A}(\mathcal{S}^{(i)}), \boldsymbol{x}'_i) - \frac{1}{N} \sum_{i=1}^{N} \mathcal{L}(\mathcal{A}(\mathcal{S}), \boldsymbol{x}'_i) \right] \right|. \tag{15}$$

*Proof.* Directly deriving from the in-domain loss, we have:

$$\begin{aligned}
\mathbb{E}_{\mathcal{S},\mathcal{A}}\left[\mathcal{L}(\mathcal{A}(\mathcal{S}),\mathcal{S})\right] &= \mathbb{E}_{\mathcal{S},\mathcal{A}}\left[\frac{1}{N}\sum_{i=1}^{N}\mathcal{L}(\mathcal{A}(\mathcal{S}),\boldsymbol{x}_i)\right] \\
&= \mathbb{E}_{\mathcal{S},\mathcal{S}',\mathcal{A}}\left[\frac{1}{N}\sum_{i=1}^{N}\mathcal{L}(\mathcal{A}(\mathcal{S}^{(i)}),\boldsymbol{x}'_i)\right] \qquad \text{(Expectation on } \boldsymbol{x}'_i) \\
&= \mathbb{E}_{\mathcal{S},\mathcal{S}',\mathcal{A}}\left[\frac{1}{N}\sum_{i=1}^{N}\mathcal{L}(\mathcal{A}(\mathcal{S}),\boldsymbol{x}'_i)\right] + \delta \\
&= \mathbb{E}_{\mathcal{S},\mathcal{A}}\left[\mathcal{L}(\mathcal{A}(\mathcal{S}),\Omega)\right] + \delta.
\end{aligned} \tag{16}$$

Then, according to Definition 3.1, we have:

$$\mathcal{E}_{\mathrm{gen}} = \delta = \mathbb{E}_{\mathcal{S},\mathcal{S}',\mathcal{A}}\left[\frac{1}{N}\sum_{i=1}^{N}\mathcal{L}(\mathcal{A}(\mathcal{S}^{(i)}),\boldsymbol{x}'_i) - \frac{1}{N}\sum_{i=1}^{N}\mathcal{L}(\mathcal{A}(\mathcal{S}),\boldsymbol{x}'_i)\right]. \tag{17}$$

$\square$

**Lemma A.2 (Convex case).** *Given the stochastic gradient method with an update rule as $G_{\alpha,\boldsymbol{x}}(\theta) = \theta - \alpha \nabla_\theta \mathcal{L}(\theta, \boldsymbol{x})$ and $\mathcal{L}$ is convex in $\theta$, then for $\alpha \le \frac{2}{\beta}$, we have $\|G_{\alpha,\boldsymbol{x}}(\theta_1) - G_{\alpha,\boldsymbol{x}}(\theta_2)\| \le \|\theta_1 - \theta_2\|$.*

*Proof.* For clarity, we denote $g = \|\nabla_\theta \mathcal{L}(\theta_1, \boldsymbol{x}) - \nabla_\theta \mathcal{L}(\theta_2, \boldsymbol{x})\|$. Then we have:

$$\begin{aligned}
\|G_{\alpha,\boldsymbol{x}}(\theta_1) - G_{\alpha,\boldsymbol{x}}(\theta_2)\|^2 &= \|\theta_1 - \theta_2 - \alpha(\nabla_\theta \mathcal{L}(\theta_1, \boldsymbol{x}) - \nabla_\theta \mathcal{L}(\theta_2, \boldsymbol{x}))\|^2 \\
&= \|\theta_1 - \theta_2\|^2 - 2\alpha\left(\nabla_\theta \mathcal{L}(\theta_1, \boldsymbol{x}) - \nabla_\theta \mathcal{L}(\theta_2, \boldsymbol{x})\right)^\mathsf{T}(\theta_1 - \theta_2) + \alpha^2 g^2 \\
&\le \|\theta_1 - \theta_2\|^2 - \frac{2\alpha}{\beta}g^2 + \alpha^2 g^2 \qquad \text{(Convexity and Assumption 3.2)} \\
&\le \|\theta_1 - \theta_2\|^2. \qquad\qquad (\alpha \le \frac{2}{\beta})
\end{aligned} \tag{18}$$

$\square$

**Convex setting** Next, we will give the proof for the convex case of Theorem 3.3(1).

*Proof.* According to Lemma A.1, we attempt to bound the generalization error in expectation $\mathcal{E}_{\mathrm{gen}}$ by analyzing the error difference between two selected sample sets. Denote $\mathcal{S}$ and $\mathcal{S}'$ as two identical sample sets of size $|\mathcal{S}|$ except for one sample. Suppose that with the stochastic gradient method

on these two sets, we can obtain two optimization trajectories $\{\theta_t\}_{t=1}^T$ and $\{\theta_t'\}_{t=1}^T$ respectively. According to Assumption 3.2, we have the following inequality:

$$\mathbb{E}\left[|\mathcal{L}(u_{\theta_t}, \boldsymbol{x}) - \mathcal{L}(u_{\theta_t'}, \boldsymbol{x})|\right] \leq L\mathbb{E}\left[\|\theta_t - \theta_t'\|\right]. \tag{19}$$

We assume two optimization trajectories, both obtained under the same random update rule and random permutation rule. Note that at the $t$-th step, with probability $(1 - \frac{1}{|\mathcal{S}|})$, the example selected by the stochastic gradient method is the same in both $\mathcal{S}$ and $\mathcal{S}'$. As for the other $\frac{1}{|\mathcal{S}|}$ probability, we have to deal with different selected samples. Thus, according to Lemma A.2, we have:

$$\mathbb{E}\left[\|\theta_{t+1} - \theta_{t+1}'\|\right] = (1 - \frac{1}{|\mathcal{S}|})\mathbb{E}\left[\|G_{\alpha_t, \boldsymbol{x}}(\theta_t) - G_{\alpha_t, \boldsymbol{x}}(\theta_t')\|\right] + \frac{1}{|\mathcal{S}|}\mathbb{E}\left[\|G_{\alpha_t, \boldsymbol{x}}(\theta_t) - G_{\alpha_t, \boldsymbol{x}'}(\theta_t')\|\right]$$

$$\leq (1 - \frac{1}{|\mathcal{S}|})\mathbb{E}[\|\theta_t - \theta_t'\|] + \frac{1}{|\mathcal{S}|}\mathbb{E}\left[\|\theta_t - \theta_t'\| + \|\alpha_t\nabla_\theta\mathcal{L}(u_\theta, \boldsymbol{x}) - \alpha_t\nabla_\theta\mathcal{L}(u_\theta, \boldsymbol{x}')\|\right]. \tag{20}$$

Due to the $L$-Lipschitz assumption of $\mathcal{L}$, the gradient $\nabla_\theta\mathcal{L}(u_\theta, \boldsymbol{x})$ is uniformly smaller than $L$, then:

$$\mathbb{E}\left[\|\theta_{t+1} - \theta_{t+1}'\|\right] \leq \mathbb{E}\left[\|\theta_t - \theta_t'\|\right] + \frac{2\alpha_t L}{|\mathcal{S}|}. \tag{21}$$

In summary, since both optimization trajectories start from the same initialization, namely $\theta_0 = \theta_0'$, the following inequality holds:

$$\mathbb{E}\left[|\mathcal{L}(u_{\theta_T}, \boldsymbol{x}) - \mathcal{L}(u_{\theta_T'}, \boldsymbol{x})|\right] \leq \frac{2L^2}{|\mathcal{S}|}\sum_{t=1}^T \alpha_t. \tag{22}$$

From Lemma A.1, we have $\mathcal{E}_{\text{gen}} \leq \frac{2L^2}{|\mathcal{S}|}\sum_{t=1}^T \alpha_t$. $\qquad\qquad\square$

**Lemma A.3** (**Non-convex case**). *Given the stochastic gradient method with an update rule as* $G_{\alpha, \boldsymbol{x}}(\theta) = \theta - \alpha\nabla_\theta\mathcal{L}(\theta, \boldsymbol{x})$, *then we have* $\|G_{\alpha, \boldsymbol{x}}(\theta_1) - G_{\alpha, \boldsymbol{x}}(\theta_2)\| \leq (1 + \alpha\beta)\|\theta_1 - \theta_2\|$.

*Proof.* This inequality can be easily obtained from the following:

$$\begin{aligned}
\|G_{\alpha, \boldsymbol{x}}(\theta_1) - G_{\alpha, \boldsymbol{x}}(\theta_2)\| &= \|\theta_1 - \theta_2 - \alpha(\nabla_\theta\mathcal{L}(\theta_1, \boldsymbol{x}) - \nabla_\theta\mathcal{L}(\theta_2, \boldsymbol{x}))\| \\
&= \|\theta_1 - \theta_2\| + \alpha\|\nabla_\theta\mathcal{L}(\theta_1, \boldsymbol{x}) - \nabla_\theta\mathcal{L}(\theta_2, \boldsymbol{x})\| \\
&\leq (1 + \alpha\beta)\|\theta_1 - \theta_2\|. \qquad\text{(Assumption 3.2)}
\end{aligned} \tag{23}$$

$\qquad\qquad\square$

**Non-convex setting**  Finally, we will give the proof for the non-convex case in Theorem 3.3(2).

*Proof.* We also consider the optimization trajectory $\{\theta_t\}_{t=1}^T$ and $\{\theta_t'\}_{t=1}^T$ from $\mathcal{S}$ and $\mathcal{S}'$, which are identical except for one element. We assume two optimization trajectories, both obtained under the same random update rule and random permutation rule. Let $\delta_t = \|\theta_t - \theta_t'\|$ and $t_0 \in \{1, \cdots, |\mathcal{S}|\}$ be a considered iteration. Here, $t \leq |\mathcal{S}|$ because for $t > |\mathcal{S}|$, we must have $\delta_{t_0} \neq 0$. Then we have:

$$\begin{aligned}
\mathbb{E}\left[|\mathcal{L}(u_{\theta_T}, \boldsymbol{x}) - \mathcal{L}(u_{\theta_T'}, \boldsymbol{x})|\right] &= \mathbb{P}(\delta_{t_0} = 0)\mathbb{E}\left[|\mathcal{L}(u_{\theta_T}, \boldsymbol{x}) - \mathcal{L}(u_{\theta_T'}, \boldsymbol{x})||\delta_{t_0} = 0\right] \\
&\quad + \mathbb{P}(\delta_{t_0} \neq 0)\mathbb{E}\left[|\mathcal{L}(u_{\theta_T}, \boldsymbol{x}) - \mathcal{L}(u_{\theta_T'}, \boldsymbol{x})||\delta_{t_0} \neq 0\right] \\
&\leq L\mathbb{E}\left[\|\theta_T - \theta_T'\||\delta_{t_0} = 0\right] + \mathbb{P}(\delta_{t_0} \neq 0)C \qquad\text{(Upper bound of }\mathcal{L}\text{)} \\
&= \frac{Ct_0}{|\mathcal{S}|} + L\mathbb{E}\left[\|\theta_T - \theta_T'\||\delta_{t_0} = 0\right].
\end{aligned} \tag{24}$$

Similar to the convex case, we analyze the expectation of parameter difference in the $(t+1)$-th iteration as follows. Since $\alpha \leq \frac{1}{\beta t}$, then we have:

$$\mathbb{E}\left[\|\theta_{t+1} - \theta'_{t+1}\| | \delta_{t_0} = 0\right] \leq (1 - \frac{1}{|\mathcal{S}|})(1 + \frac{1}{t})\mathbb{E}\left[\|\theta_t - \theta'_t\|\right] + \frac{1}{|\mathcal{S}|}\mathbb{E}\left[\|\theta_t - \theta'_t\|\right] + \frac{2L}{\beta t|\mathcal{S}|}$$

$$\leq (1 + \frac{1}{t} - \frac{1}{t|\mathcal{S}|})\mathbb{E}[\delta_t] + \frac{2L}{\beta t|\mathcal{S}|}$$

$$\leq \exp(\frac{1}{t} - \frac{1}{t|\mathcal{S}|})\mathbb{E}[\delta_t] + \frac{2L}{\beta t|\mathcal{S}|}. \qquad (1 + x \leq \exp(x))$$

$$(25)$$

Accumulating the above in equations recursively, we have:

$$\mathbb{E}\left[\|\theta_T - \theta'_T\| | \delta_{t_0} = 0\right] \leq \sum_{t=t_0+1}^{T} \left\{\Pi_{k=t+1}^{T}\exp\left(\frac{1}{t} - \frac{1}{t|\mathcal{S}|}\right)\right\} \frac{2L}{\beta t|\mathcal{S}|}$$

$$\leq \sum_{t=t_0+1}^{T} \exp\left((1 - \frac{1}{|\mathcal{S}|})\log\frac{T}{t}\right)\frac{2L}{\beta t|\mathcal{S}|} \qquad (\sum_{k=t+1}^{T}\frac{1}{k} \leq \log\frac{T}{t})$$

$$= \frac{2L}{\beta|\mathcal{S}|}T^{1-\frac{1}{|\mathcal{S}|}} \sum_{t=t_0+1}^{T} t^{-(1-\frac{1}{|\mathcal{S}|})-1}$$

$$\leq \frac{2L}{\beta|\mathcal{S}|}T^{1-\frac{1}{|\mathcal{S}|}}\frac{1}{1-\frac{1}{|\mathcal{S}|}}\left(t_0^{-(1-\frac{1}{|\mathcal{S}|})} - T^{-(1-\frac{1}{|\mathcal{S}|})}\right). \quad \text{(Integral approximation)}$$

$$(26)$$

Organizing the above inequalities, we have:

$$\mathbb{E}\left[\|\theta_T - \theta'_T\| | \delta_{t_0} = 0\right] \leq \frac{2L}{\beta(|\mathcal{S}|-1)}\left(\frac{T}{t_0}\right)^{1-\frac{1}{|\mathcal{S}|}} - \frac{2L}{\beta(|\mathcal{S}|-1)}$$

$$\leq \frac{2L}{\beta(|\mathcal{S}|-1)}\left(\frac{T}{t_0}\right) - \frac{2L}{\beta(|\mathcal{S}|-1)}.$$

$$(27)$$

According to Eq. (24), for arbitrary $T \geq 1$, we just choose $t_0 = 1$, then

$$\mathbb{E}\left[|\mathcal{L}(u_{\theta_T}, \boldsymbol{x}) - \mathcal{L}(u_{\theta'_T}, \boldsymbol{x})|\right] \leq \frac{C}{|\mathcal{S}|} + \frac{2L^2(T-1)}{\beta(|\mathcal{S}|-1)}. \tag{28}$$

$\square$

This boundary is tighter than [13, 52], where the latter omits the $\frac{2L}{\beta(|\mathcal{S}|-1)}$ term in Eq. (27).

## A.2 Proof for Properties of Region Optimization (Lemma 3.4)

Since $\mathcal{L}_r^{\text{region}}$ is defined as a region integral of $\mathcal{L}$, Lemma 3.4 can be easily obtained by:

$$\text{Bounded:} \quad \mathcal{L}_r^{\text{region}}(u_\theta, \boldsymbol{x}) = \frac{1}{|\Omega_r|}\int_{\Omega_r}\mathcal{L}(u_\theta, \boldsymbol{x} + \boldsymbol{\xi})\mathrm{d}\boldsymbol{\xi} \leq \max_{\theta, \boldsymbol{x}}\mathcal{L}(u_\theta, \boldsymbol{x})$$

$$\text{Convexity:} \quad \left(\nabla_\theta\mathcal{L}_r^{\text{region}}(u_{\theta_1}, \boldsymbol{x}) - \nabla_\theta\mathcal{L}_r^{\text{region}}(u_{\theta_2}, \boldsymbol{x})\right)^{\mathsf{T}}(\theta_1 - \theta_2)$$

$$= \frac{1}{|\Omega_r|}\int_{\Omega_r}(\nabla_\theta\mathcal{L}(u_{\theta_1}, \boldsymbol{x} + \boldsymbol{\xi}) - \nabla_\theta\mathcal{L}(u_{\theta_2}, \boldsymbol{x} + \boldsymbol{\xi}))^{\mathsf{T}}(\theta_1 - \theta_2)\mathrm{d}\boldsymbol{\xi} \geq 0$$

$$\text{Lipschitz:} \quad \|\mathcal{L}_r^{\text{region}}(u_{\theta_1}, \boldsymbol{x}) - \mathcal{L}_r^{\text{region}}(u_{\theta_2}, \boldsymbol{x})\| \tag{29}$$

$$\leq \frac{1}{|\Omega_r|}\int_{\Omega_r}\|\mathcal{L}(u_{\theta_1}, \boldsymbol{x} + \boldsymbol{\xi}) - \mathcal{L}(u_{\theta_2}, \boldsymbol{x} + \boldsymbol{\xi})\|\mathrm{d}\boldsymbol{\xi} \leq L\|\theta_1 - \theta_2\|$$

$$\text{Smoothness:} \quad \|\nabla_\theta\mathcal{L}_r^{\text{region}}(u_{\theta_1}, \boldsymbol{x}) - \nabla_\theta\mathcal{L}_r^{\text{region}}(u_{\theta_2}, \boldsymbol{x})\|$$

$$\leq \frac{1}{|\Omega_r|}\int_{\Omega_r}\|\nabla_\theta\mathcal{L}(u_{\theta_1}, \boldsymbol{x} + \boldsymbol{\xi}) - \nabla_\theta\mathcal{L}(u_{\theta_2}, \boldsymbol{x} + \boldsymbol{\xi})\|\mathrm{d}\boldsymbol{\xi} \leq \beta\|\theta_1 - \theta_2\|.$$

## A.3 Proof for Region Optimization Generalization Error (Theorem 3.5)

Similar to the proof in Appendix A.1, we will discuss the generalization error on region optimization.

**Convex setting**   Firstly, we would like to prove the convex case.

*Proof.* According to Lemma 3.4, the region loss also holds the convexity, Lipschitz and smoothness properties, which ensures that Lemma A.2 and A.3 still work for $\mathcal{L}_r^{\text{region}}$. We also focus on the selected sample sets $\mathcal{S}$ and $\mathcal{S}'$, which are identical except for one element. Thus, at $t$-step, the following equation is satisfied:

$$\mathbb{E}\left[\|\theta_{t+1} - \theta'_{t+1}\|\right] = (1 - \frac{1}{|\mathcal{S}|})\mathbb{E}\left[\|G_{\alpha_t,\boldsymbol{x}}^{\text{region}}(\theta_t) - G_{\alpha_t,\boldsymbol{x}}^{\text{region}}(\theta'_t)\|\right] + \frac{1}{|\mathcal{S}|}\mathbb{E}\left[\|G_{\alpha_t,\boldsymbol{x}}^{\text{region}}(\theta_t) - G_{\alpha_t,\boldsymbol{x}'}^{\text{region}}(\theta'_t)\|\right]$$

$$\leq (1 - \frac{1}{|\mathcal{S}|})\mathbb{E}\left[\|\theta_t - \theta'_t\|\right] + \frac{1}{|\mathcal{S}|}\mathbb{E}\left[\|G_{\alpha_t,\boldsymbol{x}}^{\text{region}}(\theta_t) - G_{\alpha_t,\boldsymbol{x}'}^{\text{region}}(\theta'_t)\|\right],$$

$$(30)$$

where $G_{\alpha,\boldsymbol{x}}^{\text{region}}(\theta) = \theta - \alpha\nabla_\theta\mathcal{L}_r^{\text{region}}(\theta, \boldsymbol{x})$. As for the second item on the right part, we also consider the upper bound of $\nabla_\theta\mathcal{L}_r^{\text{region}}(u_\theta, \boldsymbol{x})$. However, different from point-wise optimization, $\boldsymbol{x} + \Omega_r$ could overlap with $\boldsymbol{x}' + \Omega_r$. For clarity, we define the overlapped area as $\Omega_{\text{in}}$, whose size is larger than zero when $\boldsymbol{x}'$ fall into the area centered at $\boldsymbol{x}$ with size $2^{(d+1)}|\Omega_r|$. Actually, due to the boundary of $\Omega$, we cannot always ensure $(\boldsymbol{x}' + \Omega_r) \subset \Omega$. Thus, for simplification, we assume that the domain $\Omega$ can be projected to a torus, where the out-of-domain samples will be re-included to $\Omega$.

Further, $\mathbb{E}_{\boldsymbol{x},\boldsymbol{x}'\in\Omega, \text{ s.t. } |\Omega_{\text{in}}|=0}$ is simplified as $\mathbb{E}_{\Omega_{\text{in}}=0}$ and $\mathbb{E}_{\boldsymbol{x},\boldsymbol{x}'\in\Omega, \text{ s.t. } |\Omega_{\text{in}}|>0}$ is shorted as $\mathbb{E}_{\Omega_{\text{in}}>0}$. And the operator $\mathbb{I}$ is defined as $\mathbb{I}(x) = \max(0, \min(1, x))$ Thus, we can obtain the estimation for the difference between the updated model parameters through the following derivations:

$$\mathbb{E}\left[\|G_{\alpha_t,\boldsymbol{x}}^{\text{region}}(\theta_t) - G_{\alpha_t,\boldsymbol{x}'}^{\text{region}}(\theta'_t)\|\right]$$

$$= \mathbb{I}\left(\frac{|\Omega| - 2^{(d+1)}|\Omega_r|}{|\Omega|}\right)\mathbb{E}_{\Omega_{\text{in}}=0}\left[\|G_{\alpha_t,\boldsymbol{x}}^{\text{region}}(\theta_t) - G_{\alpha_t,\boldsymbol{x}'}^{\text{region}}(\theta'_t)\|\right]$$

$$+ \mathbb{I}\left(\frac{2^{(d+1)}|\Omega_r|}{|\Omega|}\right)\mathbb{E}_{\Omega_{\text{in}}>0}\left[\|G_{\alpha_t,\boldsymbol{x}}^{\text{region}}(\theta_t) - G_{\alpha_t,\boldsymbol{x}'}^{\text{region}}(\theta'_t)\|\right]$$

$$\leq \mathbb{I}\left(\frac{|\Omega| - 2^{(d+1)}|\Omega_r|}{|\Omega|}\right)\mathbb{E}_{\Omega_{\text{in}}=0}\left[\|\theta_t - \theta'_t\| + 2\alpha_t L\right]$$

$$+ \mathbb{I}\left(\frac{2^{(d+1)}|\Omega_r|}{|\Omega|}\right)\mathbb{E}_{\Omega_{\text{in}}>0}\left[\left\|\theta_t - \theta'_t - \left(\alpha_t\frac{1}{|\Omega_r|}\int_{\Omega_r}\nabla_\theta\mathcal{L}(u_{\theta_t}, \boldsymbol{x} + \boldsymbol{\xi})d\boldsymbol{\xi} - \alpha_t\frac{1}{|\Omega_r|}\int_{\Omega_r}\nabla_\theta\mathcal{L}(u_{\theta'_t}, \boldsymbol{x}' + \boldsymbol{\xi})d\boldsymbol{\xi}\right)\right\|\right]$$

$$\leq \mathbb{I}\left(\frac{|\Omega| - 2^{(d+1)}|\Omega_r|}{|\Omega|}\right)\mathbb{E}_{\Omega_{\text{in}}=0}\left[\|\theta_t - \theta'_t\| + 2\alpha_t L\right]$$

$$+ \mathbb{I}\left(\frac{2^{(d+1)}|\Omega_r|}{|\Omega|}\right)\mathbb{E}_{\Omega_{\text{in}}>0}\left[\left\|\theta_t - \theta'_t - \left(\alpha_t\frac{1}{|\Omega_r|}\int_{\Omega_{\text{in}}}\nabla_\theta\mathcal{L}(u_{\theta_t}, \boldsymbol{x} + \boldsymbol{\xi})d\boldsymbol{\xi} - \alpha_t\frac{1}{|\Omega_r|}\int_{\Omega_{\text{in}}}\nabla_\theta\mathcal{L}(u_{\theta'_t}, \boldsymbol{x} + \boldsymbol{\xi})d\boldsymbol{\xi}\right)\right\|\right]$$

$$+ \mathbb{I}\left(\frac{2^{(d+1)}|\Omega_r|}{|\Omega|}\right)\mathbb{E}_{\Omega_{\text{in}}>0}\left[\frac{|\Omega_r| - |\Omega_{\text{in}}|}{|\Omega_r|}2\alpha_t L\right]$$

$$\leq \mathbb{I}\left(\frac{|\Omega| - 2^{(d+1)}|\Omega_r|}{|\Omega|}\right)\mathbb{E}_{\Omega_r=0}\left[\|\theta_t - \theta'_t\| + 2\alpha_t L\right] + \mathbb{I}\left(\frac{2^{(d+1)}|\Omega_r|}{|\Omega|}\right)\mathbb{E}_{\Omega_{\text{in}}>0}\left[\|\theta_t - \theta'_t\| + \frac{|\Omega_r| - |\Omega_{\text{in}}|}{|\Omega_r|}2\alpha_t L\right]$$

$$\leq \mathbb{E}\left[\|\theta_t - \theta'_t\|\right] + 2\alpha_t L - \mathbb{I}\left(\frac{2^{(d+1)}|\Omega_r|}{|\Omega|}\right)\mathbb{E}_{\Omega_{\text{in}}>0}\left[\frac{|\Omega_{\text{in}}|}{|\Omega_r|}2\alpha_t L\right]$$

$$\leq \mathbb{E}\left[\|\theta_t - \theta'_t\|\right] + 2\alpha_t L(1 - \frac{|\Omega_r|}{|\Omega|}).$$

$$(31)$$

In the above inequalities, the third inequality is based on the following derivations. Firstly, denote $g = \left\|\left(\frac{1}{|\Omega_r|}\int_{\Omega_{\text{in}}}\nabla_\theta\mathcal{L}(u_{\theta_t}, \boldsymbol{x} + \boldsymbol{\xi})d\boldsymbol{\xi} - \frac{1}{|\Omega_r|}\int_{\Omega_{\text{in}}}\nabla_\theta\mathcal{L}(u_{\theta'_t}, \boldsymbol{x} + \boldsymbol{\xi})d\boldsymbol{\xi}\right)\right\|$. According to Assumption 3.2,

we have $g \leq \frac{|\Omega_{\text{in}}|}{|\Omega_r|}\beta\|\theta_t - \theta_t'\| \leq \beta\|\theta_t - \theta_t'\|$. Thus, the following inequality holds:

$$
\begin{aligned}
&\left\|\theta_t - \theta_t' - \left(\alpha_t \frac{1}{|\Omega_r|}\int_{\Omega_{\text{in}}}\nabla_\theta\mathcal{L}(u_{\theta_t}, \boldsymbol{x} + \boldsymbol{\xi})\mathrm{d}\boldsymbol{\xi} - \alpha_t \frac{1}{|\Omega_r|}\int_{\Omega_{\text{in}}}\nabla_\theta\mathcal{L}(u_{\theta_t'}, \boldsymbol{x} + \boldsymbol{\xi})\mathrm{d}\boldsymbol{\xi}\right)\right\|^2 \\
&= \|\theta_t - \theta_t'\|^2 - 2\alpha_t g^{\mathsf{T}}(\theta_t - \theta_t') + \alpha_t^2 g^2 \\
&\leq \|\theta_t - \theta_t'\|^2 - 2\frac{\alpha_t}{\beta}g^2 + \alpha_t^2 g^2 \qquad\qquad (\text{Convexity and } g \leq \beta\|\theta_t - \theta_t'\|) \\
&\leq \|\theta_t - \theta_t'\|^2. \qquad\qquad\qquad\qquad\qquad (\alpha_t \leq \frac{2}{\beta})
\end{aligned}
\tag{32}
$$

Thus, recursively accumulating the residual at the $t$-th step, we have:

$$
\mathcal{E}_{\text{gen}} \leq (1 - \frac{|\Omega_r|}{|\Omega|})\frac{2L^2}{|\mathcal{S}|}\sum_{t=1}^{T}\alpha_t.
\tag{33}
$$

For the **more general case**, we no longer assume that the domain $\Omega$ can be projected to a torus, resulting in a non-symmetric scenario when $|\Omega_{\text{in}}| > 0$. This asymmetry arises due to the presence of boundaries, since points that could potentially intersect with the set $\boldsymbol{x} + \Omega_r$ may be truncated by the boundary. Specifically, we consider $\Omega = [0, l]^{(d+1)}$ and $\Omega_r = [0, r]^{(d+1)}$. The concrete probability $\mathbb{P}(|\Omega_{\text{in}}| > 0)$ and the expectation $\mathbb{E}_{|\Omega_{\text{in}}|>0}(|\Omega_{\text{in}}|)$ can be calculated as follows:

$$
\mathbb{P}(|\Omega_{\text{in}}| > 0) = (\frac{r(2l - 3r)}{(l - r)^2})^{(d+1)}, \quad \mathbb{E}_{|\Omega_{\text{in}}|>0}(|\Omega_{\text{in}}|) = (\frac{r^2(3l - 4r)}{3(l - r)^2})^{(d+1)}.
\tag{34}
$$

Thus, the general case of Eq. 31 can be reformulated using the identities above. We assume $\frac{r}{l} < 0.5$, as when $\frac{r}{l} \geq 0.5$, it follows that $\mathbb{P}(|\Omega_{\text{in}}| > 0) = 1$. Specifically, Eq. (31) can be rewrite as:

$$
\begin{aligned}
&\mathbb{E}\left[\|G_{\alpha_t,\boldsymbol{x}}^{\text{region}}(\theta_t) - G_{\alpha_t,\boldsymbol{x}'}^{\text{region}}(\theta_t')\|\right] \\
&= \mathbb{P}(|\Omega_{\text{in}}| = 0)\mathbb{E}_{\Omega_{\text{in}}=0}\left[\|G_{\alpha_t,\boldsymbol{x}}^{\text{region}}(\theta_t) - G_{\alpha_t,\boldsymbol{x}'}^{\text{region}}(\theta_t')\|\right] + \mathbb{P}(|\Omega_{\text{in}}| > 0)\mathbb{E}_{\Omega_{\text{in}}>0}\left[\|G_{\alpha_t,\boldsymbol{x}}^{\text{region}}(\theta_t) - G_{\alpha_t,\boldsymbol{x}'}^{\text{region}}(\theta_t')\|\right] \\
&\leq \mathbb{P}(|\Omega_{\text{in}}| = 0)\mathbb{E}_{\Omega_{\text{in}}=0}[\|\theta_t - \theta_t'\| + 2\alpha_t L] \\
&\quad + \mathbb{P}(|\Omega_{\text{in}}| > 0)\mathbb{E}_{\Omega_{\text{in}}>0}\left[\left\|\theta_t - \theta_t' - \left(\alpha_t\frac{1}{|\Omega_r|}\int_{\Omega_r}\nabla_\theta\mathcal{L}(u_{\theta_t}, \boldsymbol{x} + \boldsymbol{\xi})\mathrm{d}\boldsymbol{\xi} - \alpha_t\frac{1}{|\Omega_r|}\int_{\Omega_r}\nabla_\theta\mathcal{L}(u_{\theta_t'}, \boldsymbol{x}' + \boldsymbol{\xi})\mathrm{d}\boldsymbol{\xi}\right)\right\|\right] \\
&\leq \mathbb{P}(|\Omega_{\text{in}}| = 0)\mathbb{E}_{\Omega_{\text{in}}=0}[\|\theta_t - \theta_t'\| + 2\alpha_t L] + \mathbb{P}(|\Omega_{\text{in}}| > 0)\mathbb{E}_{\Omega_{\text{in}}>0}\left[\frac{|\Omega_r| - |\Omega_{\text{in}}|}{|\Omega_r|}2\alpha_t L\right] \\
&\quad + \mathbb{P}(|\Omega_{\text{in}}| > 0)\mathbb{E}_{\Omega_{\text{in}}>0}\left[\left\|\theta_t - \theta_t' - \left(\alpha_t\frac{1}{|\Omega_r|}\int_{\Omega_{\text{in}}}\nabla_\theta\mathcal{L}(u_{\theta_t}, \boldsymbol{x} + \boldsymbol{\xi})\mathrm{d}\boldsymbol{\xi} - \alpha_t\frac{1}{|\Omega_r|}\int_{\Omega_{\text{in}}}\nabla_\theta\mathcal{L}(u_{\theta_t'}, \boldsymbol{x} + \boldsymbol{\xi})\mathrm{d}\boldsymbol{\xi}\right)\right\|\right] \\
&\leq \mathbb{P}(|\Omega_{\text{in}}| = 0)\mathbb{E}_{\Omega_r=0}[\|\theta_t - \theta_t'\| + 2\alpha_t L] + \mathbb{P}(|\Omega_{\text{in}}| > 0)\mathbb{E}_{\Omega_{\text{in}}>0}\left[\|\theta_t - \theta_t'\| + \frac{|\Omega_r| - |\Omega_{\text{in}}|}{|\Omega_r|}2\alpha_t L\right] \\
&\leq \mathbb{E}[\|\theta_t - \theta_t'\|] + 2\alpha_t L - \mathbb{P}(|\Omega_{\text{in}}| > 0)\mathbb{E}_{\Omega_{\text{in}}>0}\left[\frac{|\Omega_{\text{in}}|}{|\Omega_r|}2\alpha_t L\right] \\
&\leq \mathbb{E}[\|\theta_t - \theta_t'\|] + 2\alpha_t L\left[1 - \mathbb{P}(|\Omega_{\text{in}}| > 0)\frac{\mathbb{E}_{|\Omega_{\text{in}}|>0}(|\Omega_{\text{in}}|)}{|\Omega_r|}\right] \\
&= \mathbb{E}[\|\theta_t - \theta_t'\|] + 2\alpha_t L\left[1 - (\frac{r^2(2l - 3r)(3l - 4r)}{3(l - r)^4})^{(d+1)}\right].
\end{aligned}
\tag{35}
$$

Thus, recursively accumulating the residual at the $t$-th step, we have:

$$
\mathcal{E}_{\text{gen}} \leq \left[1 - (\frac{r^2(2l - 3r)(3l - 4r)}{3(l - r)^4})^{(d+1)}\right]\frac{2L^2}{|\mathcal{S}|}\sum_{t=1}^{T}\alpha_t.
\tag{36}
$$

Although the specific forms differ and the general case is much more complex, these two inequalities both share the same intuitive meaning: region optimization benefits from the overlap between $\boldsymbol{x} + \Omega_r$ and $\boldsymbol{x}' + \Omega_r$. Moreover, within a certain range, the benefit increases as the value of $r$ becomes larger.

Thus, to keep the bound simple and easy to understand, the main text theorems are under the assumption that $\Omega$ can be projected to a torus. Otherwise, $\frac{|\Omega_r|}{|\Omega|}$ should be replaced by $(\frac{r^2(2l-3r)(3l-4r)}{3(l-r)^4})(d+1)$.

$\square$

**Non-convex setting**  Next, we will prove the non-convex setting. Similarly, we assume $\Omega$ can be projected to a torus. Otherwise, the $\frac{|\Omega_r|}{|\Omega|}$ in the final bound should be replaced by $(\frac{r^2(2l-3r)(3l-4r)}{3(l-r)^4})(d+1)$.

*Proof.* For clarity, let $M = \frac{|\Omega_r|}{|\Omega|}$. Then, we can rewrite the Eq. (25) as follows.

If $\mathbb{E}(\delta_t) \leq \frac{2L}{\beta}$, we have:

$$
\begin{aligned}
\mathbb{E}\left[\|\theta_{t+1} - \theta'_{t+1}\| \mid \delta_{t_0} = 0\right] &\leq (1 - \frac{1}{|\mathcal{S}|})(1 + \frac{1}{t})\mathbb{E}\left[\|\theta_t - \theta'_t\|\right] + \frac{1}{|\mathcal{S}|}\mathbb{E}\left[\|G^{\text{region}}_{\alpha_t,\boldsymbol{x}}(\theta_t) - G^{\text{region}}_{\alpha_t,\boldsymbol{x}'}(\theta'_t)\|\right] \\
&\leq (1 - \frac{1}{|\mathcal{S}|})(1 + \frac{1}{t})\mathbb{E}\left[\|\theta_t - \theta'_t\|\right] + \frac{1}{|\mathcal{S}|}\left((1 + \frac{M}{t})\mathbb{E}\left[\|\theta_t - \theta'_t\|\right] + \frac{2L}{\beta t}(1 - M)\right) \\
&\leq (1 + \frac{1}{t} - \frac{1 - M}{t|\mathcal{S}|})\mathbb{E}[\delta_t] + \frac{2L}{\beta t|\mathcal{S}|}(1 - M) \\
&\leq \exp\left(\frac{1}{t} - \frac{1 - M}{t|\mathcal{S}|}\right)\mathbb{E}[\delta_t] + \frac{2L}{\beta t|\mathcal{S}|}(1 - M),
\end{aligned}
\tag{37}
$$

where the second inequality is based on the following derivations:

$$
\begin{aligned}
&\mathbb{E}\left[\|G^{\text{region}}_{\alpha_t,\boldsymbol{x}}(\theta_t) - G^{\text{region}}_{\alpha_t,\boldsymbol{x}'}(\theta'_t)\|\right] \\
&= \mathbb{I}\left(\frac{|\Omega| - 2^{(d+1)}|\Omega_r|}{|\Omega|}\right)\mathbb{E}_{\Omega_{\text{in}}=0}\left[\|G^{\text{region}}_{\alpha_t,\boldsymbol{x}}(\theta_t) - G^{\text{region}}_{\alpha_t,\boldsymbol{x}'}(\theta'_t)\|\right] \\
&\quad + \mathbb{I}\left(\frac{2^{(d+1)}|\Omega_r|}{|\Omega|}\right)\mathbb{E}_{\Omega_{\text{in}}>0}\left[\|G^{\text{region}}_{\alpha_t,\boldsymbol{x}}(\theta_t) - G^{\text{region}}_{\alpha_t,\boldsymbol{x}'}(\theta'_t)\|\right] \\
&\leq \mathbb{I}\left(\frac{|\Omega| - 2^{(d+1)}|\Omega_r|}{|\Omega|}\right)\mathbb{E}_{\Omega_{\text{in}}=0}\left[\|\theta_t - \theta'_t\| + 2\alpha_t L\right] \\
&\quad + \mathbb{I}\left(\frac{2^{(d+1)}|\Omega_r|}{|\Omega|}\right)\mathbb{E}_{\Omega_{\text{in}}>0}\left[\left\|\theta_t - \theta'_t - \left(\alpha_t\frac{1}{|\Omega_r|}\int_{\Omega_r}\nabla_\theta\mathcal{L}(u_{\theta_t}, \boldsymbol{x} + \boldsymbol{\xi})d\boldsymbol{\xi} - \alpha_t\frac{1}{|\Omega_r|}\int_{\Omega_r}\nabla_\theta\mathcal{L}(u_{\theta'_t}, \boldsymbol{x} + \boldsymbol{\xi})d\boldsymbol{\xi}\right)\right\|\right] \\
&\leq \mathbb{I}\left(\frac{|\Omega| - 2^{(d+1)}|\Omega_r|}{|\Omega|}\right)\mathbb{E}_{\Omega_{\text{in}}=0}\left[\|\theta_t - \theta'_t\| + 2\alpha_t L\right] \\
&\quad + \mathbb{I}\left(\frac{2^{(d+1)}|\Omega_r|}{|\Omega|}\right)\mathbb{E}_{\Omega_{\text{in}}>0}\left[\|\theta_t - \theta'_t\| + \left\|\left(\alpha_t\frac{1}{|\Omega_r|}\int_{\Omega_{\text{in}}}\nabla_\theta\mathcal{L}(u_{\theta_t}, \boldsymbol{x} + \boldsymbol{\xi})d\boldsymbol{\xi} - \alpha_t\frac{1}{|\Omega_r|}\int_{\Omega_{\text{in}}}\nabla_\theta\mathcal{L}(u_{\theta'_t}, \boldsymbol{x} + \boldsymbol{\xi})d\boldsymbol{\xi}\right)\right\|\right] \\
&\quad + \mathbb{I}\left(\frac{2^{(d+1)}|\Omega_r|}{|\Omega|}\right)\mathbb{E}_{\Omega_{\text{in}}>0}\left[\frac{|\Omega_r| - |\Omega_{\text{in}}|}{|\Omega_r|}2\alpha_t L\right] \\
&\leq \mathbb{I}\left(\frac{|\Omega| - 2^{(d+1)}|\Omega_r|}{|\Omega|}\right)\mathbb{E}_{\Omega_r=0}\left[\|\theta_t - \theta'_t\| + 2\alpha_t L\right] \\
&\quad + \mathbb{I}\left(\frac{2^{(d+1)}|\Omega_r|}{|\Omega|}\right)\mathbb{E}_{\Omega_{\text{in}}>0}\left[\|\theta_t - \theta'_t\| + \frac{|\Omega_{\text{in}}|}{|\Omega_r|}\alpha_t\beta\|\theta_t - \theta'_t\| + \frac{|\Omega_r| - |\Omega_{\text{in}}|}{|\Omega_r|}2\alpha_t L\right] \\
&\leq \mathbb{E}\left[\|\theta_t - \theta'_t\|\right] + \frac{\alpha_t\beta|\Omega_r|}{|\Omega|}\mathbb{E}\left[\|\theta_t - \theta'_t\|\right] + 2\alpha_t L - \mathbb{I}\left(\frac{2^{(d+1)}|\Omega_r|}{|\Omega|}\right)\mathbb{E}_{\Omega_{\text{in}}>0}\left[\frac{|\Omega_{\text{in}}|}{|\Omega_r|}2\alpha_t L\right] \\
&\leq (1 + \frac{\alpha_t\beta|\Omega_r|}{|\Omega|})\mathbb{E}\left[\|\theta_t - \theta'_t\|\right] + 2\alpha_t L(1 - \frac{|\Omega_r|}{|\Omega|}) \\
&= (1 + \frac{M}{t})\mathbb{E}\left[\|\theta_t - \theta'_t\|\right] + \frac{2L}{\beta t}(1 - M).
\end{aligned}
\tag{38}
$$

Notably, $\mathbb{E}\left[\|G^{\text{region}}_{\alpha_t,\boldsymbol{x}}(\theta_t) - G^{\text{region}}_{\alpha_t,\boldsymbol{x}'}(\theta'_t)\|\right]$ has an obvious upper bound, i.e. $\left(\mathbb{E}\left[\|\theta_t - \theta'_t\|\right] + \frac{2L}{\beta t}\right)$. And only when $\mathbb{E}(\delta_t) \leq \frac{2L}{\beta}$, the bound derived by Eq. (38) is tighter. Furthermore, the condition that $\mathbb{E}(\delta_t) \leq \frac{2L}{\beta}$ can be easily satisfied at the beginning several iterations since $\mathbb{E}(\delta_0) = 0$.

Otherwise, we still take the following equation:

$$\mathbb{E}\left[\|\theta_{t+1} - \theta'_{t+1}\| | \delta_{t_0} = 0\right] \leq \exp\left(\frac{1}{t} - \frac{1}{t|\mathcal{S}|}\right)\mathbb{E}[\delta_t] + \frac{2L}{\beta t|\mathcal{S}|}, \tag{39}$$

where we do not consider the benefits brought by the overlap area of $\boldsymbol{x} + \Omega_r$ and $\boldsymbol{x}' + \Omega_r$.

Suppose that at the first $K$ steps $\mathbb{E}[\delta_{t_0+K}] \leq \frac{2L}{\beta}$, Accumulating the above in equations recursively, we have the generalization error bound accumulated to the first $K$ steps as follows:

$$\begin{aligned}
\Delta &= \sum_{t=t_0+1}^{t_0+K}\left\{\Pi_{k=t+1}^{t_0+K}\exp\left(\frac{1}{t} - \frac{1-M}{t|\mathcal{S}|}\right)\Pi_{k=t_0+K+1}^{T}\exp\left(\frac{1}{t} - \frac{1}{t|\mathcal{S}|}\right)\right\}\frac{2L}{\beta t|\mathcal{S}|}(1-M) \\
&\leq \sum_{t=t_0+1}^{t_0+K}\exp\left((1-\frac{1}{|\mathcal{S}|})\log\frac{T}{t_0+K} + (1-\frac{1-M}{|\mathcal{S}|})\log\frac{t_0+K}{t}\right)\frac{2L}{\beta t|\mathcal{S}|}(1-M) \\
&= \sum_{t=t_0+1}^{t_0+K}\exp\left((1-\frac{1}{|\mathcal{S}|})\log\frac{T}{t} + \frac{M}{|\mathcal{S}|}\log\frac{t_0+K}{t}\right)\frac{2L}{\beta t|\mathcal{S}|}(1-M) \\
&\leq \sum_{t=t_0+1}^{t_0+K}\exp\left((1-\frac{1}{|\mathcal{S}|})\log\frac{T}{t}\right)\frac{2L}{\beta t|\mathcal{S}|}(1-M)(\frac{t_0+K}{t})^{\frac{M}{|\mathcal{S}|}} \\
&\leq \sum_{t=t_0+1}^{t_0+K}\exp\left((1-\frac{1}{|\mathcal{S}|})\log\frac{T}{t}\right)\frac{2L}{\beta t|\mathcal{S}|} - \sum_{t=t_0+1}^{t_0+K}\exp\left((1-\frac{1}{|\mathcal{S}|})\log\frac{T}{t}\right)\frac{2L}{\beta t|\mathcal{S}|}(M^2) \\
&= \sum_{t=t_0+1}^{t_0+K}\exp\left((1-\frac{1}{|\mathcal{S}|})\log\frac{T}{t}\right)\frac{2L}{\beta t|\mathcal{S}|} - JM^2,
\end{aligned} \tag{40}$$

where $J$ is a finite value that depends on the training property of beginning iterations, namely $K$ and $t_0$. The last inequality is from $(\frac{t_0+K}{t})^{\frac{M}{|\mathcal{S}|}} \leq (1+M)$, when $|\mathcal{S}|$ is sufficient enough.

Then, considering the all $T$ steps, we have

$$\begin{aligned}
\mathbb{E}\left[\|\theta_T - \theta'_T\| | \delta_{t_0} = 0\right] &\leq \Delta + \sum_{t=t_0+K+1}^{T}\left\{\Pi_{k=t+1}^{T}\exp\left(\frac{1}{t} - \frac{1}{t|\mathcal{S}|}\right)\right\}\frac{2L}{\beta t|\mathcal{S}|} \\
&\leq \sum_{t=t_0+1}^{T}\exp\left((1-\frac{1}{|\mathcal{S}|})\log\frac{T}{t}\right)\frac{2L}{\beta t|\mathcal{S}|} - JM^2 \qquad (\sum_{k=t+1}^{T}\frac{1}{k} \leq \log\frac{T}{t}) \\
&= \frac{2L}{\beta|\mathcal{S}|}T^{1-\frac{1}{|\mathcal{S}|}}\sum_{t=t_0+1}^{T}t^{-(1-\frac{1}{|\mathcal{S}|})-1} - JM^2 \\
&\leq \frac{2L}{\beta|\mathcal{S}|}T^{1-\frac{1}{|\mathcal{S}|}}\frac{1}{1-\frac{1}{|\mathcal{S}|}}\left(t_0^{-(1-\frac{1}{|\mathcal{S}|})} - T^{-(1-\frac{1}{|\mathcal{S}|})}\right) - JM^2. \quad \text{(Integral approximation)}
\end{aligned} \tag{41}$$

Thus, following a similar proof process as Theorem 3.3(2), we can obtain:

$$\mathbb{E}\left[\|\theta_T - \theta'_T\| | \delta_{t_0} = 0\right] \leq \frac{2L}{\beta(|\mathcal{S}|-1)}(\frac{T}{t_0}) - \frac{2L}{\beta(|\mathcal{S}|-1)} - JM^2. \tag{42}$$

With $t_0 = 1$, we have the generalization error under the non-convex case satisfies:

$$\mathbb{E}\left[|\mathcal{L}(u_{\theta_T}, \boldsymbol{x}) - \mathcal{L}(u_{\theta'_T}, \boldsymbol{x})|\right] \leq \frac{C}{|\mathcal{S}|} + \frac{2L^2(T-1)}{\beta(|\mathcal{S}|-1)} - JLM^2. \tag{43}$$

$\square$

## A.4 Proof for High-order Constraint Optimization (Corollary 3.6)

First, we would like to prove the following Lemma.

**Lemma A.4.** *Suppose that $\mathcal{L}$ is bounded by $C$ for all $\theta, \boldsymbol{x}$ and is $L$-Lipschitz and $\beta$-smooth for $\theta$, then the first-order $j$-th dimension loss function $\frac{\partial}{\partial x_j}\mathcal{L}_r^{\text{region}}$ is also bounded by $C$ for all $\theta, \boldsymbol{x}$ and is $2L$-Lipschitz and $2\beta$-smooth for $\theta$.*

*Proof.* For the bounded property, with the non-negative property of loss function, we have:

$$\frac{\partial}{\partial x_j}\mathcal{L}_r^{\text{region}}(u_\theta, \boldsymbol{x}) = \frac{\partial}{\partial x_j}\int_{\Omega_r}\mathcal{L}(u_\theta, \boldsymbol{x}+\boldsymbol{\xi})\mathrm{d}\boldsymbol{\xi} = \int_{\Omega_r\backslash x_j}\mathcal{L}(u_\theta, \boldsymbol{x}+\boldsymbol{\xi}_r) - \mathcal{L}(u_\theta, \boldsymbol{x}+\boldsymbol{\xi}_0)\mathrm{d}\boldsymbol{\xi} \leq C,$$

(44)

where $\boldsymbol{\xi}_r = (\cdots, r, \cdots) \in \Omega_t\backslash x_j$ and $\boldsymbol{\xi}_0 = (\cdots, 0, \cdots) \in \Omega_t\backslash x_j$.

As for the Lipschitz and smoothness, we can obtain the following inequalities:

$$\text{Lipschitz:} \quad \|\frac{\partial}{\partial x_j}\mathcal{L}_r^{\text{region}}(u_{\theta_1}, \boldsymbol{x}) - \frac{\partial}{\partial x_j}\mathcal{L}_r^{\text{region}}(u_{\theta_2}, \boldsymbol{x})\|$$

$$= \|\int_{\Omega_r\backslash x_j}\mathcal{L}(u_{\theta_1}, \boldsymbol{x}+\boldsymbol{\xi}_r) - \mathcal{L}(u_{\theta_1}, \boldsymbol{x}+\boldsymbol{\xi}_0) - \mathcal{L}(u_{\theta_2}, \boldsymbol{x}+\boldsymbol{\xi}_r) - \mathcal{L}(u_{\theta_2}, \boldsymbol{x}+\boldsymbol{\xi}_0)\mathrm{d}\boldsymbol{\xi}\|$$

$$\leq \int_{\Omega_r\backslash x_j}\|\mathcal{L}(u_{\theta_1}, \boldsymbol{x}+\boldsymbol{\xi}_r) - \mathcal{L}(u_{\theta_2}, \boldsymbol{x}+\boldsymbol{\xi}_r)\| + \|\mathcal{L}(u_{\theta_1}, \boldsymbol{x}+\boldsymbol{\xi}_0) - \mathcal{L}(u_{\theta_2}, \boldsymbol{x}+\boldsymbol{\xi}_0)\|\mathrm{d}\boldsymbol{\xi}$$

$$\leq 2L\|\theta_1 - \theta_2\|$$

$$\text{Smoothness:} \quad \|\nabla_\theta\frac{\partial}{\partial x_j}\mathcal{L}_r^{\text{region}}(u_{\theta_1}, \boldsymbol{x}) - \nabla_\theta\frac{\partial}{\partial x_j}\mathcal{L}_r^{\text{region}}(u_{\theta_2}, \boldsymbol{x})\|$$

$$= \|\nabla_\theta\int_{\Omega_r\backslash x_j}\mathcal{L}(u_{\theta_1}, \boldsymbol{x}+\boldsymbol{\xi}_r) - \mathcal{L}(u_{\theta_1}, \boldsymbol{x}+\boldsymbol{\xi}_0) - \mathcal{L}(u_{\theta_2}, \boldsymbol{x}+\boldsymbol{\xi}_r) - \mathcal{L}(u_{\theta_2}, \boldsymbol{x}+\boldsymbol{\xi}_0)\mathrm{d}\boldsymbol{\xi}\|$$

$$\leq \int_{\Omega_r\backslash x_j}\|\nabla_\theta\mathcal{L}(u_{\theta_1}, \boldsymbol{x}+\boldsymbol{\xi}_r) - \nabla_\theta\mathcal{L}(u_{\theta_2}, \boldsymbol{x}+\boldsymbol{\xi}_r)\|\mathrm{d}\boldsymbol{\xi}$$

$$+ \int_{\Omega_r\backslash x_j}\|\nabla_\theta\mathcal{L}(u_{\theta_1}, \boldsymbol{x}+\boldsymbol{\xi}_0) - \nabla_\theta\mathcal{L}(u_{\theta_2}, \boldsymbol{x}+\boldsymbol{\xi}_0)\|\mathrm{d}\boldsymbol{\xi}$$

$$\leq 2\beta\|\theta_1 - \theta_2\|.$$

(45)

Thus, $\frac{\partial}{\partial x_j}\mathcal{L}_r^{\text{region}}$ is also bounded by $C$ for all $\theta, \boldsymbol{x}$ and is $2L$-Lipschitz and $2\beta$-smooth for $\theta$. $\square$

Next, we will give the proof for Corollary 3.6.

*Proof.* According to Lemma A.3, we have the gradient update operator $G_{\alpha_t,\boldsymbol{x}}^{\text{region},x_j}$ for $\frac{\partial}{\partial x_j}\mathcal{L}_r^{\text{region}}$ satisfies the following inequality:

$$\|G_{\alpha_t,\boldsymbol{x}}^{\text{region},x_j}(\theta_1) - G_{\alpha_t,\boldsymbol{x}}^{\text{region},x_j}(\theta_2)\| \leq (1 + 2\alpha_t\beta)\|\theta_1 - \theta_2\|.$$

(46)

Let $M = \frac{|\Omega_r|}{|\Omega|}$, since $\alpha_t \leq \frac{1}{2\beta t}$, we can rewrite the Eq. (37) as follows:

$$\mathbb{E}\left[\|\theta_{t+1} - \theta'_{t+1}\| | \delta_{t_0} = 0\right]$$

$$\leq (1 - \frac{1}{|\mathcal{S}|})(1 + \frac{1}{t})\mathbb{E}\left[\|\theta_t - \theta'_t\|\right] + \frac{1}{|\mathcal{S}|}\left((1 + \frac{M}{t})\mathbb{E}\left[\|\theta_t - \theta'_t\|\right] + \frac{2L}{\beta}(1 - M)\right),$$

(47)

where the second term is derived from Eq. (38) by substituting $L$ to $2L$ and $\beta$ to $2\beta$, which is:

$$\mathbb{E}\left[\|G_{\alpha_t,\boldsymbol{x}}^{\text{region},x_j}(\theta_t) - G_{\alpha_t,\boldsymbol{x}'}^{\text{region},x_j}(\theta'_t)\|\right] \leq (1 + \frac{2\alpha_t\beta|\Omega_r|}{|\Omega|})\mathbb{E}\left[\|\theta_t - \theta'_t\|\right] + 4\alpha_tL(1 - \frac{|\Omega_r|}{|\Omega|})$$

$$= (1 + \frac{M}{t})\mathbb{E}\left[\|\theta_t - \theta'_t\|\right] + \frac{2L}{\beta t}(1 - M).$$

(48)

Thus, following the same derivation as Theorem 3.5, we have Corollary 3.6 holds. $\square$

# B  Algorithm Analysis in Section 3.2

This section contains the proof for the theoretical analysis of our proposed algorithm in Section 3.2.

## B.1  Proof for Convergence Rate of RoPINN (Theorem 3.8)

The crux of proof is to take expectation for Monte Carlo sampling.

*Proof.* From Taylor expansion, there exist $\boldsymbol{x}'$ such that:

$$
\begin{aligned}
\mathcal{L}_r^{\text{region}}(u_{\theta_{t+1}}, \boldsymbol{x}) &= \mathcal{L}_r^{\text{region}}(u_{\theta_t} - \alpha_t \nabla_\theta \mathcal{L}(u_{\theta_t}, \boldsymbol{x} + \boldsymbol{\xi}), \boldsymbol{x}) \\
&= \mathcal{L}_r^{\text{region}}(u_{\theta_t}, \boldsymbol{x}) - \alpha_t \nabla_\theta \mathcal{L}(u_{\theta_t}, \boldsymbol{x} + \boldsymbol{\xi})^\mathsf{T} \nabla_\theta \mathcal{L}_r^{\text{region}}(u_{\theta_t}, \boldsymbol{x}) \\
&\quad + \frac{1}{2} (\alpha_t \nabla_\theta \mathcal{L}(u_{\theta_t}, \boldsymbol{x}))^\mathsf{T} \nabla_\theta^2 \mathcal{L}_r^{\text{region}}(u_{\theta_t}, \boldsymbol{x}')(\alpha_t \nabla_\theta \mathcal{L}(u_{\theta_t}, \boldsymbol{x})) \\
&\leq \mathcal{L}_r^{\text{region}}(u_{\theta_t}, \boldsymbol{x}) - \alpha_t \nabla_\theta \mathcal{L}(u_{\theta_t}, \boldsymbol{x} + \boldsymbol{\xi})^\mathsf{T} \nabla_\theta \mathcal{L}_r^{\text{region}}(u_{\theta_t}, \boldsymbol{x}) + \frac{\alpha_t^2 L^2 H}{2}.
\end{aligned}
\tag{49}
$$

Taking expectations to $\boldsymbol{\xi}$ on both sides, since $\mathbb{E}[\nabla_\theta \mathcal{L}(u_{\theta_t}, \boldsymbol{x} + \boldsymbol{\xi})] = \nabla_\theta \mathcal{L}_r^{\text{region}}(u_{\theta_t}, \boldsymbol{x} + \boldsymbol{\xi})$, we have:

$$
\begin{aligned}
\mathbb{E}\left[\mathcal{L}_r^{\text{region}}(u_{\theta_{t+1}}, \boldsymbol{x})\right] &\leq \mathbb{E}\left[\mathcal{L}_r^{\text{region}}(u_{\theta_t}, \boldsymbol{x}) - \alpha_t \nabla_\theta \mathcal{L}(u_{\theta_t}, \boldsymbol{x} + \boldsymbol{\xi})^\mathsf{T} \nabla_\theta \mathcal{L}_r^{\text{region}}(u_{\theta_t}, \boldsymbol{x}) + \frac{\alpha_t^2 L^2 H}{2}\right] \\
&= \mathbb{E}\left[\mathcal{L}_r^{\text{region}}(u_{\theta_t}, \boldsymbol{x})\right] - \alpha_t \mathbb{E}\left[\left\|\nabla_\theta \mathcal{L}_r^{\text{region}}(u_{\theta_t}, \boldsymbol{x})\right\|^2\right] + \frac{\alpha_t^2 L^2 H}{2}.
\end{aligned}
\tag{50}
$$

Rearranging the terms and accumulating over $T$ iterations, we have the following sum:

$$
\begin{aligned}
\sum_{t=0}^{T-1} \alpha_t \mathbb{E}\left[\left\|\nabla_\theta \mathcal{L}_r^{\text{region}}(u_{\theta_t}, \boldsymbol{x})\right\|^2\right] &\leq \sum_{t=0}^{T-1} \left(\mathbb{E}\left[\mathcal{L}_r^{\text{region}}(u_{\theta_t}, \boldsymbol{x})\right] - \mathbb{E}\left[\mathcal{L}_r^{\text{region}}(u_{\theta_{t+1}}, \boldsymbol{x})\right]\right) + \sum_{t=0}^{T-1} \frac{\alpha_t^2 L^2 H}{2} \\
&\leq \mathcal{L}_r^{\text{region}}(u_{\theta_0}, \boldsymbol{x}) - \mathcal{L}_r^{\text{region}}(u_{\theta_T}, \boldsymbol{x}) + \frac{L^2 H}{2} \sum_{t=0}^{T-1} \alpha_t^2 \\
&\leq \mathcal{L}_r^{\text{region}}(u_{\theta_0}, \boldsymbol{x}) - \mathcal{L}_r^{\text{region}}(u_*, \boldsymbol{x}) + \frac{L^2 H}{2} \sum_{t=0}^{T-1} \alpha_t^2,
\end{aligned}
\tag{51}
$$

where $u_*$ represents the global optimum. Here we run the gradient descent for a random number of iterations $\tau$. For $\tau = t$ iterations with probability:

$$
\mathbb{P}(\tau = t) = \frac{\alpha_t}{\sum_{k=0}^{T-1} \alpha_k},
\tag{52}
$$

Thus, with $\alpha_t = \frac{1}{\sqrt{t+1}}$, we have the gradient norm is bounded by:

$$
\begin{aligned}
\mathbb{E}\left[\left\|\nabla_\theta \mathcal{L}_r^{\text{region}}(u_{\theta_\tau}, \boldsymbol{x})\right\|^2\right] &= \left(\sum_{t=0}^{T-1} \alpha_t\right)^{-1} \sum_{t=0}^{T-1} \alpha_t \mathbb{E}\left[\left\|\nabla_\theta \mathcal{L}_r^{\text{region}}(u_{\theta_t}, \boldsymbol{x})\right\|^2\right] \\
&\leq \left(\sum_{t=0}^{T-1} \alpha_t\right)^{-1} \left(\mathcal{L}_r^{\text{region}}(u_{\theta_0}, \boldsymbol{x}) - \mathcal{L}_r^{\text{region}}(u_*, \boldsymbol{x}) + \frac{L^2 H}{2} \sum_{t=0}^{T-1} \alpha_t^2\right) \\
&\lesssim (2\sqrt{T})^{-1} \left(\mathcal{L}_r^{\text{region}}(u_{\theta_0}, \boldsymbol{x}) - \mathcal{L}_r^{\text{region}}(u_*, \boldsymbol{x}) + \frac{L^2 H}{2} \log(T+1)\right) \\
&= \mathcal{O}(\frac{1}{\sqrt{T}}).
\end{aligned}
\tag{53}
$$

$\square$

## B.2 Proof for Estimation of RoPINN (Theorem 3.9)

As presented in Eq. (8) and (50), we approximate the region optimization with the Monte Carlo sampling method. For better efficiency, we propose only to sample one point at each iteration. However, this will cause an estimation error formalized in Theorem 3.9, which can be directly derived by the definition of standard deviation as follows:

*Proof.* According to the definition of $\mathcal{L}_r^{\text{region}}$ in Eq. (5), we get

$$
\begin{aligned}
&\mathbb{E}_{\boldsymbol{\xi} \sim U(\Omega_r)} \left[ \left\| \nabla_\theta \mathcal{L}(u_\theta, \boldsymbol{x} + \boldsymbol{\xi}) - \nabla_\theta \mathcal{L}_r^{\text{region}}(u_\theta, \boldsymbol{x}) \right\|^2 \right]^{\frac{1}{2}} \\
&= \mathbb{E}_{\boldsymbol{\xi} \sim U(\Omega_r)} \left[ \left\| \nabla_\theta \mathcal{L}(u_\theta, \boldsymbol{x} + \boldsymbol{\xi}) - \nabla_\theta \frac{1}{|\Omega_r|} \int_{\Omega_r} \mathcal{L}(u_\theta, \boldsymbol{x} + \boldsymbol{\xi}) \mathrm{d}\boldsymbol{\xi} \right\|^2 \right]^{\frac{1}{2}} \\
&= \left\| \sigma_{\boldsymbol{\xi} \sim U(\Omega_r)} \left( \nabla_\theta \mathcal{L}(u_\theta, \boldsymbol{x} + \boldsymbol{\xi}) \right) \right\|.
\end{aligned}
\tag{54}
$$

$\square$

## B.3 Proof for Estimation of Trust Region (Lemma 3.10 and Theorem 3.11)

First, we give the proof for Lemma 3.10.

*Proof.* According to Assumption 3.2, there exist $\boldsymbol{x}'$, such that the following equation holds:

$$
\begin{aligned}
&\nabla_\theta \mathcal{L}(u_{\theta_t}, \boldsymbol{z}_1) - \nabla_\theta \mathcal{L}(u_{\theta_{t-1}}, \boldsymbol{z}_2) \\
&= \nabla_\theta \mathcal{L}(u_{\theta_t}, \boldsymbol{z}_1) - \nabla_\theta \mathcal{L}(u_{\theta_t + \alpha_{t-1} \nabla_\theta \mathcal{L}(u_{\theta_{t-1}}, \boldsymbol{z}_2)}, \boldsymbol{z}_2) \\
&= \nabla_\theta \mathcal{L}(u_{\theta_t}, \boldsymbol{z}_1) - \nabla_\theta \mathcal{L}(u_{\theta_t}, \boldsymbol{z}_2) + \alpha_{t-1} \nabla_\theta \mathcal{L}(u_{\theta_{t-1}}, \boldsymbol{z}_2) \nabla_\theta^2 \mathcal{L}(u_{\theta_{t-1}}, \boldsymbol{x}').
\end{aligned}
\tag{55}
$$

Thus, the following inequality holds:

$$
\begin{aligned}
&\left| \left\| \nabla_\theta \mathcal{L}(u_{\theta_t}, \boldsymbol{z}_1) - \nabla_\theta \mathcal{L}(u_{\theta_{t-1}}, \boldsymbol{z}_2) \right\| - \left\| \nabla_\theta \mathcal{L}(u_{\theta_t}, \boldsymbol{z}_1) - \nabla_\theta \mathcal{L}(u_{\theta_t}, \boldsymbol{z}_2) \right\| \right| \\
&\leq \left\| \left( \nabla_\theta \mathcal{L}(u_{\theta_t}, \boldsymbol{z}_1) - \nabla_\theta \mathcal{L}(u_{\theta_{t-1}}, \boldsymbol{z}_2) \right) - \left( \nabla_\theta \mathcal{L}(u_{\theta_t}, \boldsymbol{z}_1) - \nabla_\theta \mathcal{L}(u_{\theta_t}, \boldsymbol{z}_2) \right) \right\| \\
&= \left\| \alpha_{t-1} \nabla_\theta \mathcal{L}(u_{\theta_{t-1}}, \boldsymbol{z}_2) \nabla_\theta^2 \mathcal{L}(u_{\theta_{t-1}}, \boldsymbol{x}') \right\| \\
&\leq \beta L \alpha_{t-1}.
\end{aligned}
\tag{56}
$$

$\square$

Next, we will prove Theorem 3.11.

*Proof.* This theorem can be proved by demonstrating that: for all $i, j \in \{1, \cdots, T_0\}$:

$$
\begin{aligned}
\lim_{t \to \infty} \nabla_\theta \mathcal{L}(u_{\theta_{t-i+1}}, \boldsymbol{z}_i) &= \nabla_\theta \mathcal{L}(u_{\theta_t}, \boldsymbol{z}_i) \\
\lim_{t \to \infty} \left( \nabla_\theta \mathcal{L}(u_{\theta_{t-i+1}}, \boldsymbol{z}_i) - \nabla_\theta \mathcal{L}(u_{\theta_{t-j+1}}, \boldsymbol{z}_j) \right) &= \nabla_\theta \mathcal{L}(u_{\theta_t}, \boldsymbol{z}_i) - \nabla_\theta \mathcal{L}(u_{\theta_t}, \boldsymbol{z}_j).
\end{aligned}
\tag{57}
$$

For the first equation, since $\alpha_t \to 0$, given $\forall \epsilon$, there exists a constant $M$ such that any $t > M$, $\alpha_t \leq \frac{\epsilon}{T_0 L \beta}$. Thus, for any $t > M$ and any $i, j \in \{1, \cdots, T_0\}$, the following equation is satisfied:

$$
\begin{aligned}
\left\| \nabla_\theta \mathcal{L}(u_{\theta_{t-i+1}}, \boldsymbol{z}_i) - \nabla_\theta \mathcal{L}(u_{\theta_t}, \boldsymbol{z}_i) \right\| &\leq \sum_{k=1}^{i-1} \left\| \nabla_\theta \mathcal{L}(u_{\theta_{t-i+k}}, \boldsymbol{z}_i) - \nabla_\theta \mathcal{L}(u_{\theta_{t-i+k+1}}, \boldsymbol{z}_i) \right\| \\
&\leq \sum_{k=1}^{i-1} \alpha_{t-i+k} L \beta \leq \epsilon.
\end{aligned}
\tag{58}
$$

Thus, $\lim_{t \to \infty} \nabla_\theta \mathcal{L}(u_{\theta_{t-i+1}}, \boldsymbol{z}_i) = \nabla_\theta \mathcal{L}(u_{\theta_t}, \boldsymbol{z}_i)$. Therefore, given $\forall \epsilon'$, there exist a constant $M'$, $\forall t > M'$, $\left\| \nabla_\theta \mathcal{L}(u_{\theta_{t-i+1}}, \boldsymbol{z}_i) - \nabla_\theta \mathcal{L}(u_{\theta_t}, \boldsymbol{z}_i) \right\| \leq \frac{\epsilon}{2}$.

As for the second equation, for any $t > M'$, the following equation is satisfied:

$$
\begin{aligned}
&\left\| \nabla_\theta \mathcal{L}(u_{\theta_{t-i+1}}, \boldsymbol{z}_i) - \nabla_\theta \mathcal{L}(u_{\theta_{t-j+1}}, \boldsymbol{z}_j) - \nabla_\theta \mathcal{L}(u_{\theta_t}, \boldsymbol{z}_i) - \nabla_\theta \mathcal{L}(u_{\theta_t}, \boldsymbol{z}_j) \right\| \\
&\leq \left\| \nabla_\theta \mathcal{L}(u_{\theta_{t-i+1}}, \boldsymbol{z}_i) - \nabla_\theta \mathcal{L}(u_{\theta_t}, \boldsymbol{z}_i) \right\| + \left\| \nabla_\theta \mathcal{L}(u_{\theta_{t-j+1}}, \boldsymbol{z}_j) - \nabla_\theta \mathcal{L}(u_{\theta_t}, \boldsymbol{z}_j) \right\| \leq \epsilon'.
\end{aligned}
\tag{59}
$$

Thus, Theorem 3.11 can be proved by replacing the gradient of past iterations with their limitations.

$\square$

## B.4 Proof for Region Optimization with Gradient Estimation Error (Theorem 3.12)

**Convex setting** Firstly, we would like to prove the convex case as follows.

*Proof.* Similar to the proof of region optimization in Appendix A.3, at the $t$-th step, we can obtain the following equation:

$$
\begin{aligned}
\mathbb{E}\left[\|\theta_{t+1} - \theta'_{t+1}\|\right] &= (1 - \frac{1}{|\mathcal{S}|})\mathbb{E}\left[\|G^{\mathrm{approx}}_{\alpha_t, \boldsymbol{x}}(\theta_t) - G^{\mathrm{approx}}_{\alpha_t, \boldsymbol{x}}(\theta'_t)\|\right] + \frac{1}{|\mathcal{S}|}\mathbb{E}\left[\|G^{\mathrm{approx}}_{\alpha_t, \boldsymbol{x}}(\theta_t) - G^{\mathrm{approx}}_{\alpha_t, \boldsymbol{x}'}(\theta'_t)\|\right] \\
&\leq (1 - \frac{1}{|\mathcal{S}|})\mathbb{E}\left[\|\theta_t - \theta'_t\|\right] + \frac{1}{|\mathcal{S}|}\mathbb{E}\left[\|G^{\mathrm{approx}}_{\alpha_t, \boldsymbol{x}}(\theta_t) - G^{\mathrm{approx}}_{\alpha_t, \boldsymbol{x}'}(\theta'_t)\|\right],
\end{aligned}
\tag{60}
$$

where $G^{\mathrm{approx}}_{\alpha, \boldsymbol{x}}(\theta) = \theta - \alpha\nabla_\theta\mathcal{L}^{\mathrm{approx}}_r(\theta, \boldsymbol{x}) = \theta - \alpha\nabla_\theta\mathcal{L}(\theta, \boldsymbol{x} + \boldsymbol{\xi}), \boldsymbol{\xi} \sim U(\Omega_r)$. Suppose that we have sampled $\boldsymbol{\xi}, \boldsymbol{\xi}' \in \Omega_r$, the second term on the right part can be bounded as follows:

$$
\begin{aligned}
&\mathbb{E}\left[\|G^{\mathrm{approx}}_{\alpha_t, \boldsymbol{x}}(\theta_t) - G^{\mathrm{approx}}_{\alpha_t, \boldsymbol{x}'}(\theta'_t)\|\right] \\
&\leq \mathbb{E}\left[\|\theta_t - \alpha_t\nabla_\theta\mathcal{L}(\theta_t, \boldsymbol{x} + \boldsymbol{\xi}) - \theta'_t - \alpha_t\nabla_\theta\mathcal{L}(\theta'_t, \boldsymbol{x}' + \boldsymbol{\xi}')\|\right] \\
&\leq \mathbb{E}\left[\|\theta_t - \alpha_t\nabla_\theta\mathcal{L}^{\mathrm{region}}_r(\theta_t, \boldsymbol{x}) - \theta'_t - \alpha_t\nabla_\theta\mathcal{L}^{\mathrm{region}}_r(\theta'_t, \boldsymbol{x}')\|\right] \\
&\quad + \mathbb{E}\left[\|\alpha_t\nabla_\theta\mathcal{L}(\theta_t, \boldsymbol{x} + \boldsymbol{\xi}) - \alpha_t\nabla_\theta\mathcal{L}^{\mathrm{region}}_r(\theta_t, \boldsymbol{x})\|\right] \\
&\quad + \mathbb{E}\left[\|\alpha_t\nabla_\theta\mathcal{L}(\theta_t, \boldsymbol{x}' + \boldsymbol{\xi}') - \alpha_t\nabla_\theta\mathcal{L}^{\mathrm{region}}_r(\theta'_t, \boldsymbol{x}')\|\right] \\
&\leq \mathbb{E}\left[\|\theta_t - \theta'_t\|\right] + 2\alpha_t L(1 - \frac{|\Omega_r|}{|\Omega|}) + 2\alpha_t\mathcal{E}_{r,\mathrm{grad}} \qquad \text{(Based on Eq. (31))} \\
&= \mathbb{E}\left[\|\theta_t - \theta'_t\|\right] + 2\alpha_t\left(L(1 - \frac{|\Omega_r|}{|\Omega|}) + \mathcal{E}_{r,\mathrm{grad}}\right)
\end{aligned}
\tag{61}
$$

Thus, recursively accumulating the residual at the $t$-th step, we have:

$$
\mathcal{E}_{\mathrm{gen}} \leq \left(L(1 - \frac{|\Omega_r|}{|\Omega|}) + \mathcal{E}_{r,\mathrm{grad}}\right)\frac{2L}{|\mathcal{S}|}\sum_{t=1}^T\alpha_t.
\tag{62}
$$

$\square$

**Non-convex setting** Similarly, we can prove the non-convex setting as follows.

*Proof.* It is easy to prove that $\mathcal{L}^{\mathrm{approx}}_r(\theta, \boldsymbol{x})$ is still $L$-Lipchitz-$\beta$-smoothness for $\theta$. For clarity, we define that $M = \frac{|\Omega_r|}{|\Omega|}$. Thus, based on Eq. (38), we have the following derivations.

If $\mathbb{E}(\delta_t) \leq \frac{2L}{\beta} - \frac{2}{\beta M}\mathcal{E}_{r,\mathrm{grad}}$, we have:

$$
\begin{aligned}
&\mathbb{E}\left[\|\theta_{t+1} - \theta'_{t+1}\| | \delta_{t_0} = 0\right] \\
&\leq (1 - \frac{1}{|\mathcal{S}|})(1 + \frac{1}{t})\mathbb{E}\left[\|\theta_t - \theta'_t\|\right] + \frac{1}{|\mathcal{S}|}\mathbb{E}\left[\|G^{\mathrm{approx}}_{\alpha_t, \boldsymbol{x}}(\theta_t) - G^{\mathrm{approx}}_{\alpha_t, \boldsymbol{x}'}(\theta'_t)\|\right] \\
&\leq (1 - \frac{1}{|\mathcal{S}|})(1 + \frac{1}{t})\mathbb{E}\left[\|\theta_t - \theta'_t\|\right] \\
&\quad + \frac{1}{|\mathcal{S}|}\left(\mathbb{E}\left[\|G^{\mathrm{region}}_{\alpha_t, \boldsymbol{x}}(\theta_t) - G^{\mathrm{region}}_{\alpha_t, \boldsymbol{x}'}(\theta'_t)\|\right]\right) \\
&\quad + \frac{1}{|\mathcal{S}|}\left(\mathbb{E}\left[\|\alpha_t\nabla_\theta\mathcal{L}(\theta_t, \boldsymbol{x} + \boldsymbol{\xi}) - \alpha_t\nabla_\theta\mathcal{L}^{\mathrm{region}}_r(\theta_t, \boldsymbol{x})\|\right]\right) \\
&\quad + \frac{1}{|\mathcal{S}|}\left(\mathbb{E}\left[\|\alpha_t\nabla_\theta\mathcal{L}(\theta_t, \boldsymbol{x}' + \boldsymbol{\xi}') - \alpha_t\nabla_\theta\mathcal{L}^{\mathrm{region}}_r(\theta'_t, \boldsymbol{x}')\|\right]\right) \\
&\leq (1 + \frac{1}{t} - \frac{1 - M}{t|\mathcal{S}|})\mathbb{E}[\delta_t] + \frac{2\alpha_t}{|\mathcal{S}|}(L(1 - M) + \mathcal{E}_{r,\mathrm{grad}}) \\
&\leq \exp\left(\frac{1}{t} - \frac{1 - M}{t|\mathcal{S}|}\right)\mathbb{E}[\delta_t] + \frac{2}{\beta t|\mathcal{S}|}(L(1 - M) + \mathcal{E}_{r,\mathrm{grad}}).
\end{aligned}
\tag{63}
$$

Otherwise, we still consider the following inequality:

$$\mathbb{E}\left[\|\theta_{t+1} - \theta'_{t+1}\| | \delta_{t_0} = 0\right] \leq \exp\left(\frac{1}{t} - \frac{1}{t|\mathcal{S}|}\right) \mathbb{E}[\delta_t] + \frac{2L}{\beta t|\mathcal{S}|}, \tag{64}$$

Suppose that at the first $K'$ steps $\mathbb{E}[\delta_{t_0+K'}] \leq \frac{2L}{\beta} - \frac{2}{\beta M}\mathcal{E}_{r,\text{grad}}$.

Accumulating the above in equations recursively, we have the generalization error bound accumulated to the first $K'$ steps as follows:

$$
\begin{aligned}
\Delta &= \sum_{t=t_0+1}^{t_0+K'} \left\{ \Pi_{k=t+1}^{t_0+K'}\exp\left(\frac{1}{t} - \frac{1-M}{t|\mathcal{S}|}\right) \Pi_{k=t_0+K'+1}^{T}\exp\left(\frac{1}{t} - \frac{1}{t|\mathcal{S}|}\right) \right\} \frac{2}{\beta t|\mathcal{S}|}\left(L(1-M) + \mathcal{E}_{r,\text{grad}}\right) \\
&\leq \sum_{t=t_0+1}^{t_0+K'} \exp\left((1 - \frac{1}{|\mathcal{S}|})\log\frac{T}{t_0+K'} + (1 - \frac{1-M}{|\mathcal{S}|})\log\frac{t_0+K'}{t}\right) \frac{2}{\beta t|\mathcal{S}|}\left(L(1-M) + \mathcal{E}_{r,\text{grad}}\right) \\
&= \sum_{t=t_0+1}^{t_0+K'} \exp\left((1 - \frac{1}{|\mathcal{S}|})\log\frac{T}{t} + \frac{M}{|\mathcal{S}|}\log\frac{t_0+K'}{t}\right) \frac{2}{\beta t|\mathcal{S}|}\left(L(1-M) + \mathcal{E}_{r,\text{grad}}\right) \\
&\leq \sum_{t=t_0+1}^{t_0+K'} \exp\left((1 - \frac{1}{|\mathcal{S}|})\log\frac{T}{t}\right) \frac{2}{\beta t|\mathcal{S}|}\left(L(1-M) + \mathcal{E}_{r,\text{grad}}\right)(\frac{t_0+K'}{t})^{\frac{M}{|\mathcal{S}|}} \\
&\leq \sum_{t=t_0+1}^{t_0+K'} \exp\left((1 - \frac{1}{|\mathcal{S}|})\log\frac{T}{t}\right) \frac{2L}{\beta t|\mathcal{S}|} - \sum_{t=t_0+1}^{t_0+K'} \exp\left((1 - \frac{1}{|\mathcal{S}|})\log\frac{T}{t}\right) \frac{2}{\beta t|\mathcal{S}|}\left(LM^2 + \mathcal{E}_{r,\text{grad}}(1+M)\right) \\
&= \sum_{t=t_0+1}^{t_0+K'} \exp\left((1 - \frac{1}{|\mathcal{S}|})\log\frac{T}{t}\right) \frac{2L}{\beta t|\mathcal{S}|} - J'LM^2 + J'\mathcal{E}_{r,\text{grad}}(1+M),
\end{aligned}
\tag{65}
$$

where $J'$ is a finite value that depends on the training property of beginning iterations, namely $K'$ and $t_0$. The last inequality is from $(\frac{t_0+K'}{t})^{\frac{M}{|\mathcal{S}|}} \leq (1+M)$, when $|\mathcal{S}|$ is sufficient enough.

Then, considering the all $T$ steps, we have

$$
\begin{aligned}
&\mathbb{E}\left[\|\theta_T - \theta'_T\| | \delta_{t_0} = 0\right] \\
&\leq \Delta + \sum_{t=t_0+K+1}^{T} \left\{ \Pi_{k=t+1}^{T}\exp\left(\frac{1}{t} - \frac{1}{t|\mathcal{S}|}\right) \right\} \frac{2L}{\beta t|\mathcal{S}|} \\
&\leq \sum_{t=t_0+1}^{T} \exp\left((1 - \frac{1}{|\mathcal{S}|})\log\frac{T}{t}\right) \frac{2L}{\beta t|\mathcal{S}|} - J'M^2 + J'\mathcal{E}_{r,\text{grad}}(1+M) \qquad (\sum_{k=t+1}^{T}\frac{1}{k} \leq \log\frac{T}{t}) \\
&= \frac{2L}{\beta|\mathcal{S}|}T^{1-\frac{1}{|\mathcal{S}|}} \sum_{t=t_0+1}^{T} t^{-(1-\frac{1}{|\mathcal{S}|})-1} - J'M^2 + J'\mathcal{E}_{r,\text{grad}}(1+M) \\
&\leq \frac{2L}{\beta|\mathcal{S}|}T^{1-\frac{1}{|\mathcal{S}|}}\frac{1}{1-\frac{1}{|\mathcal{S}|}}\left(t_0^{-(1-\frac{1}{|\mathcal{S}|})} - T^{-(1-\frac{1}{|\mathcal{S}|})}\right) - J'M^2 + J'\mathcal{E}_{r,\text{grad}}(1+M). \quad \text{(Integral approximation)}
\end{aligned}
\tag{66}
$$

Next, following the proof in Appendix A.3, we can obtain the generalization bound as follows:

$$\mathbb{E}\left[|\mathcal{L}(u_{\theta_T}, \boldsymbol{x}) - \mathcal{L}(u_{\theta'_T}, \boldsymbol{x})|\right] \leq \frac{C}{|\mathcal{S}|} + \frac{2L^2(T-1)}{\beta(|\mathcal{S}|-1)} - J'LM^2 + J'\mathcal{E}_{r,\text{grad}}(1+M). \tag{67}$$

Note that $K'$ does not exist when $\frac{2L}{\beta} < \frac{2}{\beta M}\mathcal{E}_{r,\text{grad}}$, then $J' = 0$, which corresponds to the situation that the region size is too large and brings serious gradient estimation error. Introducing "region" cannot bring a better generalization bound in this case. $\qquad \square$

## C Implementation Details

This section provides experiment details, including **benchmarks**, **metrics** and **implementations**.

### C.1 Benchmarks

To comprehensively test our algorithm, we include the following four benchmarks. The first three benchmarks cover three typical PDEs (plotted in Figure 7), which are widely used in exploring the PINN optimization [24, 37]. The last one is an advanced comprehensive benchmark with 20 different PDEs. Here are the details.

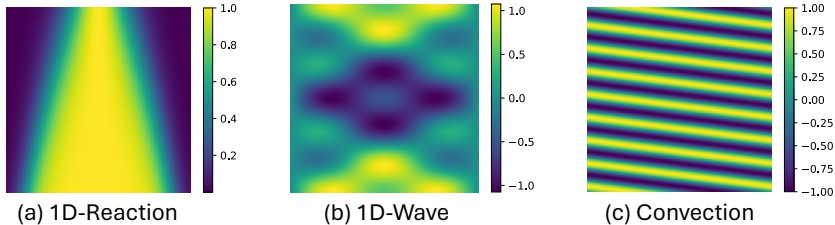

(a) 1D-Reaction      (b) 1D-Wave      (c) Convection

Figure 7: Visualization of the solution $u$ for the first three benchmarks.

**1D-Reaction** This problem is a one-dimensional non-linear ODE, which describes the chemical reactions. The concrete equation that we studied here can be formalized as follows:

$$\frac{\partial u}{\partial t} - \rho u(1 - u) = 0, \, x \in (0, 2\pi), t \in (0, 1),$$
$$u(x, 0) = \exp\left(-\frac{(x - \pi)^2}{2(\pi/4)^2}\right), \, x \in [0, 2\pi], \tag{68}$$
$$u(0, t) = u(2\pi, t), \, t \in [0, 1].$$

The analytical solution to this problem is $u(x, t) = \frac{h(x)e^{\rho t}}{h(x)e^{\rho t} + 1 - h(x)}$ and $h(x) = \exp\left(-\frac{(x-\pi)}{2(\pi/4)^2}\right)$. In our experiments, we set $\rho = 5$. This problem is previously studied as "PINN failure mode" [24], which is because of the non-linear term of the equation [29]. Besides, as shown in Figure 7(a), it contains sharp boundaries for the center high-value area, which is also hard to learn for deep models.

Following experiments in PINNsFormer [58], we uniformly sampled 101 points for initial state $\Omega_0$ and boundary $\partial\Omega$ and a uniform grid of 101×101 mesh points for the residual domain $\Omega$. For evaluation, we employed a 101×101 mesh within the residual domain $\Omega$. This strategy is also adopted for 1D-Wave and Convection experiments.

**1D-Wave** This problem presents a hyperbolic PDE that is widely studied in acoustics, electromagnetism, and fluid dynamics [1]. Concretely, the PDE can be formalized as follows:

$$\frac{\partial^2 u}{\partial t^2} - 4\frac{\partial^2 u}{\partial x^2} = 0, \, x \in (0, 1), t \in (0, 1),$$
$$u(x, 0) = \sin(\pi x) + \frac{1}{2}\sin(\beta \pi x), \, x \in [0, 1], \tag{69}$$
$$\frac{\partial u(x, 0)}{\partial t} = 0, \, x \in [0, 1],$$
$$u(0, t) = u(1, t) = 0, \, t \in [0, 1].$$

The analytic solution for this PDE is $u(x, t) = \sin(\pi x)\cos(2\pi t) + \frac{1}{2}\sin(\beta\pi x)\cos(2\beta\pi t)$. We set $\beta = 3$ for our experiments. As presented in Figure 7(b), the solution is smoother than the other two datasets, thereby easier for deep models to solve in some aspects. However, the equation contains second-order derivative terms, which also brings challenges in automatic differentiation. That is why gPINN [55] fails in this task (Table 2).

Table 4: Details of datasets in PINNacle [12] (16 different PDEs included in our experiments), including the dimension of inputs, highest order of PDEs, number of train/test points and concrete equations. Here we only present the simplified PDE formalizations for intuitive understanding. More detailed descriptions of PDE type and coefficient meanings can be found in their paper [12].

| PDE | | Dimension | Order | $N_{\text{train}}$ | $N_{\text{test}}$ | Key Equations |
|---|---|---|---|---|---|---|
| Burges | 1d-C | 1D+Time | 2 | 16384 | 12288 | $\frac{\partial \boldsymbol{u}}{\partial t} + \boldsymbol{u} \cdot \nabla \boldsymbol{u} - \nu \Delta \boldsymbol{u} = 0$ |
| | 2d-C | 2D+Time | 2 | 98308 | 82690 | |
| Poisson | 2d-C | 2D | 2 | 12288 | 10240 | $-\Delta \boldsymbol{u} = 0$ |
| | 2d-CG | 2D | 2 | 12288 | 10240 | $-\Delta \boldsymbol{u} + k^2 \boldsymbol{u} = f(x,y)$ |
| | 3d-CG | 3D | 2 | 49152 | 40960 | $-\mu_i \Delta \boldsymbol{u} + k_i^2 \boldsymbol{u} = f(x,y,z), i = 1,2$ |
| | 2d-MS | 2D | 2 | 12288 | 10329 | $-\nabla(a(x)\nabla \boldsymbol{u}) = f(x,y)$ |
| Heat | 2d-VC | 2D+Time | 2 | 65536 | 49189 | $\frac{\partial \boldsymbol{u}}{\partial t} - \nabla(a(x)\nabla \boldsymbol{u}) = f(x,t)$ |
| | 2d-MS | 2D+Time | 2 | 65536 | 49189 | $\frac{\partial \boldsymbol{u}}{\partial t} - \frac{1}{(500\pi)^2}\boldsymbol{u}_{xx} - \frac{1}{\pi^2}\boldsymbol{u}_{yy} = 0$ |
| | 2d-CG | 2D+Time | 2 | 65536 | 49152 | $\frac{\partial \boldsymbol{u}}{\partial t} - \Delta \boldsymbol{u} = 0$ |
| NS | 2d-C | 2D | 2 | 14337 | 12378 | $\boldsymbol{u} \cdot \nabla \boldsymbol{u} + \nabla p - \frac{1}{Re}\Delta \boldsymbol{u} = 0, \nabla \cdot \boldsymbol{u} = 0$ |
| | 2d-CG | 2D | 2 | 14055 | 12007 | |
| Wave | 1d-C | 1D+Time | 2 | 12288 | 10329 | $\boldsymbol{u}_{tt} - 4\boldsymbol{u}_{xx} = 0$ |
| | 2d-CG | 2D+Time | 2 | 49170 | 42194 | $\left[\nabla^2 - \frac{1}{c(x)}\frac{\partial^2}{\partial t^2}\right] u(x,t) = 0$ |
| Chaotic | GS | 2D+Time | 2 | 65536 | 61780 | $\boldsymbol{u}_t = \varepsilon_1 \Delta \boldsymbol{u} + b(1 - \boldsymbol{u}) - \boldsymbol{u}\boldsymbol{v}^2$ $\boldsymbol{v}_t = \varepsilon_2 \Delta \boldsymbol{v} - d\boldsymbol{v} + \boldsymbol{u}\boldsymbol{v}^2$ |
| High-dim | PNd | 5D | 2 | 49152 | 67241 | $-\Delta \boldsymbol{u} = \frac{\pi^2}{4}\sum_{i=1}^{n}\sin\left(\frac{\pi}{2}x_i\right)$ |
| | HNd | 5D+Time | 2 | 65537 | 49152 | $\frac{\partial \boldsymbol{u}}{\partial t} = k\Delta \boldsymbol{u} + f(x,t)$ |

**Convection**    This problem is also a hyperbolic PDE that can be used to model fluid, atmosphere, heat transfer and biological processes [41]. The concrete PDE that we studied in this paper is:

$$
\begin{aligned}
&\frac{\partial u}{\partial t} + \beta \frac{\partial u}{\partial t} = 0, x \in (0, 2\pi), t \in (0, 1), \\
&u(x,0) = \sin(x), x \in [0, 2\pi], \\
&u(0,t) = u(2\pi, t), t \in [0, 1].
\end{aligned}
\tag{70}
$$

The analytic solution for this PDE is $u(x,t) = \sin(x - \beta t)$, where $\beta$ is set as 50 in our experiments. Note that although the final solution seems to be quite simple, it is difficult for PINNs in practice due to the highly complex and high-frequency patterns. And the previous research [24] has shown that the loss landscape of the Convection equation contains many hard-to-optimize sharp cones.

**PINNacle**    This benchmark [12] is built upon the DeepXDE [30], consisting of a wide range of PDEs and baselines. In their paper, the authors included 20 different PDE-solving tasks, covering diverse phenomena in fluid dynamics, heat conduction, etc and including PDEs with high dimensions, complex geometrics, nonlinearity and multiscale interactions. To ensure a comprehensive evaluation, we also benchmark RoPINN with PINNacle.

During our experiments, we found that there are several subtasks that none of the previous methods can solve, such as the 2D Heat equation with long time (Heat 2d-LT), 2D Navier-Stokes equation with long time (NS 2d-LT), 2D Wave equation with long time (Wave 2d-MS) and Kuramoto-Sivashinsky equation (KS). In addition to the challenges of high dimensionality and complex geometry mentioned by PINNacle, we discover unique challenges in these tasks caused by long periods and high-order derivatives of governed PDEs, making them extremely challenging for current PINNs. To solve these problems, we might need more powerful PINN backbones. Since we mainly focus on the PINN training paradigm, we omit the abovementioned 4 tasks to avoid the meaningless comparison and experiment with the left 16 tasks. Our datasets are summarized in Table 4.

## C.2 Metrics

In our experiments, we adopt the following three metrics. Training loss, rMAE and rMSE. And the training loss has been defined in Eq. (2). Here are the calculations for rMSE and rMAE:

$$\text{rMAE:} \sqrt{\frac{\sum_{\boldsymbol{x} \in \mathcal{S}} |u_\theta(\boldsymbol{x}) - u_*(\boldsymbol{x})|}{\sum_{\boldsymbol{x} \in \mathcal{S}} |u_*(\boldsymbol{x})|}} \quad \text{rMSE:} \sqrt{\frac{\sum_{\boldsymbol{x} \in \mathcal{S}} (u_\theta(\boldsymbol{x}) - u_*(\boldsymbol{x}))^2}{\sum_{\boldsymbol{x} \in \mathcal{S}} (u_*(\boldsymbol{x}))^2}}, \tag{71}$$

where $u_*$ denotes the ground truth solution. Note that the model output and ground truth can be negative and positive, respectively. Thus, these two metrics could be larger than 1.

## C.3 Implementations

For classical base models PINN [36], QRes [3] and FLS [50], we adopt the conventional configuration from previous papers [58]. As for the latest model PINNsFormer [58] and KAN [28], we use their official code. Next, we will detail the implementations of optimization algorithms.

**RoPINN**  As we described in the main text, we set the initial region size $r = 10^{-4}$, past iteration number $T_0 \in \{5, 10\}$ and only sample 1 point for each region at each iteration for all datasets. The corresponding analyses have been included in Figure 2 for $r$, Figure 3 for sampling points and Appendix D.1 for $T_0$ to demonstrate the algorithm property under different hyperparameter settings.

In addition, our formalization for region optimization in Eq. (8) only involves the equation, initial and boundary conditions, where we can still calculate their loss values after random sampling in the extended region. This definition perfectly matches the setting of 1D-Reaction, 1D-Wave and Convection. However, in PINNacle [12], some tasks also involve the data loss term, such as the inverse problem (Appendix D.2), which means we can only obtain the correct values for several observed or pre-calculated points. Since these points are pre-selected, we cannot obtain their new values after sampling. Thus, we do not apply region sampling to these points in our experiments. Actually, the data loss term only involves the forward process of deep models, which is a pure data-driven paradigm and is distinct from the other PDE-derived terms in PINNs. Therefore, the previous methods, gPINN and vPINN, also do not consider the data loss term in their algorithms.

**gPINN**  For the first three benchmarks, we add the first-order derivatives for spatial and temporal dimensions as the regularization term. We also search the weights of regularization terms in $\{1, 0.1, 0.01\}$ and report the best results. As for the PINNacle, we report the results of canonical PINN following their paper [12] and experiment with other base models by only replacing the model.

**vPINN**  We follow the code base in PINNacle, and implement it to the first three benchmarks. The test functions are set as Legendre polynomials and the test function number is set as 5. The number of points used to compute the integral within domain $\Omega$ is set as 10, and the number of grids is set differently for each subtask, with values of $\{4, 8, 16, 32\}$ for PINNacle and the same to other baselines for the first three benchmarks.

**Other baselines**  In Table 3 of the main text, we also experiment with the loss-reweighting method NTK [47] and data-resampling method RAR [51]. For NTK, we follow their official code and recalculate the neural tangent kernel to update loss weights every 10 iterations. And the kernel size is set as 300. As for RAR, we use the *residual-based adaptive refinement with distribution* algorithm.

# D  Additional Results

In this section, we provide more results as a supplement to the main text, including additional hyperparameter analysis, new experiments and more showcases.

## D.1  Hyperparameter Sensitivity on $T_0$

As we stated in Algorithm 1, we adopt the gradient variance of past $T_0$ iterations to approximate the sampling error defined in Theorem 3.9. In our experiments, we choose $T_0$ from $\{5, 10\}$, which can

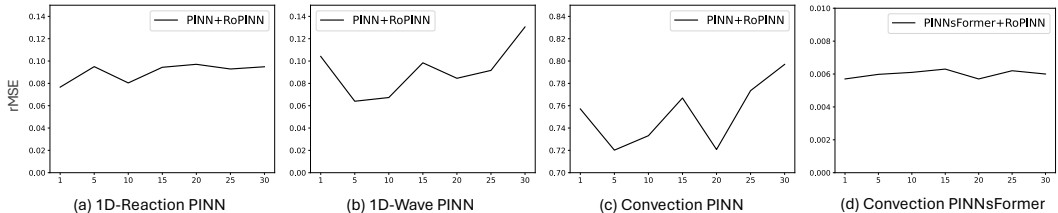

Figure 8: Hyperparameter analyses for $T_0$ in RoPINN based on PINN [36] and PINNsFormer [58] on different benchmarks. We change $T_0$ in $\{1, 5, 10, 15, 20, 25, 30\}$ and record the rMSE.

achieve consistently good and stable performance among different benchmarks and PDEs. To analyze the effect of this hyperparameter, we further add experiments with different choices in Figure 8.

As shown in Figure 8, we can find that under all the choices in $\{1, 5, 10, 15, 20, 25, 30\}$, RoPINN performs better than the vanilla PINN. Specifically, in both 1D-Reaction and 1D-Wave (Figure 8(a-b)), the model performs quite stable under different choices of $T_0$. As for Convection in Figure 8(c-d), the influence of $T_0$ is relatively significant in PINN. This may caused by the deficiency of PINN in solving Convection, where all the PINN-based experiments fail to generate an accurate solution for Convection (rMSE>0.5, Table 2). If we adopt a more powerful base model, such as PINNsFormer [58], this sensitivity will be alleviated. Also, it is worth noticing that, even though in Convection, RoPINN surpasses the vanilla PINN under all hyperparameter settings of $T_0$.

Besides, we can observe that the model performance slightly decreases when we set $T_0$ with a relatively large value. This may come from the difference between parameters $\theta_t$ and $\theta_{t+29}$, which will make the gradient variance approximation less reliable (Eq. (58) in the Theorem 3.11 proof).

## D.2 Experiments with Data Loss (Inverse Problem)

As we stated in the implementations (Appendix C.3), RoPINN can also be applied to tasks with data loss. Here we also include an inverse problem in PINNacle to testify to the performance of RoPINN in this case, which requires the model to reconstruct the diffusion coefficients of the Poisson equation from observations on 2500 uniform grids with additional Gaussian noise.

Table 5: Experiments on the Possion inverse problem (PInv) of PINNacle.

| Method | rMAE | rMSE |
|---|---|---|
| PINN [36] | 7.3e-2 | 8.2e-2 |
| +gPINN [55] | 7.3e-2(-0.2%) | 8.0e-2(2.1%) |
| +vPINN [18] | 1.3e+0 | 1.8e+0 |
| +RoPINN | **6.7e-2**(8.8%) | **7.3e-2**(11.4%) |

As presented in Table 5, in this task, RoPINN can also boost the performance of PINN with over 10% in the rMSE metric and outperform the other baselines (gPINN and vPINN) that cannot bring improvements. Note that in this experiment, we failed to reproduce the performance of vPINN reported by PINNacle. Thus, we report the results of vPINN by directly running the official code in PINNacle.

## D.3 Standard Deviations

Considering the limited resources, we repeat all the experiments on the first three typical benchmarks and our method on the PINNacle three times and other experiments one time. The official paper of PINNacle has provided the standard deviations for PINN, gPINN and vPINN on all benchmarks.

We summarize the standard deviations of PINN in Table 6. As for other base models, the standard deviations of FLS, QRes and KAN are within 0.005 on 1D-Wave and Convection, and within 0.001 for 1D-Reaction. PINNsFormer's standard deviations are smaller than 0.001 for all three benchmarks.

Table 6: Standard deviations for canonical PINN on three typical benchmarks. The confidence for RoPINN achieving the best performance is over 99% in all three benchmarks.

| rMSE±Standard Deviations | 1D-Reaction | 1D-Wave | Convection |
|---|---|---|---|
| PINN [36] | 0.981±5e-4 | 0.335±1e-3 | 0.840±5e-4 |
| +gPINN [55] | 0.978±3e-4 | 0.399±3e-3 | 0.935±3e-3 |
| +vPINN [18] | 0.982±3e-3 | 0.173±1e-3 | 0.743±2e-3 |
| +RoPINN (Ours) | **0.095**±8e-4 | **0.064**±1e-3 | **0.720**±2e-3 |

## D.4 More Showcases

As a supplement to the main text, we provide the showcases of RoPINN in Figure 9. From these showcases, we can observe that RoPINN can consistently boost the model performance and benefit the solving process of boundaries, discontinuous phases and periodic patterns.

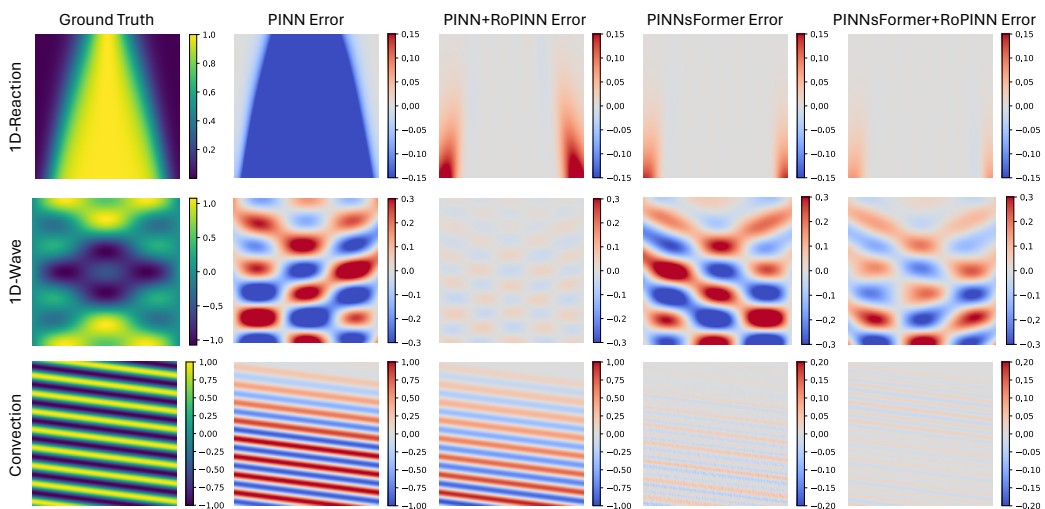

Figure 9: Showcases of RoPINN on the first three datasets based on PINN and PINNsFormer.

## D.5 Experiments with Advanced Quadrature Methods

In our implementation, RoPINN employs a simple Monte Carlo sampling to approximate integral. Obviously, we can adopt more advanced quadrature methods, such as Gaussian quadrature [16]. Thus, we also experiment with 2D space Gaussian quadrature, which requires square number points and the one-point-sampling situation will degenerate to the center value. As shown in Table 7, we can find that under our official setting (only sampling one point), Monte Carlo is better, while Gaussian quadrature is better in more points. Note that although Gaussian quadrature

Table 7: Comparison between Monte Carlo approximation (*Monte Carlo*) and Gaussian quadrature (*Gaussian*) on 1D Reaction. rMSE is recorded.

| PINN+RoPINN | Monte Carlo | Gaussian |
|---|---|---|
| Sample 1 Point | **0.095** | 0.109 |
| Sample 4 Points | 0.066 | **0.059** |
| Sample 9 Points | 0.033 | **0.030** |

has the potential to achieve better performance, sampling more points may contradict our motivation of boosting PINNs without extra backpropagation or gradient calculation. Thus, we choose the Monte Carlo method, which works better under high-efficiency settings.

## E  Full Results on PINNacle

In Table 2 of the main text, due to the context limitation, we only present the proportion of improved tasks over the total tasks. Here we provide the complete results for 5 based models for PINNacle (16 different tasks) in Table 8 and Table 9, where we can have the following observations:

- *RoPINN presents favorable generality in varied PDEs and base models.* As we described in Table 4, this benchmark contains of extensive physics phenomena. It is impressive that our proposed RoPINN can boost the performance of such extensive base models on a wide scope of PDEs, highlighting the generalizability of our algorithm.

- *RoPINN is numerically stable and efficient for computation.* As a training paradigm, RoPINN does not require extra gradient calculation and also does not add sampled points, which makes the algorithm computation efficient. In contrast, other baselines may generate poor results or encounter NaN or OOM problems in some PDEs.

Table 8: Full results of gPINN [55], vPINN [18] and RoPINN under different base models on PINNacle [12] (16 different PDEs). A lower rMAE or rMSE with higher relative promotion indicates better performance. The promotion over vanilla is recorded in parentheses. For clarity, we highlight the value with blue if it surpasses the vanilla PINN, gray if it fails (over 10 times worse than the vanilla PINN), or is numerically unstable (NaN) or out-of-memory (OOM).

| (Part I) | | PDE | Vanilla | | gPINN [55] | | vPINN [18] | | RoPINN (Ours) | |
|---|---|---|---|---|---|---|---|---|---|---|
| | | | rMAE | rMSE | rMAE | rMSE | rMAE | rMSE | rMAE | rMSE |
| **PINN [36]** | Burges | 1d-C | 1.1e-2 | 3.3e-2 | 3.7e-1 | 5.1e-1 | 4.0e-2(-272.2%) | 3.5e-1(-952.3%) | **9.1e-3**(15.8%) | **1.4e-2**(56.6%) |
| | | 2d-C | 4.5e-1 | 5.2e-1 | 4.9e-1(-8.7%) | 5.4e-1(-3.8%) | 6.6e-1(-46.4%) | 6.4e-1(-23.0%) | **4.3e-1**(4.0%) | **4.9e-1**(5.3%) |
| | Poisson | 2d-C | 7.5e-1 | 6.8e-1 | 7.7e-1(-3.0%) | 7.0e-1(-4.0%) | 4.6e-1(38.6%) | **4.9e-1**(27.4%) | **4.1e-1**(44.9%) | 6.6e-1(2.5%) |
| | | 2d-CG | 5.4e-1 | 6.6e-1 | 7.4e-1(-37.1%) | 7.9e-1(-21.0%) | **2.4e-1**(54.7%) | **2.9e-1**(56.4%) | 4.1e-1(24.1%) | 6.0e-1(8.1%) |
| | | 3d-CG | **4.2e-1** | 5.0e-1 | 4.3e-1(-4.2%) | 5.2e-1(-3.5%) | 8.0e-1(-91.4%) | 7.4e-1(-46.7%) | 4.7e-1(-11.9%) | **4.6e-1**(8.7%) |
| | | 2d-MS | 7.8e-1 | 6.4e-1 | **6.7e-1**(13.2%) | **6.2e-1**(2.3%) | 9.6e-1(-23.6%) | 9.7e-1(-52.1%) | 7.7e-1(0.4%) | 6.4e-1(0.3%) |
| | Heat | 2d-VC | 1.2e+0 | 9.8e-1 | 1.9e+1 | 1.6e+1 | 8.8e-1(26.9%) | 9.4e-1(4.5%) | **8.7e-1**(27.6%) | **7.9e-1**(19.7%) |
| | | 2d-MS | 4.7e-2 | 6.9e-2 | 1.0e+0 | 8.5e-1 | 9.3e-1 | 9.3e-1 | **4.4e-2**(6.4%) | **3.4e-2**(51.1%) |
| | | 2d-CG | 2.7e-2 | 2.3e-2 | 1.9e-1(-588.8%) | 2.1e-1(-787.3%) | 3.1e+0 | 9.3e-1 | **1.5e-2**(43.5%) | **2.0e-2**(12.6%) |
| | NS | 2d-C | 6.1e-2 | 5.1e-2 | 6.4e-1(-958.6%) | 4.9e-1(-877.5%) | 2.0e-1(-225.2%) | 2.9e-1(-478.1%) | **4.1e-2**(32.2%) | **4.2e-2**(16.1%) |
| | | 2d-CG | 1.8e-1 | 1.1e-1 | 4.2e-1(-134.5%) | 2.9e-1(-167.6%) | 9.9e-1(-454.6%) | 9.9e-1(-812.2%) | **1.5e-1**(19.0%) | **9.8e-2**(10.2%) |
| | Wave | 1d-C | 5.5e-1 | 5.5e-1 | 7.0e-1(-27.0%) | 7.2e-1(-31.9%) | 1.4e+0(-155.3%) | 8.4e-1(-53.5%) | **3.8e-1**(31.1%) | **3.9e-1**(28.0%) |
| | | 2d-CG | 2.3e+0 | 1.6e+0 | 9.9e-1(56.6%) | 1.0e+0(37.4%) | 1.1e+0(52.8%) | 8.0e-1(50.5%) | **7.1e-1**(68.8%) | **7.9e-1**(51.3%) |
| | Chaotic | GS | 2.1e-1 | 9.4e-2 | 3.4e-2(-61.0%) | 9.5e-2(-1.0%) | 8.9e-1 | 1.2e+0 | **2.1e-2**(2.1%) | **9.3e-2**(0.4%) |
| | High-dim | PNd | 1.2e-3 | 1.1e-3 | 2.6e-3(-119.5%) | 2.7e-3(-137.7%) | NaN | NaN | **6.7e-4**(42.9%) | **6.4e-4**(43.6%) |
| | | HNd | 1.2e-3 | 5.3e-3 | 3.6e-3(71.0%) | 4.6e-3(13.6%) | NaN | NaN | **5.6e-4**(95.5%) | **7.3e-4**(86.2%) |
| **Proportion of improved tasks** | | | 18.8% | 18.8% | 25.0% | 25.0% | 93.8% | 100.0% | | |
| **QRes [3]** | Burges | 1d-C | 5.8e-3 | 2.0e-2 | 3.6e-1 | 5.1e-1 | 2.0e-2(-241.0%) | 8.1e-2(-309.3%) | **5.7e-3**(1.4%) | **1.8e-2**(6.8%) |
| | | 2d-C | 3.2e-1 | 4.8e-1 | 4.9e-1(-53.1%) | 5.4e-1(-11.1%) | 1.1e+0(-256.9%) | 1.3e+0(-161.2%) | **3.2e-1**(0.3%) | **4.7e-1**(1.8%) |
| | Poisson | 2d-C | 2.9e-1 | 6.8e-1 | 7.6e-1(-159.4%) | 7.2e-1(-5.9%) | 3.2e-1(-7.6%) | **3.0e-1**(55.6%) | **2.9e-1**(0.6%) | 7.0e-1(-3.7%) |
| | | 2d-CG | 3.1e-1 | 7.4e-1 | 7.3e-1(-136.9%) | 7.8e-1(-6.0%) | 6.4e-1(-108.9%) | 7.3e-1(0.4%) | **2.9e-1**(4.0%) | **7.2e-1**(2.5%) |
| | | 3d-CG | 9.0e-2 | 5.8e-1 | 5.9e-1(-558.8%) | 6.4e-1(-10.7%) | 8.0e-1(-790.4%) | 7.4e-1(-28.4%) | **8.9e-2**(1.2%) | **5.6e-1**(3.5%) |
| | | 2d-MS | 1.7e+0 | 7.9e-1 | **5.5e-1**(67.2%) | **5.0e-1**(36.5%) | 9.7e-1(41.8%) | 9.8e-1(-24.4%) | 1.9e+0(-13.1%) | 9.1e-1(-15.0%) |
| | Heat | 2d-VC | 2.0e-1 | 1.3e+0 | 7.2e+0 | 5.9e+0(-363.0%) | 3.3e-1(-63.9%) | **3.2e-1**(74.9%) | **1.6e-1**(19.6%) | 1.1e+0(16.8%) |
| | | 2d-MS | 1.7e-2 | 1.4e-1 | 4.4e-1 | 3.5e-1(-158.6%) | 4.0e-1 | 3.8e-1(-174.6%) | **8.7e-3**(48.5%) | **7.2e-2**(47.2%) |
| | | 2d-CG | **1.9e-2** | 2.4e-2 | 1.1e-1(-476.7%) | 1.3e-1(-440.1%) | 6.1e-1 | 7.0e-1 | 2.0e-2(-5.2%) | **2.1e-2**(9.9%) |
| | NS | 2d-C | 3.9e-3 | 4.5e-2 | 3.9e-1 | 3.0e-1(-565.9%) | 2.8e-1 | 2.4e-1(-427.7%) | **2.1e-3**(47.6%) | **3.4e-2**(23.6%) |
| | | 2d-CG | 1.2e-2 | 7.7e-2 | 2.5e-1 | 1.6e-1(-110.9%) | 1.0e+0 | 1.0e+0 | **1.2e-2**(0.8%) | **7.6e-2**(1.5%) |
| | Wave | 1d-C | 2.2e-1 | 4.8e-1 | 7.0e-1(-216.0%) | 7.1e-1(-47.7%) | **4.0e-2**(81.8%) | **4.7e-1**(2.5%) | 2.1e-1(3.6%) | 4.4e-1(8.6%) |
| | | 2d-CG | **1.5e-1** | **9.2e-1** | 9.8e-1(-572.5%) | 1.0e+0(-8.4%) | 1.3e+0(-773.5%) | 1.2e+0(-29.6%) | 2.2e-1(-53.6%) | 1.3e+0(-38.3%) |
| | Chaotic | GS | 1.1e-2 | 9.3e-2 | 2.0e-2(-74.5%) | 9.4e-2(-0.3%) | 8.6e-1 | 9.7e-1(-937.8%) | **9.9e-3**(13.2%) | **9.2e-2**(1.8%) |
| | High-dim | PNd | 1.5e-2 | 5.3e-3 | 3.4e-2(-132.2%) | 3.3e-2(-532.7%) | NaN | NaN | **5.7e-3**(60.5%) | **1.8e-3**(66.5%) |
| | | HNd | 2.9e-2 | 1.1e-2 | **2.1e-3**(92.7%) | **2.1e-3**(81.4%) | NaN | NaN | 2.1e-2(25.6%) | 8.3e-3(25.8%) |
| **Proportion of improved tasks** | | | 12.5% | 12.5% | 12.5% | 25.0% | 81.3% | 81.3% | | |
| **FLS [50]** | Burges | 1d-C | 9.0e-3 | 1.4e-2 | 3.3e-1 | 4.8e-1 | 6.5e-2(-620.9%) | 3.0e-1 | **9.0e-3**(0.2%) | **1.3e-2**(8.6%) |
| | | 2d-C | 4.4e-1 | 4.9e-1 | 4.9e-1(-13.0%) | 5.4e-1(-10.1%) | 1.3e+0(-195.0%) | 1.3e+0(-171.4%) | **4.3e-1**(0.4%) | **4.9e-1**(0.2%) |
| | Poisson | 2d-C | **6.8e-1** | 6.7e-1 | 7.3e-1(-6.6%) | 6.7e-1(-1.2%) | 9.0e-1(-30.9%) | 9.4e-1(-40.5%) | 7.2e-1(-5.4%) | **6.3e-1**(5.2%) |
| | | 2d-CG | 6.3e-1 | 7.0e-1 | 7.8e-1(-22.7%) | 8.1e-1(-16.2%) | 6.0e-1(5.6%) | 7.0e-1(-0.6%) | **5.9e-1**(6.8%) | **6.9e-1**(1.2%) |
| | | 3d-CG | 4.2e-1 | 5.0e-1 | 4.5e-1(-7.8%) | **4.6e-1**(9.5%) | 8.1e-1(-95.5%) | 7.5e-1(-48.8%) | **4.1e-1**(1.3%) | 5.1e-1(-0.5%) |
| | | 2d-MS | 8.9e-1 | 7.5e-1 | **5.1e-1**(43.5%) | **4.7e-1**(36.4%) | 9.0e-1(-0.0%) | 9.4e-1(-25.7%) | 8.1e-1(9.5%) | 6.7e-1(10.0%) |
| | Heat | 2d-VC | 1.5e+0 | 1.3e+0 | 3.4e+1 | 2.7e+1 | 1.2e+0(16.9%) | 1.1e+0(11.5%) | **1.2e+0**(21.1%) | **1.1e+0**(17.2%) |
| | | 2d-MS | **6.2e-1** | **4.8e-2** | 1.2e+0 | 9.0e-1 | 1.0e+0 | 9.9e-1 | 1.4e-1(-132.3%) | 8.4e-2(-74.2%) |
| | | 2d-CG | 1.5e-2 | 2.5e-2 | 9.9e-2(-553.1%) | 1.2e-1(-394.9%) | 3.4e+0 | 4.4e+0 | **1.2e-2**(20.4%) | **2.5e-2**(0.6%) |
| | NS | 2d-C | 8.3e-2 | 6.9e-2 | 4.9e-1(-483.0%) | 3.7e-1(-430.2%) | 2.8e-1(-236.9%) | 2.5e-1(-262.9%) | **4.8e-2**(42.4%) | **4.9e-2**(29.3%) |
| | | 2d-CG | 1.8e-1 | 1.2e-1 | 3.9e-1(-119.7%) | 2.7e-1(-124.6%) | 9.9e-1(-458.2%) | 1.0e+0(-727.3%) | **1.4e-1**(19.8%) | **9.9e-2**(18.0%) |
| | Wave | 1d-C | 4.0e-1 | 4.1e-1 | 6.3e-1(-57.7%) | 6.4e-1(-55.7%) | **1.1e-2**(97.2%) | **1.1e-2**(97.3%) | 3.8e-1(5.3%) | 3.9e-1(4.9%) |
| | | 2d-CG | 2.5e+0 | 2.4e+0 | **1.1e+0**(58.7%) | 1.9e+0(22.4%) | 2.1e+0(19.1%) | 2.0e+0(15.8%) | 1.7e+0(31.6%) | **1.7e+0**(30.4%) |
| | Chaotic | GS | **2.1e-2** | 9.4e-2 | 2.6e-1 | 2.4e-1(-155.5%) | 9.7e-1 | 1.0e+0 | 2.2e-2(-5.8%) | **9.0e-2**(3.9%) |
| | High-dim | PNd | 8.9e-4 | 1.0e-3 | 2.0e-3(-123.0%) | 2.3e-3(-127.2%) | NaN | NaN | **5.4e-4**(39.6%) | **6.4e-4**(37.5%) |
| | | HNd | 3.7e-3 | 4.0e-3 | 9.6e-3(-160.8%) | 9.9e-3(-144.2%) | NaN | NaN | **1.3e-3**(63.8%) | **1.4e-3**(66.1%) |
| **Proportion of improved tasks** | | | 12.5% | 18.8% | 25.0% | 18.8% | 81.3% | 87.5% | | |

Table 9: Full results of gPINN [55], vPINN [18] and RoPINN under different base models on PINNacle [12] (16 different PDEs). A lower rMAE or rMSE with higher relative promotion indicates better performance. The promotion over vanilla is recorded in parentheses. For clarity, we highlight the value with  blue  if it surpasses the vanilla PINN,  gray  if it fails (over 10 times worse than the vanilla PINN), or is numerically unstable (NaN) or out-of-memory (OOM). For PINNsFormer, it fails in most of the tasks due to the OOM problem. We omit these tasks in calculating proportion.

| (Part II) | PDE | | Vanilla | | gPINN [55] | | vPINN [18] | | RoPINN (Ours) | |
|---|---|---|---|---|---|---|---|---|---|---|
| | | | rMAE | rMSE | rMAE | rMSE | rMAE | rMSE | rMAE | rMSE |
| PINNsformer [58] | Burges | 1d-C | 9.3e-3 | 1.4e-2 | 6.5e-1 | 6.7e-1 | 5.5e-1 | 5.7e-1 | **8.0e-3**(13.3%) | **1.0e-2**(26.5%) |
| | | 2d-C | OOM | OOM | OOM(-%) | OOM(-%) | OOM(-%) | OOM(-%) | OOM(-%) | OOM(-%) |
| | Poisson | 2d-C | 7.2e-1 | 6.6e-1 | 1.0e+0(-39.4%) | 1.0e+0(-52.7%) | 1.0e+0(-39.4%) | 1.0e+0(-52.7%) | **6.9e-1**(4.1%) | **6.2e-1**(5.9%) |
| | | 2d-CG | 5.4e-1 | 6.3e-1 | 1.0e+0(-86.6%) | 1.0e+0(-59.4%) | 1.0e+0(-86.6%) | 1.0e+0(-59.4%) | **4.7e-1**(13.0%) | **5.5e-1**(12.1%) |
| | | 3d-CG | OOM | OOM | OOM(-%) | OOM(-%) | OOM(-%) | OOM(-%) | OOM(-%) | OOM(-%) |
| | | 2d-MS | 1.3e+0 | 1.1e+0 | OOM | OOM | OOM | OOM | **7.2e-1**(42.4%) | **6.0e-1**(45.0%) |
| | Heat | 2d-VC | OOM | OOM | OOM(-%) | OOM(-%) | OOM(-%) | OOM(-%) | OOM(-%) | OOM(-%) |
| | | 2d-MS | OOM | OOM | OOM(-%) | OOM(-%) | OOM(-%) | OOM(-%) | OOM(-%) | OOM(-%) |
| | | 2d-CG | OOM | OOM | OOM(-%) | OOM(-%) | OOM(-%) | OOM(-%) | OOM(-%) | OOM(-%) |
| | NS | 2d-C | OOM | OOM | OOM(-%) | OOM(-%) | OOM(-%) | OOM(-%) | OOM(-%) | OOM(-%) |
| | | 2d-CG | 1.0e-1 | 7.0e-2 | 7.4e-1(-626.0%) | 7.2e-1(-939.9%) | 1.0e+0(-870.9%) | 1.0e+0 | **9.8e-2**(4.7%) | **6.3e-2**(9.0%) |
| | Wave | 1d-C | 5.0e-1 | 5.1e-1 | 8.3e-1(-63.5%) | 8.5e-1(-65.5%) | 5.2e-1(-3.5%) | 5.3e-1(-3.4%) | **4.7e-1**(7.4%) | **4.8e-1**(6.8%) |
| | | 2d-CG | OOM | OOM | OOM(-%) | OOM(-%) | OOM(-%) | OOM(-%) | OOM(-%) | OOM(-%) |
| | Chaotic | GS | OOM | OOM | OOM(-%) | OOM(-%) | OOM(-%) | OOM(-%) | OOM(-%) | OOM(-%) |
| | High dim | PNd | OOM | OOM | OOM(-%) | OOM(-%) | OOM(-%) | OOM(-%) | OOM(-%) | OOM(-%) |
| | | HNd | OOM | OOM | OOM(-%) | OOM(-%) | OOM(-%) | OOM(-%) | OOM(-%) | OOM(-%) |
| **Proportion of improved tasks** | | | 0.0% | 0.0% | 0.0% | 0.0% | 100.0% | 100.0% | | |
| KAN [28] | Burges | 1d-C | 2.7e-1 | 5.4e-1 | 5.6e-1(-111.8%) | 6.7e-1(-24.0%) | 3.9e-1(-47.5%) | 5.5e-1(-1.3%) | **2.4e-1**(8.2%) | **4.8e-1**(12.5%) |
| | | 2d-C | 8.7e-1 | 9.4e-1 | 1.5e+0(-71.8%) | 1.7e+0(-83.5%) | NaN | NaN | **8.3e-1**(4.6%) | **9.2e-1**(2.4%) |
| | Poisson | 2d-C | 7.3e-1 | 6.8e-1 | 7.4e-1(-1.4%) | 6.8e-1(-0.1%) | **1.6e-1**(78.2%) | **1.5e-1**(78.4%) | 6.3e-1(13.4%) | 6.8e-1(-0.1%) |
| | | 2d-CG | 7.7e-1 | 8.1e-1 | 8.1e-1(-5.5%) | 8.6e-1(-6.9%) | 6.5e-1(14.9%) | 7.3e-1(9.8%) | **6.5e-1**(15.8%) | **6.9e-1**(14.0%) |
| | | 3d-CG | 7.5e-1 | 1.4e+0 | 2.0e+0(-163.6%) | 1.8e+0(-25.5%) | NaN | NaN | **7.2e-1**(4.1%) | **6.8e-1**(51.8%) |
| | | 2d-MS | 9.5e-1 | 9.8e-1 | 1.0e+0(-5.0%) | 1.0e+0(-2.7%) | 9.9e-1(-4.6%) | 1.0e+0(-2.3%) | **9.2e-1**(3.4%) | **9.6e-1**(2.1%) |
| | Heat | 2d-VC | 1.9e+1 | 1.5e+1 | 3.3e+0(82.5%) | 2.7e+0(81.8%) | **8.9e-1**(95.3%) | **9.1e-1**(94.0%) | 6.0e+0(68.7%) | 4.7e+0(68.4%) |
| | | 2d-MS | 1.4e+0 | 1.1e+0 | 2.0e+0(-39.8%) | 1.4e+0(-28.4%) | 8.3e-1(42.8%) | 7.5e-1(31.4%) | **7.4e-1**(49.1%) | **6.9e-1**(36.8%) |
| | | 2d-CG | 5.0e-1 | 5.3e-1 | 8.2e-1(-63.1%) | 7.3e-1(-37.2%) | 8.1e-1(-60.8%) | 9.0e-1(-69.7%) | **4.9e-1**(1.7%) | **5.0e-1**(4.9%) |
| | NS | 2d-C | 5.0e-1 | 4.1e-1 | 8.1e-1(-60.6%) | 6.7e-1(-65.8%) | **2.1e-1**(57.7%) | **1.7e-1**(58.6%) | 4.1e-1(18.2%) | 3.9e-1(2.9%) |
| | | 2d-CG | 9.9e-1 | 6.4e-1 | 5.4e-1(45.6%) | 4.1e-1(36.7%) | 1.0e+0(-1.1%) | 1.0e+0(-56.1%) | **4.6e-1**(53.7%) | **3.5e-1**(44.8%) |
| | Wave | 1d-C | 4.7e-1 | 4.7e-1 | 8.6e-1(-85.2%) | 8.8e-1(-89.2%) | **1.9e-1**(58.4%) | **2.1e-1**(55.8%) | 4.6e-1(1.7%) | 4.6e-1(1.9%) |
| | | 2d-CG | 1.7e+0 | 1.6e+0 | 1.1e+0(37.4%) | 1.1e+0(33.4%) | NaN | NaN | **9.8e-1**(43.2%) | **9.6e-1**(41.1%) |
| | Chaotic | GS | NaN | NaN | 1.1e+0(-%) | 9.4e-1(-%) | 8.5e-1(-%) | 9.6e-1(-%) | **7.5e-1**(-%) | **7.0e-1**(-%) |
| | High-dim | PNd | 4.6e-4 | 5.6e-4 | 2.8e-3(-515.3%) | 3.5e-3(-518.4%) | NaN | NaN | **3.7e-4**(19.9%) | **5.3e-4**(5.5%) |
| | | HNd | 1.9e-3 | 5.4e-3 | **5.9e-4**(68.9%) | 8.0e-4(85.3%) | NaN | NaN | 1.5e-3(20.8%) | **6.5e-4**(88.0%) |
| **Proportion of improved tasks** | | | 31.3% | 31.3% | 43.8% | 43.8% | 100.0% | 93.8% | | |

# F   Related Work

This section will discuss some related works as a supplement to Section 2. We will first discuss some PINN research and then we will also clarify some looking similar but completely distinct topics.

**PINN optimizers**   As we mentioned in the second paragraph of the introduction, many previous works focus on developing efficient and effective deep-model optimizers for PINNs [55, 37], which may help the optimization process tackle the ill-conditioned Hessian matrix or naturally balance multiple loss terms [53]. As we formalized in Algorithm 1, RoPINN is not restricted to a certain optimizer. The researchers can easily replace the Adam [21] or L-BFGS [27] with other advanced optimizers. Since we mainly focus on the objective function, these works are orthogonal to us.

**Numerical differentiation for objective functions**   In addition to the regularization or variational-based methods, some researchers attempt to replace the automatic differentiation with numerical

approximations [39, 9], which can tackle the expensive computation cost caused by calculating high-order derivatives. However, this paradigm does not attempt to change the objection function definition, just focuses on the calculation of point optimization PINN loss, which is distinct from our proposed region optimization paradigm.

**Sampling-based methods** Strategies for sampling collocation points perform an important role in training PINN models [46]. Previous sampling-based methods mainly focus on accumulating collocation points to high-residual areas [51, 23] or considering the temporal causality [45], whose theoretical analyses are usually based on the quadrature theorem [32]. Distinct from these methods, RoPINN is motivated by the optimization deficiency of PINN models and can be seamlessly integrated with sampling-based methods with significant promotion (Table 3), indicating that RoPINN works orthogonally to sampling methods. Besides, the theoretical analysis of RoPINN also starts from the optimization perspective, which reveals that one key advancement of RoPINN is a better balance between optimization error and generalization error (Theorem 3.12).

In addition, RoPINN is also distinct from data augmentation or adversarial training techniques in the following aspects: (1) Theorem difference: although our proposed practical algorithm is based on Monte Carlo sampling in a region, the underlying theoretical support and insights are a region-based objective function (Theorem 3.5). (2) Implementation difference: In our algorithm, not only is the input changed, but the objective is also correspondingly changed. Thus, this paper is foundationally different from augmentation and adversarial training in that the ground truth label is fixed. Our design is tailored to the physics-informed loss function of PINNs, where we can accurately calculate the equation residual at any point within the input domain.

# G  Limitations

This paper presents region optimization as a new PINN training paradigm and provides both theoretical analysis and practical algorithms, supported by extensive experiments. However, there are still several limitations. In the theoretical analysis, we assume that the canonical loss function is $L$-Lipschitz-$\beta$-smooth, which may not be guaranteed in practice. Besides, RoPINN involves several hyperparameters, such as initial region size $r$, and number of past iterations $T_0$. Although we have studied the sensitivity w.r.t. them in Figures 2 and Appendix D.1 and demonstrate that they are easy to tune in most cases, we still need to adjust them for better performance in practice.

According to our experiments and theorems, we provide some recipes for hyperparameter tuning in the following, which may be helpful to the usage of RoPINN:

- As shown in Figure 2, region size $r$ will be progressively adjusted by RoPINN. Setting $r$ in $[10^{-6}, 10^{-4}]$ can work well. According to Theorem 3.12, the choice of $r$ should balance optimization and generalization, which may be inherently decided by the PDE smoothness.

- As analyzed in Figures 3 and 4, sampling 1-30 points can gain consistent promotion but will linearly increase the computation costs. Following our default setting (sampling one point) can already achieve a competitive performance in a wide range of PDEs.

- As presented in Figure 8, number of past iterations $T_0$ is easy to tune in $[1, 20]$. Setting $T_0 \in \{5, 10\}$ can be a good choice, which has been widely verified in our paper.

- Some hyperparameter tuning tools, such as Weights and Bias (Wandb[1]), may mitigate this limitation to some extent, which has already been used in previous related work [37].

# H  Broader Impacts

In this paper, we develop a new region optimization training paradigm for PINNs and provide both theorem analyses and practical algorithms. This new perspective may inspire the subsequent research of PINNs, especially rethinking the canonical objective function. In addition, our proposed RoPINN shows favorable efficiency and generalizes well in different base models and PDEs, which can be used to boost the precision of PINNs and generally benefit the downstream tasks, such as physics phenomenon simulation, biological property analysis, etc. Since we purely focus on the training algorithm of PINNs, there are no potential negative social impacts or ethical risks.

---

[1]https://github.com/wandb/wandb

