# OpenReview forum: "RoPINN: Region Optimized Physics-Informed Neural Networks"
_NeurIPS.cc/2024/Conference — NeurIPS 2024 poster_

### Official Review · Reviewer_zNBp · 2024-07-03

**Soundness:** 2
**Presentation:** 2
**Contribution:** 3
**Rating:** 4
**Confidence:** 3

**Summary:**

The preprint proposes to replace the collocation based PINN loss by a sum of local continuous integrals over regions around the collocation points. These continuous integrals are then again discretized using Monte Carlo integration with a single quadrature pint. The authors furthermore propose to adapt the region size during training using gradient statistics.

**Strengths:**

The authors report good empirical performance on a number of benchmark problems.

**Weaknesses:**

The introduction of continuous integrals over regions around the collocation points that subsequentially are discretized by Monte Carlo integration again seems tautological. After all, the loss function in PINNs is already a Monte Carlo discretization of a continuous integral (over the whole computational domain). Furthermore, the analysis that the authors present for the modified loss in equation (5) should not be carried out with the continuous integral over the regions $\Omega_r$ but with its Monte Carlo approximation. Otherwise, the comparison to the discretized PINN loss is unfair.

**Questions:**

I struggle to see why the proposed method should work theoretically. I acknowledge the adaptive nature of the regions for the sampling but struggle to see how this might help to accumulate integration points in regions of, e.g., high residual. A visualization that this, or something along these lines that explains why the proposed method works well, happens would be helpful. Furthermore, I am not convinced that the theoretical analysis presented is meaningful, as it analyzes the integrals over the region as continuous objects. Please comment or clarify misconception.

**Limitations:**

See questions

---

> ### Author Rebuttal · Authors · 2024-08-06
>
> # To Reviewer zNBp
>
> Many thanks to Reviewer zNBp for providing an insightful review and valuable suggestions.
>
> > **Clarify misconception.**
> >
> > "After all, the loss function in PINNs is already a Monte Carlo discretization over the whole computational domain."
>
> Firstly, we want to highlight that our theorem **considers the training process (see proof in $\underline{\text{Appendix A}}$)**, which is distinct from previous quadrature-based theorems [Siddhartha Mishra et al., IMAJNA 2023] that only focus on the integral approximation and do not consider the collocation point change during training.
>
> **Thus, it is clearly one-sided to think about our method only from the integral approximation view.** Under the model training context, the reviewer described "whole-domain-discretization PINN loss" corresponds to "(Global sampling) Point optimization" paradigm (defined below), which is different from the "point optimization" in our paper.
>
> |                        | Paradigms                            | Implementation                                               |
> | - | - | - |
> | Our paper              | Region optimization                  | sample within regions around collocation points and **keep changing during training.** |
> | Our paper              | (Fixed) Point optimization           | sample over the whole domain **at the beginning, but fixed during training** (canonical PINN [33 of our paper]) |
> | **Reviewer mentioned** | (Global sampling) Point optimization | sample over the whole domain and **keep changing during training** (RAR [46 of our paper]) |
>
> > **Q1:** "The introduction of continuous integrals over regions around the collocation points that subsequentially are discretized again seems tautological.
> >
> > "The analysis should not be carried out with the continuous integral over the regions Ω𝑟 but with its Monte Carlo approximation. The comparison to the discretized PINN loss is unfair."
>
> **(1) Requested theorem: region optimization with MC approximation.**
>
> As requested, we prove the generalization bound with MC approximation in $\underline{\text{Figure 3 of Global Response PDF}}$, which is extended from $\underline{\text{Theorem 3.5 of main text}}$ by incorporating gradient estimation error.
>
> For clarity, we present the convex version here. See the $\underline{\text{PDF}}$ for the non-convex version.
> $$
> \text{Generalization\ error}\leq ((1-|\Omega_r|/|\Omega|)L+\mathcal{E}_{r, \mathrm{grad}})\frac{2L}{|\mathcal{S}|}\sum\alpha_t
> $$
>
> $\mathcal{E}_{r, \mathrm{grad}}$ denotes the gradient estimation error caused by MC approximation.
>
> We can find that the above two point paradigms can be unified in our region optimization formalization:
>
> - **(Fixed) point optimization** corresponds to an extreme case: $|\Omega_r|=0$ and $\mathcal{E}_{r, \mathrm{grad}}=0$.
> - **(Global sampling) point optimization** is equivalent to another extreme case: $\Omega_r=\Omega$. Since the larger region is generally harder to approximate, this may cause a large $\mathcal{E}_{r, \mathrm{grad}}$.
> - **RoPINN** adopts the trust region calibration to adaptively adjust $|\Omega_r|$ during training.
>
> **Thus, introducing "region" is not tautological**, which
>
> - Provides a general theoretical framework for three paradigms.
> - Reveals the balance of generalization bound and optimization error.
> - Motivates RoPINN as a practical algorithm for balancing.
>
> **(2) All the experiments are fair.**
>
> Since we only sample 1 point in practice, RoPINN will not bring extra gradient calculation than "discretized PINN loss". **Under a comparable efficiency, RoPINN consistently boosts 5 different PINN models in 19 tasks ($\underline{\text{Tables 2,3,7,8}}$)**, which should not be overlooked.
>
> > **Q2:** "Why proposed method should work theoretically. I acknowledge the adaptive regions but struggle to see how this help to accumulate points in high residual."
>
> RoPINN is not a collocation-point sampling algorithm, whose contribution is orthogonal to RAR [46 of our paper] (see $\underline{\text{Table 3 of main text}}$). Next, we will show why RoPINN works well.
>
> **(1) Theoretical understanding.**
>
> As discussed in **Q1-(1)**, the generalization bound has two parts, the first term $(1-\frac{|\Omega_r|}{|\Omega|})$ is inversely proportional to $|\Omega_r|$ and the second term $\mathcal{E}_{r, \mathrm{grad}}$ is generally proportional to $|\Omega_r|$.
>
> - (Fixed) and (Global sampling) point optimization are just two special cases. Obviously, they **lack flexibility and cannot adaptively balance optimization and generalization**.
> - RoPINN presents an adaptive algorithm, which can **adjust the region size for a better balance of the above two terms**. A visualization of the balance process is in $\underline{\text{Figure 2 of main text}}$.
>
> **(2) Experiment statistics.**
>
> We also record the standard deviation of parameter gradients on the last 100 iterations, which reflects the training stability. Fixed point optimization leads to very stable training and Global Samlping brings more perturbations, while RoPINN achieves a relatively balanced value.
>
> | 1D-Reaction     | Paradigm                | rMSE      | Gradient Std           |
> | - | - | - | - |
> | **PINN+RoPINN** | **Region**              | **0.095** | 0.310                  |
> | PINN Vanilla    | (Fixed) Point           | 0.981     | 1.262$\times 10^{-10}$ |
> | PINN+RAR        | (Global Samlping) Point | 0.981     | 1.128                  |
>
> > **Q3:** "I am not convinced that the theoretical analysis presented is meaningful, as it analyzes the integrals over the region as continuous objects."
>
> Following your suggestion, we have extended our theorem to practical algorithms (see $\underline{\text{Q1-(1)}}$). Our original theorem analyses are under a perfect approximation, that is $\mathcal{E}_{r, \mathrm{grad}}=0$, which is still meaningful in:
>
> - Providing a unified framework for three paradigms.
> - Considerating the optimization process, thereby easily to be extended to practical algorithms.

---

> > ### Comment · Reviewer_zNBp · 2024-08-12
> >
> > I thank the authors for their replies. I still struggle to see the value of the proposed region optimization.
> >
> > Could the authors explain to me how the proposed method "informs" the sampling procedure? All sampling techniques for PINN type loss functions I am aware of reallocate points to regions that are challenging to learn for the network. This is informed by quantities that relate to how well the PDE is currently solved -- typically the PDE residual. I cannot really see that in your proposed method. Does this somehow implicitly happen?
> >
> > I acknowledge that Figure 2 in the main text illustrates that the regions change in size during the training process, but this does not answer my question.
> >
> > I remain critical of the work.

---

> ### Author Response · Authors · 2024-08-12
> **Thanks for your response and more explanations about how RoPINN works**
>
> We sincerely thank the reviewer's response and detailed descriptions of the question.
>
> We kindly remind you that our previous rebuttal has included the requested theorem for practical algorithm in $\underline{\text{Q1-(1)}}$, which shows that introducing "region" is not tautological and ensures a fair theoretical comparison w.r.t. discretized PINN loss. Thus, we think the theoretical value of introducing "region" is clear.
>
> Next, we will explain how RoPINN helps the PDE-Solving in practice.
>
> > Could the authors explain to me how the proposed method "informs" the sampling procedure?
>
> Firstly, we list the comparison between RoPINN and a sampling method RAR to clarify "what RoPINN informs the sampling" and "why it works". More details are included in the following.
>
> |                       | Information to sampling                                      | Intuitive understanding of benefits                          |
> | - | - | - |
> | RAR [46 of our paper] | PDE residual                                                 | Make the algorithm aware of which area is hard to solve.     |
> | RoPINN                | Gradient estimation error (gradient variance of successive iterations) | Make the algorithm aware if the current sampling range can bring a "good" optimization, which refers to a relatively stable training for convergence and is sufficient to make the model "explore" new areas for generalization. |
>
> **(1) RoPINN "informs" the sampling procedure with "gradient estimation error" (gradient variance of successive iterations).**
>
> As shown in $\underline{\text{Algorithm 1 of main text}}$, RoPINN sets the sampling region size $r$ of each iteration as $r/\sigma_t$, where $\sigma_t$ denotes the gradient variance of successive iterations. In $\underline{\text{Theorem 3.9, 3.11 of main text}}$, we have also proved that $\sigma_t$ can be used to approximate gradient estimation error. Thus, RoPINN's design can introduce the gradient estimation error (gradient variance) into the sampling procedure.
>
> Why introducing gradient estimation error is beneficial? Here are the explanations.
>
> - **Theoretical analysis:** As shown in $\underline{\text{Q1-(1) of previous rebuttal}}$, we have proved that in practice, the generalization bound is also affected by gradient estimation error. The detailed explanations for the effect of $r$ are included in the previous rebuttal. One key point is that a proper sampling region size $r$ can better balance $(1-|\Omega_r|/|\Omega|)L$ and $\mathcal{E}_{r, \mathrm{grad}}$, bringing a better performance. What RoPINN does is change $r$ adaptively.
> - **Intuitive understanding:** **The gradient estimation error of MC can be used to represent the “consistency” of optimization direction within a region** ($\underline{\text{Theorem 3.9 of main text}}$). A larger gradient estimation error (corresponding to a lower consistency of gradients in successive iterations) will lead to an unstable training process, which may overwhelm the model or even fail to converge. On the other side, too stable training is also insufficient to make the model "explore" new areas and damage the generalization. RoPINN can achieve a balanced result, which is supported by our new statistics in $\underline{\text{Q2-(2) of previous rebuttal}}$.
>
> The above "intuitive understanding" explanation has been partially discussed in $\underline{\text{Lines 207-210 of main text}}$. In the revised paper, we will rephrase this paragraph for a more detailed explanation.
>
> **(2) RoPINN can be combined with the "sampling techniques" that you mentioned.**
>
> Actually, in $\underline{\text{Q2 of previous rebuttal}}$, we have pointed out that the contribution of RoPINN is orthogonal to previous sampling methods. The key idea of RoPINN is to extend optimization targets from collocation points to their regions. Thus, RoPINN can be combined with your mentioned "sampling techniques". Here are part of the results. Please see $\underline{\text{Table 3 of main text}}$ for full results.
>
> | rMSE            | 1D-Reaction | 1D-Wave   | Convection |
> | - | - | - | - |
> | PINN            | 0.981       | 0.335     | 0.840      |
> | +RAR            | 0.981       | 0.126     | 0.771      |
> | +RoPINN         | 0.095       | 0.064     | 0.720      |
> | **+RAR+RoPINN** | **0.080**   | **0.030** | **0.695**  |
>
> To better explain how our model works, we will rephrase our paper by:
>
> - Incorporating the requested theorem in $\underline{\text{Q1-(1) of previous rebuttal}}$ in the main text to illustrate the balance between optimization and generalization.
> - Incorporating the new statistics in $\underline{\text{Q2-(2) of previous rebuttal}}$ to show the balanced results of RoPINN.
> - Explaining that region optimization is a general framework for Fixed or Global sampling point optimization.
> - Adding a new section to explain the relation and difference w.r.t. previous sampling methods.
>
> We hope these new results can resolve your concerns and we are happy to answer any further questions.

---

> ### Author Response · Authors · 2024-08-12
> **More discussions about sampling-based methods**
>
> **We believe that accumulating points in high-residual areas is NOT the only principle to design sampling-based methods.**
>
> Actually, although the previous sampling methods (e.g. RAR) can make the collocation point accumulate to high-residual areas, **they may also face optimization difficulty due to too many hard-to-optimize points.** For example, if one point is high-residual but extremely hard to optimize, the model optimization may be over-attracted by this point and be misguided.
>
> RoPINN is based on a distinct idea, which **considers the model optimization process**. Specifically, the calibrated region size $r$ can finely balance optimization difficulty and "exploration" to unseen areas (i.e. larger sampling regions).
>
> |                           | Design                       | Pros                                                         | Cons                                                        |
> | - | - | - | - |
> | Previous methods e.g. RAR | Sampling high-residual areas | Make the model solve hard areas better       |  optimization difficulty |
> | RoPINN                    | Adaptive sampling region size      | Balance optimization difficulty and "exploration" to unseen areas | Without specific optimization to hard areas        |
>
> The Cons of RoPINN can be solved by integrating RoPINN and RAR ($\underline{\text{Table 3 of main text}}$). Note that we are not saying that RoPINN is better than RAR. They contribute orthogonally.
>
> **We do hope that the reviewer can leave the sampling high-residual design behind (which is distinct from our design) and think about our paper based on our proposed "optimization-based" theorems.**
>
> Many thanks for your dedication to our paper.

---

> ### Comment · Reviewer_zNBp · 2024-08-13
>
> Thank you for the detailed answer. In fact, I can understand your point more clearly now and like the idea of informing the sampling based on gradient statistics instead of or in combination with residual-based methods.
>
> I still find the explanation in your write-up hard to digest, introducing regions around sampling points which are treated like integrals and then again discretized by one collocation point was confusing to me (and also to another reviewer) and I apologize it took some time before I could understand the underlying idea.
>
> Furthermore, I disagree that you propose an optimization method. You propose an adaptive sampling method with different criteria than the residual.
>
> To conclude, I will raise my score but I think the manuscript would benefit from a different presentation that puts the sampling viewpoint as the core novelty. I am still against publication in the current form but now mainly because of the presentation.

---

> ### Author Response · Authors · 2024-08-13
> **Thanks for your reply and acknowledging our "good" contribution**
>
> We sincerely thank you for your response, appreciating our idea and acknowledging our "good" contribution.
>
> > "I disagree that you propose an optimization method."
>
> **We do NOT attempt to say that RoPINN is an optimization method**. What we want to do is to design a new training paradigm considering the optimization process of PINN models, **which is why we name RoPINN as "region optimized PINN" not "region optimizer" in the title**.
>
> Thus, we respectfully point out that there may exist some misunderstandings in this comment.
>
> > "You propose an adaptive sampling method with different criteria than the residual."
>
> As per your request, we explain RoPINN in a simpling-based method style in the previous reply to provide a better understanding for you.
>
> However, we have to emphasize that **the contribution of region optimization is more than different criteria than residual**. As we stated in the $\underline{\text{Clarify misconception in previous rebuttal}}$, both theorem and algorithm are proposed based on the idea of considering the training process.
>
> **Thus, our key contribution is taking the optimization process into consideration and building a complete theorem and practical algorithm to implement this idea.** We believe that this new theoretical framework can be a good assistant to previous methods, which is also acknowledged by you.
>
> > "I think the manuscript would benefit from a different presentation that puts the sampling viewpoint as the core novelty. I am still against publication in the current form but now mainly because of the presentation."
>
> We do appreciate the reviewer's suggestion in the paper presentation, which gives us another perspective to think about our paper. Many thanks for the detailed descriptions in your question, which help us a lot to understand your concern.
>
> **However, we have to clarify that the sampling-based explanation is just following your request. We do hope that you can rethink our paper from our original "optimization-based" insights (see $\underline{\text{Q1-(1) in original rebuttal}}$).** In this context, current writing about region optimization theorem and practical algorithm is more fluent and smoother than sampling-based ideas. We believe that this disagreement is only about preference in academic perspectives.
>
> Besides, we also want to highlight our experiment results, **consistently improving 5 base models on 19 different PDEs with over 50% gain in most cases**. These significant results well support that our idea may be more foundational than sampling.
>
> ## Our promise in revision and respectful request to reconsider our paper presentation
>
> As we stated in the previous response, we will polish our presentation by incorporating the **newly proved theorem, experimental statistics and discussion w.r.t. previous sampling methods** into our paper, which will clearly present our paper from an "optimization-based" view. We will consider your valuable suggestions and do our best to make our paper easy to understand.
>
> With the greatest respect, we also request you to reconsider your score. We believe that **both sample-based and optimization-based viewpoints can give a reasonable explanation to our design**, where the optimization-based writing is also $\underline{\text{accepted by the other two reviewers}}$. Thus, **we kindly say that your concern in the presentation may be caused by two different academic perspectives.**
>
> Greatest thanks for your time in reviewing our paper and discussion.

---

> ### Author Response · Authors · 2024-08-14
> **More Discussion about Presentation**
>
> Many thanks for your time in reviewing our paper and new opinion about writing.
>
> For a convenient overview, we respectfully list some key reasons why we think our current writing from an "optimization" view is more suitable than sampling:
>
> **(1) All the theoretical analyses and proof of our paper are based on an optimization-based view. Previous sampling papers are under different theoretical frameworks from us.**
>
>   Here is a comparison of two different perspectives. We believe that the previous theoretical framework of sampling methods is distinct from ours, which has also been clarified in $\underline{\text{Clarify misconception of original rebuttal}}$.
>
> |  Theoretical framework            | Our Paper                            | Previous Sampling Paper (e.g. RAR)                           |
> | ------------ | ------------------------------------ | ------------------------------------------------------------ |
> | Key Question | Will the PINN model be well-trained? | Will the sampling well approximate integral?                 |
> | Theorem      | **Optimization-based theorem**   | **Quadrature-based theorems** [Siddhartha Mishra et al., IMAJNA 2023] |
>
> **(2) The optimization-based view can better present how our method works.** In our current paper, all the analyses are based on an optimization perspective. Since we do not change the collocation points (center of region) during training and only adjust the region size, the visualization of point distribution change is not suitable for us. Here is a comparison.
>
> |   Analysis                        | Our paper                                          | Previous Sampling Paper (e.g. RAR) |
> | ------------------------- | -------------------------------------------------- | ---------------------------------- |
> | Training Curve            | $\underline{\text{Figures 2,3 of main text}}$      | N/A                                |
> | Gradient std Statistics   | $\underline{{\text{Q2-(2) of original rebuttal}}}$ | N/A                                |
> | Point distribution change | N/A                                                | The most widely used analysis tool     |
>
> **(3) Our contribution is orthogonal to previous sampling papers with significant promotion**. In $\underline{\text{Table 3 of main text}}$, we present that RoPINN can be combined with RAR to further boost their performance by a great margin, which means RoPINN is from a new idea (may be more foundational). Here are part of the results.
>
> |  rMSE                        | 1D-Reaction | 1D-Wave | Convection |
> | ------------------------ | ----------- | ------- | ---------- |
> | PINN vanilla             | 0.981       | 0.335   | 0.840      |
> | +RAR                     | 0.981       | 0.126   | 0.771      |
> | +RAR+RoPINN              | 0.080       | 0.030   | 0.695      |
> | **Promotion w.r.t. RAR** | **92%**     | **76%** | **10%**    |
>
> Since the discussion period will end in a few hours, we do hope that the reviewer can reconsider our presentation.
>
> Sincerely thanks for your time and active discussion.

---

### Official Review · Reviewer_KCE6 · 2024-07-12

**Soundness:** 3
**Presentation:** 3
**Contribution:** 3
**Rating:** 6
**Confidence:** 4

**Summary:**

This paper extends the optimization process of PINNs from isolated points to their continuous neighborhood regions, which can theoretically decrease the generalization error, especially for hidden high-order constraints of PDEs. A practical training algorithm, Region Optimized PINN (RoPINN), is seamlessly derived from this new paradigm, which is implemented by a straightforward but effective Monte Carlo sampling method. By calibrating the sampling process into trust regions, RoPINN finely balances sampling efficiency and generalization error. Experimentally, RoPINN consistently boosts the performance of diverse PINNs on a wide range of PDEs without extra backpropagation or gradient calculation.

**Strengths:**

1. The idea of extending the optimization process of PINNs from isolated points to their continuous neighborhood regions is novel.
2. Theoretical results on generalization error, convergence rate and estimation error of sampling are provided.
3. A practical training algorithm, Region Optimized PINN (RoPINN), is seamlessly derived from the region optimization paradigm and associated theoretical results,
4. RoPINN consistently boosts the performance of diverse PINNs on a wide range of PDEs without extra backpropagation or gradient calculation.

**Weaknesses:**

1. It is better to include the main proof idea of theoretical results in the main text.
2. Although generalization error bound is provided, an intuitive explanation of the reason behind the success of region optimization is desirable. For example, when sampling one point in each region, why is the total loss decreased compared with point optimization?

**Questions:**

1. Which results in section 4 are for comparisons with the losses with high-order terms and variational methods?
2. How tight is the generalization error bound?

**Limitations:**

The case of sampling more than one points in each region is not discussed.

---

> ### Author Rebuttal · Authors · 2024-08-06
>
> # To Reviewer KCE6
>
> Many thanks to Reviewer KCE6 for providing the insightful review and questions.
>
> > **Q1:** "It is better to include the main proof idea of theoretical results in the main text."
>
> Following your suggestion, we will add the following descriptions into the main text as a brief proof for $\underline{\text{Theorem 3.5}}$:
>
> "The proof is based on an optimization perspective [13 of our paper]. Firstly, based on the Lipschitz assumption, we can transform the generalization bound to $L$ times the expectation of parameter distance. Further, region optimization paradigm will bring more 'consistent' gradient optimization direction than point optimization at each iteration, thereby benefitting the generalization bound."
>
> > **Q2:** "Although generalization error bound is provided, an intuitive explanation of the reason behind the success of region optimization is desirable. For example, when sampling one point in each region, why is the total loss decreased compared with point optimization?"
>
> We will clarify this question from both theoretical and optimization views.
>
> **(1) Theoretical view: generalization bound for practical algorithm.**
>
> Just as the reviewer mentioned, the generalization bound is based on an ideal assumption, that is, we can accurately obtain the loss of a region. However, in practice, as the reviewer mentioned, we have to use some approximation methods.
>
> To provide a more direct understanding, we also prove the generalization bound for practical algorithm in the $\underline{\text{Figure 3 of Global Response PDF}}$. This new theorem is extended from $\underline{\text{Theorem 3.5 of main text}}$ by considering the gradient estimation error caused by Monte Carlo approximation.
>
> For clarity, we present the convex version here. See the $\underline{\text{Global Response PDF}}$ for the non-convex version and proof.
> $$
> \text{Generalization\ error}\leq ((1-|\Omega_r|/|\Omega|)L+\mathcal{E}_{r, \mathrm{grad}})\frac{2L}{|\mathcal{S}|}\sum\alpha_t
> $$
>
>
> $\mathcal{E}_{r, \mathrm{grad}}$ denotes the gradient estimation error caused by approximation methods, which is generally proportional to $|\Omega_r|$. We have the following observations:
>
> - **Canonical point optimization corresponds to an extreme case:** $|\Omega_r|=0$ and $\mathcal{E}_{r, \mathrm{grad}}=0$. In this case, the above equation degenerates to the bound that we proved in $\underline{\text{Theorem 3.3}}$.
> - RoPINN presents a trust region calibration algorithm, which can **finely balance benefits from region loss** (the first term $(1-\frac{|\Omega_r|}{|\Omega|})$) **and gradient estimation error** (the second term $\mathcal{E}_{r,\mathrm{grad}}$), which leads to a lower generalization bound.
>
> Thus, the success of RoPINN is from our trust region calibration, which can achieve a better balance between generalization bound and gradient estimation error. We also have visualized the balancing process (change of region size during training) in $\underline{\text{Figure 2 of main text}}$.
>
> **(2) Optimization view: A multi-iteration understanding of why sampling one point works.**
>
> Although we only sample one point at each iteration, $\underline{\text{Lemma 3.10 of main text}}$ shows that during training, gradients difference on different sampling points of $\underline{\text{successive iterations}}$ will be close to different sampling points at the $\underline{\text{same iteration}}$.
>
> This means (informally) that if we think about the training process in a multi-iteration view, the optimization is getting closer to sampling multiple points during training, despite these sampled points being dispatched into multiple successive iterations.
>
> > **Q3:** "Which results in section 4 are for comparisons with the losses with high-order terms and variational methods?"
>
> As stated in $\underline{\text{Line 232 of main text}}$, gPINN is with high-order regularization and vPINN is based on variational formulation. We have compared them in every experiment of $\underline{\text{Table 2 of main text}}$, which provides solid support for the advancement of RoPINN over them.
>
> > **Q4:** "How tight is the generalization error bound?"
>
> Firstly, we would like to thank the reviewer's great question, which makes us calibrate the proof of $\underline{\text{Theorem 3.5}}$ and obtain a tighter generalization bound by refining one-step derivation.
>
> See $\underline{\text{Figure 2 of Global Response PDF}}$ for the complete refined theorem and proof. In short, the refined bound is:
> $$
> \text{Generalization\ error}\leq (1-|\Omega_r|/|\Omega|)\frac{2L^2}{|\mathcal{S}|}\sum\alpha_t
> $$
>
> We think this generalization bound is quite tight. All the derivations of the proof can be strictly equal in some cases. Furthermore, if we consider two extreme cases, this generalization bound works perfectly:
>
> - $\Omega_r=0$, this bound degenerates to the bound of point optimization ($\underline{\text{Theorem 3.3 of main text}}$).
> - $\Omega_r=\Omega$ means directly optimizing the whole domain, which corresponds to the ideal loss function. In this case, our proved generalization bound is just equal to 0. Note that, as discussed in $\underline{\text{Q2}}$, this case will still face a large optimization error in practice.
>
> We will also update $\underline{\text{Theorem 3.5}}$ as the refined version in the revised paper.
>
> > **Q5:** "The case of sampling more than one point in each region is not discussed."
>
> Actually, we have discussed sampling more points in $\underline{\text{Figure 3 of main text}}$ as one of the main analysis experiments.
>
> Further, during rebuttal, we also provide more comprehensive results in the $\underline{\text{Figure 1 of Global Response PDF}}$. The key finding is that (1) computation costs will grow linearly when adding points; (2) more points will bring better performance but will saturate around 10 points.

---

> > ### Comment · Reviewer_KCE6 · 2024-08-12
> >
> > Thank the authors for their replies. Aftering reading the authors' responses to me and other reviewers, especially why introducing gradient estimation error is benefit and the comparision between RoPINN and other sampling method,  I got the rough idea of why region optimization works, though I did not check the proof details.  "A larger gradient estimation error (corresponding to a lower consistency of gradients in successive iterations) will lead to an unstable training process", this seems to be related with the loss landscape of PINNs, how does this gradient consistency promote optimization convergence considering the smoothness property of loss landscape?
> >
> > I am satisfied with the answers to my other questions. I would like to keep my score, and suggest the authors to explain in detail the intuition behind RoPINN  and the  difference w.r.t. previous sampling methods in the updated submission.

---

> ### Author Response · Authors · 2024-08-12
> **Thanks for your response and acknowledge our rebuttal**
>
> We would like to sincerely thank you for your valuable response and suggestions.
>
> (1) As for "A larger gradient estimation error will lead to an unstable training process", as we stated in the response to Reviewer zNBp, this is just an intuitive understanding. Here are more details.
>
> Please recall the optimization algorithm SGD and GD. SGD generally faces more optimization fluctuations, thereby converging more slowly than GD. Similarly, if the gradient among successive iterations has a low consistency, the optimization direction of PINN models will face fluctuations, which will make it harder to arrive at a convergence point. If the gradient among successive iterations is consistent, the optimization direction is confirmed, thereby converging faster.
>
> (2) As for "considering the smoothness property of loss landscape", we think the loss landscape will affect the gradient consistency, where the gradient consistency is a dependent variable. Suppose that the target PDE is easy to solve, RoPINN can learn to explore larger region size $r$ in pursuing a better balance between generalization bound and optimization (gradient consistency).
>
> Following your suggestion, we will add more discussion about the intuition behind RoPINN (including the newly proved theorem for practical algorithm) and the difference w.r.t. previous sampling methods in the revised paper.
>
> Sincerely thanks for your time in reviewing our paper.

---

### Official Review · Reviewer_mhfV · 2024-07-12

**Soundness:** 4
**Presentation:** 3
**Contribution:** 3
**Rating:** 6
**Confidence:** 4

**Summary:**

The authors developed a region optimized PINN to improve the prediction accuracy compared to the scatter-point based PINN.

**Strengths:**

The authors proposed the region optimization paradigm and conducted a theoretical analysis.

**Weaknesses:**

The practical application scope is limited.

**Questions:**

(1)  It is suggested to add some descriptions of training difficulty factors for the canonical PINN on 1D-Reaction in Section 4.2.
(2)  Can the proposed method find a good number of sampling points well balancing the computational cost and convergence speed in Figure 3.
(3)  The motivation of using Monte Carlo approximation should be elaborated. Why don’t the authors choose some other more advanced methods to provide better accuracy.
(4)  The authors should add more details about the possible practical applications with the canonical loss function of L-Lipschitz-β-smooth.

**Limitations:**

Some initial guess methods can be developed to efficiently determine the preferable region size and sample number.

---

> ### Author Rebuttal · Authors · 2024-08-06
>
> # To Reviewer mhfV
>
> We sincerely thank Reviewer mhfV for providing valuable feedback and suggestions in new experiments.
>
> > **Q1:** "Add some descriptions of training difficulty factors for the canonical PINN on 1D-Reaction in Section 4.2."
>
> We will add "Previous research [22 of our paper] demonstrates that 1D-Reaction contains sharp transitions, which is hard to approximate" into $\underline{\text{Section 4.2}}$.
>
> > **Q2:** "Can the proposed method find a good number of sampling points well balancing the computation and convergence."
>
> Number of sampling points is set manually. Our method can only adaptively adjust region size.
>
> **(1) The official design of RoPINN is sampling 1 point in each region, which is already a good and well-verified choice.**
>
> Note that RoPINN is proposed in the spirit of boosting PINNs **without extra backpropagation or gradient calculation (see $\underline{\text{Abstract}}$).** Thus, as described in $\underline{\text{Section 3.2 and Algorithm 1}}$, **we only sample 1 point in practice and adopt this as the official design.**
>
> $\underline{\text{Tables 2,3}}$ demonstrate that this official setting achieves significant promotion and comparable efficiency w.r.t. diverse PINNs in all 19 tasks. So, as for your question, we think sampling 1 point is already a good choice in balancing performance and efficiency.
>
> **(2) Discussion about sampling more points.**
>
> Since the convergence of deep models is affected by many factors (e.g. task, base model, optimizer), we cannot give a universal choice of sampling points. But we experiment RoPINN with more sampling points and plot an overview curve in $\underline{\text{Figure 1 of Global Response PDF}}$. Here are part of the results.
>
> | 1D-Reaction      | RMSE  | 100 iters time |
> | - | - | - |
> | Vanilla PINN     | 0.981 | 18.47s         |
> | RoPINN 1 Point   | 0.095 | 20.04s         |
> | RoPINN 5 Points  | 0.050 | 46.48s         |
> | RoPINN 9 Points  | 0.033 | 67.98s         |
> | RoPINN 13 Points | 0.035 | 92.48s         |
> | RoPINN 30 Points | 0.037 | 196.41s        |
>
> We can observe that adding points will bring better performance but will saturate around 10 points. The performance fluctuations of 9, 13, 30 points are within three times the standard deviations (Appendix D.4).
>
> Thus, according to our experiments, we first recommend sampling 1 point (our official design). If they want to obtain better performance, try [1, 10].
>
> > **Q3:** "The motivation of using Monte Carlo. Why don’t the authors choose some other more advanced methods."
>
> **Again, RoPINN is proposed to boost PINNs without extra backpropagation or gradient calculation (see $\underline{\text{Abstract}}$).** Monte Carlo approximation can work well with 1 sampling point, which is easy to implement and without extra gradient calculation. As for other advanced methods, they usually need more points to achieve an accurate approximation.
>
> We experiment with 2D space Gaussian quadrature, which requires square number points and the 1-point situation will degenerate to the center value. Here are the results of PINN+RoPINN.
>
> | 1D-Reaction rMSE | Monte Carlo | Gaussian Quadrature |
> | - | - | - |
> | 1 point   | **0.095**   | 0.109               |
> | 4 points  | 0.066       | **0.059**           |
> | 9 points  | 0.033       | **0.030**           |
>
> We can find that under our official setting (sample 1 point), Monte Carlo is better. Although Gaussian quadrature is better in more points, these settings defeat our purpose in "no extra gradient calculation".
>
> > **Q4:** "The practical application scope is limited." "More details about the possible practical applications with the canonical loss function of L-Lipschitz-β-smooth."
>
> **(1) Theoretical assumption will not affect the practicability of algorithms.**
>
> A typical example is Adam, whose convergence can only be analyzed under strict assumptions, e.g. convex or L-Lipschitz-β-smooth [1]. However, it has been used as a foundation optimizer.
>
> In our paper, the L-Lipschitz-β-smooth assumption is only in theoretical analysis for a basic understanding of our paradigm and will not affect the practicability of RoPINN.
>
> [1] Implicit Bias of AdamW: ℓ∞ Norm Constrained Optimization, ICML 2024
>
> **(2) RoPINN achieves consistent promotion for 5 diverse backbones, covering 19 tasks ($\underline{\text{Tables 2,3,7,8}}$).**
>
> Our experiments are much more extensive than the latest papers, such as PINNsFormer (ICLR 2024) and KAN (arXiv 2024). As described in $\underline{\text{Appendix C.1}}$, we experiment with diverse PDEs and do not constrain the loss function to be L-Lipschitz-β-smooth in practice. We believe that such extensive experiments strongly support the practicability of RoPINN.
>
> **(3) Our assumption is widely used in other theoretical papers.**
>
> The papers that we cited in $\underline{\text{Theorem 3.3 of main text}}$ for point optimization analysis are based on L-Lipschitz-β-smooth assumption [13, 47 in our paper]. Other papers that we cited in $\underline{\text{Section 4 of main text}}$ also assume the smoothness of loss function [20 in our paper] or even "inﬁnitely wide neural networks" [42 in our paper].
>
> > **Q5:** Some initial guess methods can be developed to efficiently determine the region size and sample number.
>
> **As stated in $\underline{\text{Section 3.2 and 4 of main text}}$, we adopt the same hyperparameter setting (initial region size $r=10^{-4}$, sample 1 point) for all benchmarks, across 19 different tasks and 5 different base models, which works well consistently.** Thus, this configuration can be a good initial guess.
>
> Besides, we can obtain some empirical guidance from:
>
> - $\underline{\text{Figure 2}}$: Region size $r$ will be progressively adjusted by RoPINN. Setting $r$ in $[10^{-6}, 10^{-4}]$ can work well.
> - $\underline{\text{Figure 3}}$ and results in **Q2**: Sampling 1-30 points can gain consistent promotion.
>
> As discussed in $\underline{\text{Appendix G}}$, some tools (e.g. Weights and Bias) can also be helpful.

---

> > ### Author Response · Authors · 2024-08-13
> > **Summary of rebuttal to Reviewer mhfV**
> >
> > We sincerely thank your dedication in reviewing our paper.
> >
> > Since this is the last day of the review-author discussion period, we summarize the key points of our rebuttal as follows for a convenient overview:
> >
> > - **Clarify the concern about practical application scope:** (1) We highlighted that "Theoretical assumption will not affect the practicability of algorithms". (2) The extensive experiment results (5 base models on 19 PDEs) have well supported RoPINN's practicability. (3) Our theoretical assumption is widely used in other papers.
> > - **Add new experiments about more sampling points:** We highlight that only simpling 1 point is already a good choice and also add experiments on more points to provide an overview of RoPINN.
> > - **Add comparison w.r.t. advanced quadrature methods:** We show that the advanced methods need more points to achieve a better performance, which defeats our purpose in "no extra gradient calculation".
> > - **Give practical guidance to hyperparameters** based on analysis experiments in the main text.
> >
> > **Sincerely thank you for ranking our paper with "excellent" soundness, "good" presentation and "good" contribution**. We do hope our rebuttal can fully resolve your questions to your satisfaction. If so, with the greatest respect, we also hope you can kindly consider raising the score correspondingly.
> >
> > Many thanks for your time and looking forward to your response.

---

### Official Review · Reviewer_aU22 · 2024-07-12

**Soundness:** 3
**Presentation:** 3
**Contribution:** 3
**Rating:** 6
**Confidence:** 4

**Summary:**

The paper proposes a novel optimization method for training physics-informed neural networks (PINNs): Region optimization, which extends a regular pointwise optimization of PINNs to neighborhood regions, named RoPINNs. The paper provides theoretical analysis explaining the decrease of generalization error with RoPINNs and high-order PDE constraints satisfaction. Then the paper presents a practical algorithm to enable the RoPINNs paradigm by introducing Monte-Carlo approximation of region integral and a region calibration scheme. Lastly, the paper assesses the performance on several well-known benchmark problems and showed the improved performance over the considered baselines.

**Strengths:**

- The paper is well-motivated and tackles the important problem in training PINNs (leveraging more information than just a point-to-point type mapping).

- The paper presents theoretical analysis on benefits of RoPINNs, decreased generalization errors and satisfaction of higher-order PDE constraints.

- The experimental results show that the proposed algorithm is effective in solving some challenging benchmark problems (known as failure modes) and is capable of producing more accurate solution approximates.

**Weaknesses:**

- Although shown to be very effective in several benchmark problems, the paper does not seem to provide general guidelines on how to set some important hyper-parameters such as initial region size and the number of sampling points. (While acknowledging that the authors indicate this as one of the limitations,) it would be great to see some experts’ guidelines.

- If the authors could provide some analysis with regards to computational wall time, that would provide more complete pictures on how the proposed method performs. For example, it would be information to see a figure depicting a scatter plot showing computational wall time versus final rMSE type information, where a point in the plot corresponds to a different hyper-parameter setting (that is, the number of sample points).

- [minor] there is a typo in the second paragraph of Section 4.2: line 289 Figure 2 => Figure 3.

**Questions:**

- Eq (5) seems to suggest that region optimization is applied to the boundary condition as well as L = L_bc + L_ic + L_pde. Is this the correct understanding or is it a typo?

- The proposed optimization method seems to benefit significantly in the case of 1D reaction case while the benefit in 1D Wave or 1D convection cases are not as significant as that of 1D reaction. That is, rMSE for example in Table 2 of 1D reaction achieves an order (or orders) of magnitude improvement over the second best performing methods. Do the authors have some explanation on why?

**Limitations:**

- some more discussions on practical guidelines would be needed for users who want to utilize this method in different applications

- some additional experiments (regarding computational wall time) would be needed to provide a complete picture of the proposed method.

---

> ### Author Rebuttal · Authors · 2024-08-06
>
> # To Reviewer aU22
>
> We would like to sincerely thank Reviewer aU22 for providing a detailed review and insightful questions.
>
> > **Q1:** "The paper does not seem to provide general guidelines on how to set some important hyper-parameters. It would be great to see some experts’ guidelines." "More discussions on practical guidelines would be needed."
>
> Many thanks for reviewer's valuable suggestions.
>
> **At first, we would like to highlight that we adopt the same hyperparameter setting (initial region size $r=10^{-4}$, sample 1 point) for all benchmarks, across 19 different tasks and 5 different base models, which works well.** This verifies the consistent effectiveness of our algorithm.
>
> Next are some guidelines for RoPINN. We will include them in the revised paper.
>
> **(1) Experiment guidance: recap hyperparameter analyses in our paper.**
>
> We have provided analyses on every hyperparameter of RoPINN in $\underline{\text{our original submission}}$, which deliver the following empirical guidance:
>
> - $\underline{\text{Figure 2 of Section 4.2}}$: Initial region size $r$ will be progressively adjusted during training.
> - $\underline{\text{Figure 3 of Section 4.2}}$: Increasing the number of sampling points will speed up the convergence but will bring more computation costs (see $\underline{\text{Q2}}$ for efficiency comparison).
>
> **(2) Theorem guidance: a new generalization bound for practical algorithm.**
>
> To provide a more direct understanding of the practical algorithm, we prove a new theorem in the $\underline{\text{Figure 3 of Global Response PDF}}$, which considers the gradient estimation error:
>
> $\text{Generalization\ error}\leq \big((1-|\Omega_r|/|\Omega|)L+\mathcal{E}_{r, \mathrm{grad}}\big)\frac{2L}{|\mathcal{S}|}\sum\alpha_t$
>
> $\mathcal{E}_{r, \mathrm{grad}}$ denotes the gradient estimation error. Here are the guidances derived from this new theorem:
>
> - Increasing the initial region size can benefit the first term of the bound, but may increase gradient estimation error. Fortunately, it can be adaptively adjusted by RoPINN during training, making this hyperparameter easier to tune.
> - Sampling more points can reduce the gradient estimation error.
>
> **(3) Practical suggestions.**
>
> Based on experiments and the new theorem, we suggest that
>
> - Firstly, researchers can use the same configuration as us, whose effectiveness has already been verified in 19 tasks and diverse base models. Specifically, as stated in $\underline{\text{Section 3.2 and 4 of main text}}$, initial region size $r=10^{-4}$ and number of sample points $=1$.
> - Further, they can adjust the initial region size according to the training loss curve. A jitter curve indicates that you should decrease the initial region size value. Besides, the number of sampling points should be set according to the efficiency demand, where more points will generally bring better results.
> - As discussed in $\underline{\text{Limitations in Appendix G}}$, some tools (e.g. Weights and Bias) can be helpful.
>
> > **Q2:** "It would be information to see a figure depicting a scatter plot showing computation time versus rMSE type information." "Experiments (computational wall time) would be needed to provide a complete picture of the proposed method."
>
> Firstly, we want to emphasize that **RoPINN is proposed in the spirit of boosting PINNs without extra backpropagation or gradient calculation (see $\underline{\text{Abstract}}$).** Thus, we only sample 1 point in all experiments, which has already achieved significant promotion and comparable efficiency w.r.t. PINNs.
>
> As per the reviewer's request, we plot the performance and efficiency under different numbers of points in $\underline{\text{Figure 1 of Global Response PDF}}$. Here are the results of 1D-Reaction. 1D Wave is also included in the $\underline{\text{PDF}}$.
>
> We can find that (1) computation costs will grow linearly when adding points; (2) more points will bring better performance, but will saturate around 10 points. The performance fluctuations of 9, 13, and 30 points are within three times the standard deviations (Appendix D.4).
>
> | Number of Points | RMSE  | 100 iters time | GPU Memory |
> | - | - | - | - |
> | PINN             | 0.981 | 18.47s         | 1.44GB     |
> | RoPINN 1 Point   | 0.095 | 20.04s         | 1.48GB     |
> | RoPINN 5 Points  | 0.050 | 46.48s         | 4.24GB     |
> | RoPINN 9 Points  | 0.033 | 67.98s         | 6.74GB     |
> | RoPINN 13 Points | 0.035 | 92.48s         | 9.24GB     |
> | RoPINN 30 Points | 0.037 | 196.41s        | 19.80GB    |
>
> > **Q3:** "A typo in the second paragraph of Section 4.2: line 289 Figure 2 => Figure 3."
>
> Many thanks for your detailed review. We will correct this in the revised paper.
>
> > **Q4:** "Eq (5) seems to suggest that region optimization is applied to the boundary condition as well as L = L_bc + L_ic + L_pde. Is this the correct understanding or is it a typo?"
>
> Your understanding is correct. As described in $\underline{\text{Lines 113-115}}$, region optimization is applied to the inner domain, boundary and initial conditions. However, we will restrict all the sampled points within the definition domain of the PDE. We will include these details and rephrase Eq (5) in the revised paper.
>
> > **Q5:** "The proposed method seems to benefit significantly in the case of 1D reaction while the benefit in 1D Wave or 1D convection cases are not as significant as that of 1D reaction."
>
> Since we experiment with RoPINN on diverse base models and various PDEs, **the value of relative promotion is affected by both base model capability and PDE difficulty,** which makes comparisons among different tasks a large variation.
>
> For example, for the base model PINN, the promotion on 1D-Reaction is larger than 1D-Wave and Convection. In contrast, as for KAN, the promotions on 1D-Wave and Convection are larger than 1D-Reaction. Promotions for PINNsFormer on three tasks are close to each other.
>
> Thus, comparing relative promotion values among different tasks may be meaningless.

---

> > ### Comment · Reviewer_aU22 · 2024-08-12
> >
> > Thank you for sharing the new results. They provide useful insight into the tradeoff between additional computation and accuracy. I remain positive about the paper and will maintain the current score.

---

> ### Author Response · Authors · 2024-08-13
> **Thanks for your response and positive support**
>
> Thanks for your prompt response and for acknowledging our new experiments in computation-accuracy balancing. Your positive support and valuable suggestions help us a lot in revising our paper. We will include all the new results in the revised paper.

---

### Author Rebuttal · Authors · 2024-08-06

## Global Response and Summary of Revisions

We sincerely thank all the reviewers for their insightful reviews and valuable comments, which are instructive for us to improve our paper further.

This paper proposes and theoretically studies a **new training paradigm as region optimization**, which can benefit both generalization bound and hidden high-order constraints of PDEs. RoPINN is derived from the theory as a practical training algorithm, which **consistently boosts the performance of 5 different PINN models on 19 PDEs without extra backpropagation or gradient calculation**. Detailed visualizations and theoretical analysis are provided.

The reviewers generally held positive opinions of our paper, in that the proposed method is "**novel**", "**practical**", "**well-motivated**", "**tackles an important problem**", and "**effective in solving some challenging problems**"; we have experimented "**on a wide range of PDEs**" and shown "**good empirical performance**".

The reviewers also raised insightful and constructive concerns. We made every effort to address all the concerns by providing detailed clarification, requested results and theorems. Here is the summary of the major revisions:

- **Provide practical guidelines for initial region size, number of sampling points (Reviewer aU22, mhfV):** Firstly, we highlight that our experiments are all under the same hyperparameter setting, which can be a good and widely-verified choice for initial guess. Secondly, we recall the hyperparameter analysis experiments and newly proved theorem to show both empirical and theoretical guidance. The requested experiment on more sampling points is provided for an overview of RoPINN.
- **Experiment with more advanced approximation methods (Reviewer mhfV):** Firstly, we emphasize the design principle of RoPINN: do not bring extra gradient calculation, which motivates us to choose the easy-to-implement Monte Carlo method. Secondly, we provide experiments on Gaussian quadrature, which needs more sampling points for an accurate approximation.
- **Explain how tight the generalization bound is (Reviewer KCE6):** Following the reviewer's question, we calibrate the proof of our theorem and obtain a tighter bound by only updating one-step derivation. The refined theorem is quite tight and can seamlessly cover two extreme cases. We will include the refined theorem in the revised paper.
- **Explain why our method works theoretically (Reviewer KCE6, zNBp):** Following the reviewer's request, we prove a new theorem for practical algorithm. By considering the gradient estimation error caused by MC approximation, we demonstrate that point optimization is just a special case of our theorem, which lacks flexibility. In contrast, RoPINN can adaptively balance generalization and optimization errors.
- **Explain the meaning of our region optimization theorem (Reviewer zNBp):** Firstly, we clarify that our theorem considers the training process, which may result in some misconceptions. Secondly, we show that our region optimization theorem provides a unified theoretical framework for different optimization paradigms and can be easily extended to practical algorithms.

The valuable suggestions from reviewers are very helpful for us to revise the paper to a better shape. All the above revisions will be included in the final paper. We'd be very happy to answer any further questions.

Looking forward to the reviewer's feedback.

### **The mentioned materials are included in the following PDF file.**

- **Figure 1 (Reviewer aU22, mhfV)**: Efficiency & performance w.r.t. number of sampling points.
- **Figure 2 (Reviewer KCE6, zNBp)**: Refined Theorem 3.5: A tighter generalization bound.
- **Figure 3 (Reviewer KCE6, zNBp)**: New Theorem: Generalization bound under Monte Carlo approximation.

Due to the compilation difficulty, we can only provide a brief description of new theorems in the OpenReview reply. A formal version is in the PDF.

---

### Decision · Program_Chairs · 2024-09-25

**Decision:**

Accept (poster)

**Comment:**

The submission has received mixed ratings after extensive post-rebuttal discussion between the authors and the referees. One of the referees recommends rejection based mainly on presentation aspects. The other three referees recommend acceptance, and they also rate the presentation as sufficiently clear. The authors are advised to take into account the detailed reviews as well as the clarifications provided in their rebuttal while revising the submission.